# Synergy between compost and cover crops in a Mediterranean row crop system leads to increased subsoil carbon storage

Daniel Rath[1], Nathaniel Bogie[2], Leonardo Deiss[3], Sanjai J. Parikh[1], Daoyuan Wang[4], Samantha Ying[5], Nicole Tautges[6], Asmeret Asefaw Berhe[7], Teamrat A. Ghezzehei [7], Kate M. Scow[1]

[1]Department of Land, Air and Water Resources, University of California Davis, Davis, CA 95618, USA
[2]Department of Geology, San Jose State University, San Jose, CA 95192, USA
[3]School of Environment and Natural Resources, The Ohio State University, Wooster, OH 44691, USA
[4]Department of Environmental Science and Engineering, Shanghai University, Shanghai, 200444, CHN
[5]Department of Environmental Sciences, University of California Riverside, Riverside, CA 92521, USA
[6]Michael Fields Agricultural Institute, East Troy, WI 53120, USA
[7]Department of Life and Environmental Sciences, University of California Merced, Merced, CA 95342, USA

*Correspondence to*: Daniel Rath (darath@ucdavis.edu)

**Abstract.** Subsoil carbon (C) stocks are a prime target for efforts to increase soil C storage for climate change mitigation. However, subsoil C dynamics are not well understood, especially in soils under long term intensive agricultural management. We compared subsoil C storage and soil organic matter (SOM) composition in tomato-corn rotations after 25 years of differing C and nutrient management in the California Central Valley: CONV (mineral fertilizer), CONV+WCC (mineral fertilizer + cover crops) and ORG (composted poultry manure + cover crops). The cover crop mix used in these systems are a mix of  oat (*Avena sativa* L.), faba bean (*Vicia faba* L.) and hairy vetch (*Vicia villosa* Roth). Our results showed a ~19 Mg/ha increase in SOC stocks down to 1m under ORG systems, no significant SOC increases under CONV+WCC or CONV systems, and an increased abundance of carboxyl rich C in the subsoil (60-100 cm) horizons of ORG and CONV+WCC systems. Our results show the potential for increased subsoil C storage with compost and cover crop amendments in tilled agricultural systems, and identify potential pathways for increasing C transport and storage in subsoil layers.

## 1 Introduction

Agricultural subsoils (>60cm) have the potential to store large amounts of C (Rumpel et al., 2012), for a longer period of time (Paul et al., 1997, 2001) relative to surface soils (<15cm). Surface soils are much easier to sample than subsoils, and respond more quickly to management, which makes them the focus of most studies of how soil organic C (SOC) is formed and stored. However, an increased focus on interrogating the surface soil to answer questions about processes in the entire soil profile exacerbates the risk of subsoils being treated merely as "more dilute surface soils" (Salomé et. al 2010) and ignores decades of research into the unique role that subsoils play in increasing soil C stocks (Rapalee et al., 1998; Rumpel and Kögel-Knabner, 2011). A focus on surface soils is particularly problematic in agricultural studies, given how practices such as cover cropping can have drastically different effects on surface versus subsoil SOC accumulation (Bernal et al., 2016; Harrison et al., 2011; Tautges et al., 2019) depending on the cover crop used. In addition, recent studies have highlighted that subsoil SOC may be vulnerable to loss under changing environmental conditions, such as warming (Hicks Pries et al., 2018) and drought (Min et

al., 2020). To maximize C stored in the entire soil profile, we need to understand and capitalize on the numerous and interacting physical, chemical, and biological changes throughout the profile (Angst et al., 2018; Fierer et al., 2003b; Kautz et al., 2013).

The fact that the average soil sampling depth has decreased from 53 to 27 cm in studies published in the last 30 years (Yost and Hartemink, 2020) may be based on ease of sampling and a focus on surface microbiological processes, but also due to the lack of agreement on where surface soils end and subsoils begin. Depending on the goals of the study, the lower limit of surface soils may be anywhere between the top 7.5 and top 30 cm of the profile, while the upper limit of subsoils can range anywhere from 20-100cm (Soong et. al 2021, Chen et. al 2018, Lorenz and Lal 2005, Whitmore et. al 2015). There may also be an "intermediate" or "transition" zone that is operationally defined, often corresponding to the maximum tillage depth (Mobley et. al 2015). In this study, we define surface soils as the top 0-15 cm of the profile, and subsoils as the lower 60-100 cm, with the intervening 15-60 cm as an intermediate zone based on previous work carried out at our study site, and the relative lack of horizon formation in these young soils.

The unique role that subsoils play in storing SOC is due in part to the extensive, site-specific changes that happen across the soil profile. Often, there are changes in bulk density and mineralogy due to clay accumulation, but the exact magnitude and direction of this change varies depending on depositional environment and soil forming factors (Brady and Weill 2015, Soil Survey Staff 2014, Jenny 1941). Subsoils also experience much less disturbance than surface soils, with lower fluctuations in temperature and moisture content (Smitii 1932, Cole and Matthews 1939, Zeynoddin et. al 2019, de Quieroz et. al 2020) and less mechanical disturbance such as tillage (though it is important to note that tillage events deeper than 30 cm are not altogether rare in many systems). Inputs of oxygen, water, C, and nutrients are usually lower to subsoil than surface soils and mostly occur via transport through the soil pore network constructed from intra-aggregate pore spaces, root channels, and cracks that form as the soil dries (Pagliai 2004, Sanderman and Amundson 2008). The types of C input are much less varied in subsoils, mostly coming from biomass and exudates of plants with deep roots (Sokol and Bradford, 2019) and transport of dissolved organic carbon (DOC). This downward transport of DOC is described by the "cascade theory" (Kaiser and Kalbitz, 2012), where subsoil DOC inputs undergo a series of successive sorption, desorption, microbial processing, and transport steps. This results in a gradual increase in the age of C as we move through the soil profile, with subsoil C molecules as old as $10^3$-$10^4$ years (Rumpel et. al 2012) compared to younger $C_{14}$ ages of $10^2$-$10^3$ years in the top 30 cm. The transport of C into the subsoil via both roots and DOC movement leads to more heterogeneous C distribution (Chabbi et al., 2009; Syswerda et al., 2011) that is closely associated with the soil pore network.

Since the soil pore networks responsible for dissolved carbon transport are also hotspots of microbial activity (Banfield et. al 2017), C molecules found in the subsoil have often undergone extensive microbial transformation and processing. However, once that C does enter the subsoil, it is less likely to undergo further microbial processing due to the combination of heterogeneous C distribution, decoupled microbial-carbon presence (Dungait et al., 2012), greater metabolic and physical restrictions on C decomposition (Fierer et al., 2003a) and lower microbial biomass (Taylor et al., 2002). This leads to higher concentrations of simpler, microbially-derived carbohydrates, aliphatics and carboxylates in subsoils, in contrast to the more complex aromatic structures in cellulose and lignin present in surface soils (Roth et. al 2019). These simple microbial products may preferentially associate with mineral surfaces (Samson et. al 2020, Williams et. al 2018), driving the formation of mineral-associated organic matter and further rendering that C inaccessible to microbes. Subsoil microbial communities have adapted to this relative scarcity of C and nutrients (Salomé et al., 2010; Sanaullah et al., 2011) by increasing the proportion of Gram+ bacteria, whose thicker cell walls make them more resilient to adverse environmental conditions. These subsoil microbes may also optimize for survival rather than population growth, being less efficient at C assimilation than surface microbes (Spohn et al., 2016), and thus more likely to mineralize SOC to $CO_2$. Low C use efficiency would also be expected in soils with unfavourable carbon-nutrient stoichiometry for biomass production (Ng et. al 2014, Coonan et. al 2020).

Using existing methods to examine subsoils under different land management practices can help explain how the size and concentration of soil C stocks are related to types of C input and the status of the soil microbial community (Sradnick et. al 2014). While it is difficult to accurately estimate whether the C, nutrient and water status of a particular subsoil will promote or hamper microbial SOC decomposition (Soong et. al 2020); some insight can be obtained by looking at microbial stress levels. Phospholipid fatty acid analysis (PLFA) targets metabolically active cells in soil (Zhang et. al 2019) and is effective

in measuring rapid changes in active microbial cell walls and membranes (Frostegård et. al 2010). Measurements of Gram negative : Gram positive ratios via PLFA agree with those obtained via the more recent 16s rRNA metabarcoding (Orwin et. al 2018) and are useful as an indicator of microbial nutrient limitation under different land management practices. Understanding how these practices then affect the molecular composition of soil organic matter (SOM) is more difficult, as the most accurate method for quantifying specific C functional groups in soil (nuclear magnetic resonance, NMR) is sensitive to C concentrations, the presence of iron oxides, and requires extensive sample preparation if samples contain low C concentrations or a high abundance of paramagnetic species (Fe, Mn) (Bleam 1991, Smernik and Oades 2022). However, Fourier transform infrared spectroscopy (FTIR) presents a rapid, lower-cost method that allows pseudo-quantification of the relative abundance of certain carbon functional groups (Margenot et. al 2016). While FTIR may also be used as a high-throughput method to predict soil properties (Dangal et. al 2021, Deiss et. al 2020), it is particularly useful when comparing changes in SOM structure over time via spectral subtractions (Margenot et. al 2019).

Agricultural practices can increase or decrease subsoil SOC by modifying the physical, chemical and biological processes that control microbial mineralization of soil C including occlusion in soil aggregates, sorption to soil minerals, microbial processing of residues and C transport into the subsoil (Rumpel and Kögel-Knabner, 2011). Crop root exudates can be efficiently transformed by microbes into stable soil C (Sokol and Bradford, 2019), but the same exudates can also destabilize aggregates and carbon-mineral bonds that are key for protecting C from mineralization (Keiluweit et al., 2015). Large inputs of dissolved organic C and nutrients can prime subsoil microbial biomass to decompose native SOC (Bernal et al., 2016; Kuzyakov, 2010), or provide the nutrients needed for microbes to process soil C (Coonan et al., 2020; Kirkby et al., 2013) and promote the formation of mineral-associated organic matter (MAOM) (Lavallee et. al 2020). Cover crops may not only increase soluble organic C inputs (Steenwerth and Belina, 2008) but can also influence C dynamics indirectly by increasing soil macroporosity and pore connectivity (Scott et al., 1994, Haruna et al., 2018; Çerçioğlu et al., 2019; Gulick et al., 1994), as well as increasing topsoil disturbance due to the processes of planting, mowing, and incorporation. These indirect cover crop effects can lead to increases in both infiltration and hydraulic conductivity in fine textured soils and potential increases in soluble C transport, particularly over longer time scales. It is clear that to accurately predict whether a specific farming practice will increase or decrease subsoil SOC storage in a changing climate, it is necessary to perform studies that explicitly examine deeper soils.

Given that small, cumulative subsoil management impacts may take decades to become detectable, impacts of agricultural management practices may not be detectable in the two to three year focus of most agronomic field studies (Dick, 1992; Johnston and Poulton, 2018; Keel et al., 2019). Additionally, measurements of soil C and available nutrients may be highly variable throughout the year (Wuest 2014, Drenovsky et. al 2004), necessitating sampling at multiple timepoints. We conducted our study at the Century Experiment at the Russell Ranch (RR) Sustainable Agricultural Facility in Davis, CA, where inputs and management history have been tracked over the last 25 years and are representative of row crop systems of the California Central valley (Wolf et al., 2018). This agricultural region is one of the world's most productive (Pathak et al., 2018) and is quite susceptible to negative impacts of climate warming and subsequent C losses (Medellín-Azuara et al., 2011). A previous study at the Century Experiment found that after 19 years of management cover cropping (oats, fava beans and vetch) combined with mineral fertilizer application increased C stocks above 30 cm by ~3.5%, but decreased C over the entire 2 m profile by 7% (Tautges et al., 2019). The same mix of cover crops combined with compost both increased C stocks above 30 cm by 5%, and increased C over the whole 2 m profile by 12.6%. At the same time, these systems had similar tomato and corn yields (Scow et. al 2012). Other studies have also demonstrated that surface and subsoil SOC respond differently to agricultural management practices that are primarily concentrated at the soil's surface (Chenu et al., 2019; Syswerda et al., 2011). Estimates of whole-profile C sequestration based solely on data from surface soils can lead to inaccurate estimates of C storage potential in agricultural systems (VandenBygaart et al., 2011).

The goal of this study was to explore some of the potential mechanisms behind the observed differences in carbon storage in different RR management systems, and to see how these carbon stores have changed after an additional 7 years. In particular, we focus on the role of cover crops in promoting hydraulic conductivity, and how those hydraulic changes impact water, C chemistry, nutrient distribution, microbial biomass and community composition in the subsoil under the addition of additional C (compost) and N (nitrogen fertilizer). We hypothesized that the combination of cover crops and additional C input would result in large amounts of soluble C and nutrients being transported deeper via hydraulic transport and the cascade process,

leading to more microbially processed carbon and increased carbon stocks in the subsoil. We also hypothesized that these differences are not due to seasonal variation, and that increased soluble C and nutrient stocks will be consistent at multiple timepoints throughout the year.

## 2 Methods

### 2.1 Field Site and Historical Management

The experiment was conducted at the Century Experiment at the Russell Ranch Sustainable Agricultural Facility in Davis, CA, in the southern region of the Sacramento Valley at an elevation of 16 m. A detailed description of management history at the Century Experiment is provided in Tautges and Chiartas et al (2019) and is described here only briefly. Davis experiences hot summers and cool winters, with a 2018-2019 average temperature of 16°C from November to March when cover cropping occurs, and 29°C during the normal vegetable production period of April to September. Average annual rainfall for the 2018-2019 year was 812 mm, most of which fell between December - April in keeping with the xeric moisture regime in this area. (Supplementary Figure A5) (http://atm.ucdavis.edu/weather/uc-davis-weather-climate-station/).

The site has two soil types: (a) Yolo silt loam (Fine-silty, mixed, superactive, nonacid, thermic Mollic Xerofluvent) and (b) Rincon silty clay loam (fine, smectitic, thermic Mollic Haploxeralf). Detailed soil horizon information (classification, texture and depths) can be found in Supplementary Table A3 and the Century Experiment published dataset in Wolf et al. (2018). Abbreviations used in this paper (CONV, CONV+WCC, ORG) correspond to the abbreviations used in Wolf et. al 2018 (CMT, LMT, OMT), and are identical to those used in Tautges & Chiartas et. al 2019 for ease of comparison.

The experimental design is a randomized complete block design (RCBD) with three blocks and nine systems. Two blocks are placed on the Rincon silty clay loam, and the third block is on the Yolo silt loam. Experimental plots were 64 m x 64 m (0.4 ha). Only three systems of the nine described in Tautges and Chiartas et al. (2019) were measured in the current paper: CONV (mineral fertilizer), CONV+WCC (mineral fertilizer + cover cropped) and ORG (composted poultry manure + cover cropped). All plots are in a two-year maize-tomato rotation, with three replicate plots of each crop in any given year. Each treatment sampled in this manuscript consisted of 3 plots under tomato and 3 plots under corn, to give a total of 9 corn plots and 9 tomato plots in total. All plots were irrigated with subsurface drip at the time of sampling, having converted from furrow irrigation to subsurface drip in 2014. All plots also received 4 tillage passes to a depth of 20.5cm, and ORG and CONV+WCC plots received additional tillage passes to 6.5cm to incorporate cover crop and compost residue (Supplementary Table A5). While the lack of a compost-only treatment at Russell Ranch precludes conclusions about the impact of compost application alone, comparing the CONV+WCC treatment to the ORG treatment allows us to highlight how adding compost to a cover cropped plot impacts surface and subsoil C stocks, and provides insight into why these impacts occur.

### 2.2 Historic Carbon, Nutrient and Bulk Density Values

Historical cover crop shoot, compost, and crop residue inputs were calculated based on the Century Experiment published dataset in Wolf et al. (2018). Total C and N of composted manure, aboveground cover crop biomass, and crop residues were determined on a CS 4010 Costech Elemental Analyzer (Costech Analytical Technologies). Total aboveground C and N incorporated was calculated by multiplying percent C and N of residues by total harvest biomass. Due to compost nutrient analysis not being performed every year, estimates from 1993-2000 used %C, N, P and S values averaged for that 7-year period, while estimates from 2000-2018 used %C, N, P and S values averaged for that 18-year period. Total aboveground C, N, P and S inputs were calculated by summing above ground crop residue, WCC, mineral fertilizer and compost inputs per plot per year. Calculated N inputs represent the total N content of the aboveground added WCC and crop residue biomass, and do not differentiate between fixed N and N uptake from the soil in the case of cover crop legumes. Measurements do not include estimates of belowground biomass due to a lack of data.

Soil % C and nitrogen (N) values for 0-15, 15-30, 30-60 and 60-100 cm in 1993 and 2012 were taken from Tautges & Chiartas et. al (2019), while values for the same depths in 2003 were taken from the Century Experiment published dataset in Wolf et. al (2018). C and N analyses used in this paper were all performed using the same methods (Tautges and Chiartas et. al 2019, Wolf et, al 2018) on ball-milled, air dried samples in a CS 4010 Costech Elemental Analyzer (Costech Analytical

Technologies). Total C and N values for 15-60 cm in 1993 were calculated by performing a weighted average of C and N % values from 15-30 and 30-60 cm.

Bulk density values used in this paper were sampled using a Giddings hydraulic probe to 2m in 1993, 2007 and 2012 (2007 values taken from Wolf et. al 2018 and 1993, 2012 values taken from Tautges et. al 2019). In 1993, bulk density was collected in 0–25, 25–50, 50–100, and 100–200 cm depth layers with an 8.25 cm diameter probe. In 2007 and 2012, bulk density was collected in 0–15, 15–30, 30–60, and 60–100 cm depth layers, with a 4.7 cm diameter probe. In 1993, 2007 and 2012, cores were collected from four random locations within each plot. Bulk densities were determined using mass of oven-dried soil (105°C, 24 hr.) and total volume of the core averaged for each depth increment (Blake and Hartge, 1986). Bulk density depths from 1993, 2007 and 2012 were adjusted to 2018 depths through the calculation of weighted averages using adjacent depth layers for comparison. Historical C stocks from 0-100 cm for 1993, 2003 and 2012 were calculated via depth weighted sum (Tautges and Chiartas et. al 2019) using bulk density values taken in 1993, 2007 and 2012 respectively. Depth-adjusted 2012 bulk density values were then used to calculate 2018 C and nutrient stocks due to the lack of more recent bulk density measurements for all plots. Bulk density values below 30 cm were assumed to have not undergone large changes between 2012-2018 (Tautges and Chiartas et. al 2019), while bulk density sampling from 0-20 cm in select Century Experiment plots indicated a limited difference in bulk density (less than 3%) from 2012-2019 (Wang, unpublished data).

## 2.3 Field Operations

Cover crop planting and incorporation in ORG and CONV+WCC systems in 2017-2018 followed the trend of previous years, being planted onto 15 cm raised beds 1.5 m apart with a mixture of oat (*Avena sativa* L., 42.0 %C, 2.5 %N), faba bean (*Vicia faba* L., 44.1 %C, 3.5 %N) and hairy vetch (*Vicia villosa* Roth, 44.5 %C, 5.2 %N), and terminated by mowing plus 2-3 disking passes in March. Cover crop biomass was sampled by cutting aboveground biomass from one 4.5 m$^2$ area in each plot prior to termination. Corn and tomato biomass residues were measured by cutting aboveground biomass at two 1.5 m$^2$ locations per plot after harvest. Biomass samples were oven dried at 65 °C for 4 days and ground to 2 mm prior to total C and N analysis. Fertilization during the 2017-2018 growing season was also similar to previous years, with CONV and CONV+WCC plots receiving 325 kg/ha 8-24-6 (26 kg N/ha, 78 Kg P/ha, 19.5 kg K/ha) starter fertilizer at the time of planting. Tomato CONV plots also received ammonium sulfate at a total rate of 200 kg N/ha, while maize CONV plots received ammonium sulfate at a total rate of 235 kg N/ha.

From 1993-2018, ORG plots normally received a spring application (February 2018) of composted poultry manure at a rate of 3.6 Mg/ha (24.9 % C, 3.5 % N, 1.6 %P, 1.47 %S). However, during the 2018 season, these plots switched from spring to fall compost application, resulting in an additional application of 3.6 Mg/ha compost in September 2018.

## 2.4 Soil Sampling

Soil sample collection took place in the 2018-2019 growing season. Plots were sampled at 4 timepoints: February 2018 (Pre-CC Incorporation), June 2018 (Mid-Season), September/October 2018 (Post-Harvest), and February 2019 (Pre-CC Incorporation). A substantial amount of variation in both extractable organic carbon (EOC) and mineral N measurements can occur during the growing season (Li et. al 2018). Our sampling regime at multiple timepoints was meant to account for that variation in both winter and summer months to give a more accurate snapshot of C and nutrient availability during the growing season. All sampling took place in the raised beds between furrows. Samples in February 2018, September/October 2018 and February 2019 were taken using a tractor-mounted Giddings probe with a diameter of 3 cm from all replicate plots of each system (n = 6 plots per treatment). Samples taken in June 2018 were taken using an auger to 100 cm and were only taken in the experimental plots planted with tomato (n = 3 plots per treatment). Three replicate cores were taken per plot, sectioned into 0-15, 15-60 and 60-100 cm depths, composited, and then subsampled. Aliquots of each soil were frozen at -20 °C for PLFA analysis within 48 hours of sampling, while the remaining samples were sieved to 8 mm and stored at 4 °C until analyzed.

## 2.5 Carbon, Nutrient and Aggregation Analysis

All analyses described below were carried out on samples taken during the 2018-2019 growing season. Extractable organic carbon was determined using a 0.5 M potassium sulfate extraction within 48 hours of sampling. For each sample, 6 g of soil

were extracted with 0.5 M $K_2SO_4$ in a 1:5 ratio, shaken for one hour, filtered through Q5 filter paper and analyzed within 48 hours on a Shimadzu TOC-L Total Organic Carbon analyzer according to Jones and Willett (2006). Due to the moisture limited conditions present during summer at our study site, we chose an EOC extraction method as opposed to DOC sampling via tension lysimeters in order to compare soluble C measurements at different timepoints and soil water contents. Measurements of EOC are commonly used to estimate soluble C (Slessarev et. al 2020, Matlou et. al 2007) and may be more sensitive to

recent C and litter inputs, making them more suitable for answering questions on the impacts of C input, N amendment and tillage (Li et. al 2018).

Aliquots of the $K_2SO_4$ extract were immediately frozen at -20°C and later analyzed for nitrate by reacting with vanadium(III) chloride according to Doane and Horwath (2003); and ammonium via the Berthelot reaction as laid out in Rhine et al. (1998). Available calcium, phosphorus and sulfur were measured on 2 mm sieved air-dried samples using the Mehlich-3 soil test

(Mehlich, 1984). Total soil C and N values were measured on a CS 4010 Costech Elemental Analyzer (Costech Analytical Technologies) using air-dried, ball milled samples. The 2018 C and nutrient stocks were calculated using depth-weighted sums (Tautges et. al 2019) with bulk density values from 2012.

Aggregation measurements were carried out using the method outlined in Wang et al. (2017), adapted from the wet-sieving method outlined in Elliott (1986). Soils were gently passed through an 8mm sieve, and a 50g representative sample was

submerged in room temperature water on top of a 2 mm sieve. This sieve was moved up and down for 2 min (50 submersions per minute) using an audio metronome to keep track of the number of submersions. The soil and water passed through the 2mm sieve were gently transferred by rinsing onto a 250 μm sieve and submerged again. The process was repeated using a 53 μm sieve to generate 4 aggregate size fractions (8 mm-2 mm, 2 mm-250 μm, 250 μm-50 μm, >50 μm) which were rinsed into pre-weighed aluminum pans, oven-dried at 60 °C, and weighed. Mean weight diameter of the aggregate fractions was

calculated as the weighted average of the four aggregate size fractions (van Bavel, 1950).

## 2.6 Phospholipid Fatty Acid (PLFA) Analysis

PLFA analysis was carried out on 2018 samples using the high-throughput method outlined in Buyer and Sasser (2012). Briefly, freeze-dried aliquots were extracted using Bligh-Dyer extractant. Phospholipid fractions were separated from the neutral lipid and glycolipid fractions using solid phase extraction columns. Phospholipids were then dried under N2 gas,

transesterified, and methylated. After methylation, the samples were dried again with $N_2$ gas and redissolved in hexane containing a known concentration of an internal standard (19:0) (Microbial ID, Newark, DE, USA). PLFAs were identified using the Sherlock software from Microbial Identification Systems and quantified using a gas chromatograph equipped with a flame ionization detector. A total of 56 different PLFAs were identified. PLFAs were assigned to Gram-positive, Gram negative, Cyclopropyl precursors, Saturated and Monounsaturated groups as outlined in Bossio and Scow (1998)

(Supplementary Table A1).

## 2.7 Hydraulic Conductivity and Moisture Content

Three 20 $cm^3$ cores were collected in September 2018 for saturated hydraulic conductivity from each plot that had been under tomato in 2017-2018 (a total of 9 plots). Cores were taken from a depth of 35 cm. Unfortunately, two cores were damaged during measurement, giving a total of 25 cores measured from the three treatments. Care was taken to transport the cores in

foam holders to avoid creating compaction or preferential flow paths in transit. Cores were stored at 5 °C until measurement. A KSAT device was used to measure the cores with a falling head technique per the manufacturers manual and conductivity data was normalized to 20 °C using the Ksat software from the manufacturer (Meter Group, Pullman, Washington USA).

Soil moisture content was measured with a multi-depth profile capacitance probe in carbon fiber access tubes that were

installed according to the manufacturer's recommendations with great care taken to avoid air gaps along the tube (PR 2/6, Delta-T Devices, Cambridge, UK). The factory calibration of the profile probe was used with an accuracy of ± 0.04 $m^3$ $m^{-3}$. Volumetric soil moisture was measured at six depths (10, 20, 30, 40, 60, 100cm) (PR 2/6, Delta-T Devices, Cambridge, UK). Access tubes were installed in the field with a custom auger taking care to make the holes smooth and straight according to the manufacturer's recommendations. A total of 27 tubes were installed, with 3 tubes per subplot for a total of n = 9 per

treatment (ORG, CONV+WCC, CONV). The measurements were made on 8 dates between January 12 - March 1, 2019. Data

was processed using R (R Core Team 2014), and soil moisture depth from 10-100 cm was calculated using trapezoidal integration.

## 2.8 Fourier transform infrared Spectroscopy

Fourier transform infrared (FTIR) spectra of soil samples from 1993 and 2018 were collected using diffuse reflectance infrared Fourier transform spectroscopy (DRIFT; PIKE Technologies EasiDiff) with soil (air dried) diluted to 10% with KBr (Deiss et al., 2020). Spectra from 1993 samples were collected from air-dried, homogenized, archived soils from the Century Experiment Archive, while 2018 spectra were collected from air-dried, homogenized samples taken in 2018. 1993 spectra from 15-30 cm and 30-60 cm were combined into a single 15-60 cm spectra via weighted average for comparison with 2018 samples. The variation between these averaged 15-30 and 30-60 cm soils was found to be negligible for all three systems (Figure A6). All DRIFT spectra were collected using a Thermo Nicolet 6700 FTIR spectrometer (Thermo Scientific) using 256 scans, 4 cm$^{-1}$ resolution, and a DTGS detector. Three replicate samples were used, and average spectra were created for analysis. Peak intensity ratios of aromatic to carboxyl moieties [$v$(C=C):vas(COO$^-$) (1662 cm$^{-1}$:1631cm$^{-1}$)] were calculated using peak areas.

While FTIR is not a strictly quantitative tool for identifying specific compounds in mixed samples, it can be used pseudo-quantitatively due to the fact that the absorption of IR light by a specific molecular bond at a specific electromagnetic frequency follows the Beer-Lambert Law (Beer's Law) (e.g., Margenot et al 2016, Smith 2001). Therefore, the height and area of a spectral peak are proportional to the abundance of molecules in a sample (linear relationship), and comparing the presence and absence of peaks and the relative differences in spectral contributions from each peak in a subtraction can suggest differences in C chemistry. However, it is important to note that spectral reflectance can lead to some non-linearity in concentration and absorbance, and thus pseudo quantification. Previous studies with DRIFTS in both the near-infrared (Dalal and Henry, 1986) and mid-infrared regions (Demyan et al., 2012; Margenot et al., 2015; West et al., 2020; Deiss et al., 2021) have shown direct associations between soil organic C concentration and absorbance at specific frequencies (depicted as peak height or area of single peaks or peak ratios). Spectral subtractions were performed using Omnic 9.8.286 (Thermo Fisher Scientific) and corrected for non-linearity of concentration and absorbance by using the Kubelka-Munk (KM) function. Plots of FTIR spectra were made using Origin 2018b (OriginLab Corporation). Subtractions were performed in two ways: 1) mean spectra, for each treatment and depth, of the 1993 spectra were separately subtracted from the corresponding 2018 spectra to reveal C chemistry changes over this period; and 2) the 2018 mean spectra, for each depth, were subtracted (ORG-CONV, ORG-CONV+WCC, CONV+WCC-CONV) to show the difference in C chemistry by treatment.

## 2.9 Statistical Analysis

All data analysis and graph production were done using R v. 4.0.2, (R Core Team, 2020) using the tidyverse package (Wickham et al., 2019). Analysis of variance (ANOVA) was conducted using a linear model to determine the effects of management system, depth, and time point. We first checked for normality and assumptions of the linear model prior to ANOVA, then fit a mixed effect model with "block" as a random effect. Since "block" was not significant for any of the variables measured, we removed it from the model. Statistical differences between management systems were analyzed separately for each depth using paired t-tests with Bonferroni correction for multiple tests at 5% significance level. Data and code used for this paper are archived at https://zenodo.org/badge/latestdoi/181972884.

## 3 Results

### 3.1 Nutrient Inputs

The cumulative estimated aboveground C input over 25 years was 186 Mg ha$^{-1}$, 123 Mg ha$^{-1}$ and 113 Mg ha$^{-1}$ for ORG, CONV+WCC and CONV systems respectively. Averaged per year over 25 years, C inputs to each system were 7.44 Mg ha$^{-1}$, 4.92 Mg ha$^{-1}$ and 4.52 Mg ha$^{-1}$ for ORG, CONV+WCC and CONV systems respectively (Supplementary Table A4). Due to the combination of compost and cover crop residue and root inputs, ORG systems received approximately 1.5x more C than CONV+WCC. Although CONV+WCC produced similar amounts of tomato residue and less maize residue than CONV, the presence of cover crops meant that CONV+WCC systems received 1.1x more C than CONV systems.

Due to combined N inputs from cover crop and compost, ORG systems received 1.4x as much external N inputs (7.5 Mg ha$^{-1}$) as CONV+WCC systems (5.4 Mg ha-1), and 1.65x as much N as CONV systems (4.5 Mg ha$^{-1}$). External N inputs to CONV+WCC systems were close to 1 Mg ha$^{-1}$ higher than CONV systems over 25 years, with ~40% of the external N inputs to CONV+WCC systems coming from the decomposition of cover crop residue, and the other ~60% from mineral fertilizer application, compared to 100% of total N inputs in the CONV coming from mineral fertilizer application. ORG systems received over 3x as much phosphorus via compost (3.23 Mg ha$^{-1}$) as CONV+WCC (1.09 Mg ha$^{-1}$) and CONV (0.99 Mg ha$^{-1}$) did from P fertilizer. ORG systems also received 1.15 Mg ha$^{-1}$ of sulfur from compost, approximately 0.5x as much as CONV+WCC (2.19 Mg ha$^{-1}$) or CONV (1.98 Mg ha$^{-1}$) systems received.

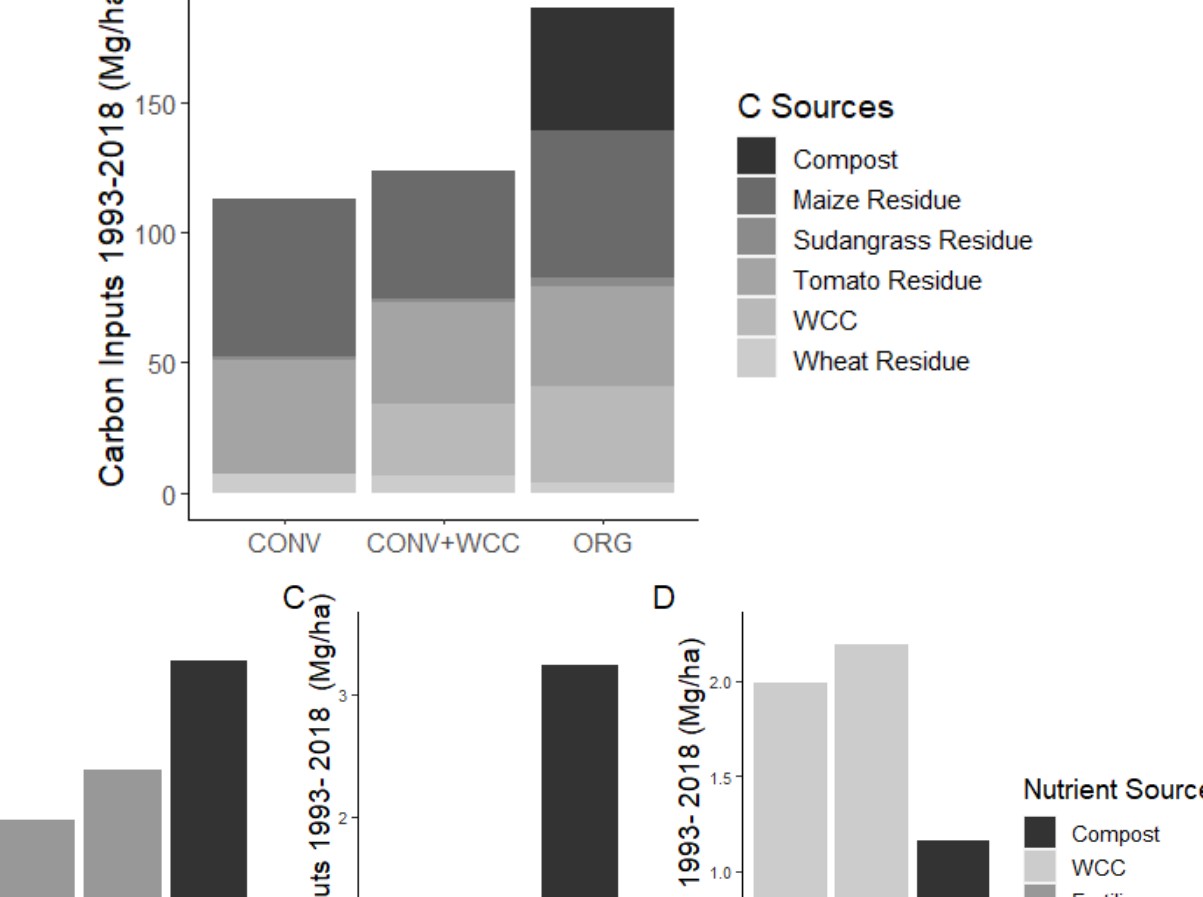

315

**Figure 1a-d. Total aboveground C, N, phosphorus, and sulfur added per plot to ORG, CONV+WCC and CONV systems between 1993-2018. All values are given on a mass basis (Megagrams/hectare).**

## 3.2 Soil Carbon Content Changes over 25 years

Carbon stocks in the 1 m profile of ORG systems showed an increase of ~19 megagrams/hectare from 1993-2018 (p=0.06) (Figure 2). Most of this C gain was concentrated in the 0-15 cm (~5 Mg ha[-1,] p<0.01) and 15-60 cm depths (~10 Mg ha[-1], p=0.1). Due to the large amount of variation present in these observations and the limited number of replicates, it was difficult to spot strong trends in C stock changes, as shown in the lack of significant change in the bottom 60-100 cm (~3 Mg ha[-1], p=0.26). No significant changes in C stocks in the 1 m profile were noted in CONV or CONV+WCC systems from 1993-2018 (p=0.47, p=0.51). When depth intervals were considered separately, only CONV systems showed a decrease in C stocks (~ -3 Mg ha[-1]), at the 0-15 (p<0.01) depth (Figure 3). CONV+WCC systems did not show a clear trend of C decrease at any individual depth with the significance testing used.

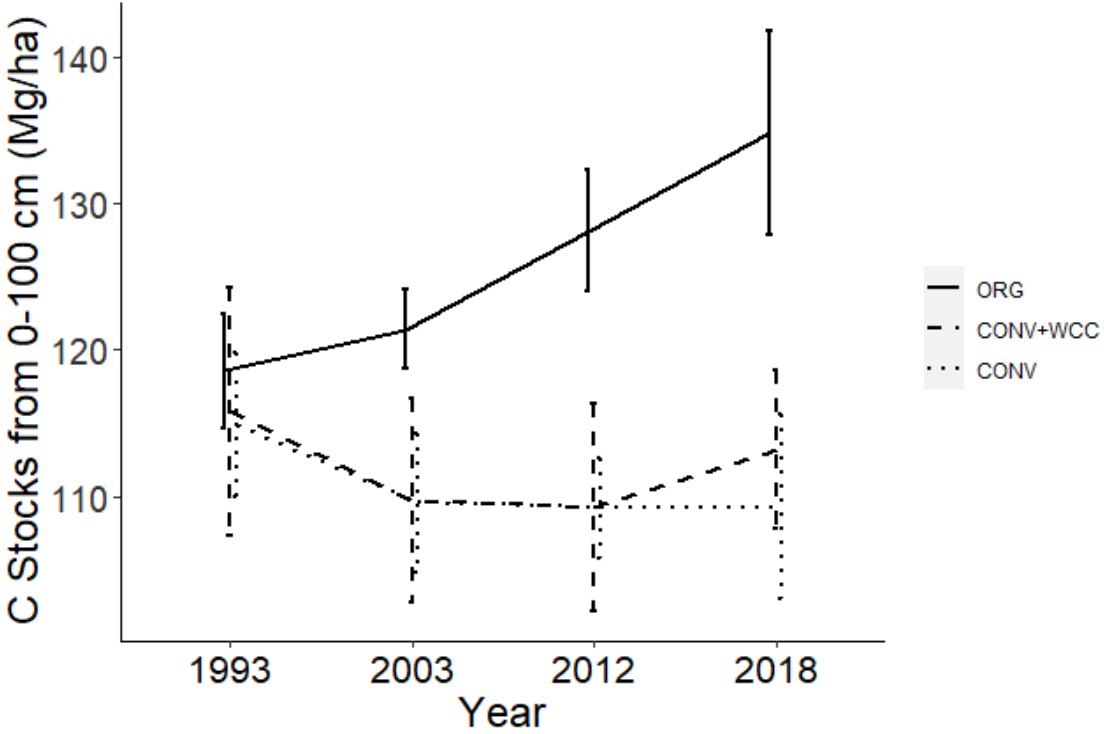

**Figure 2. Carbon stocks of the 1m profiles of ORG, CONV+WCC and CONV systems from 1993 to 2018. Carbon stocks are given in Mg ha[-1]. Error bars denote standard error. Please note that all systems transitioned from furrow to drip irrigation in 2014.**

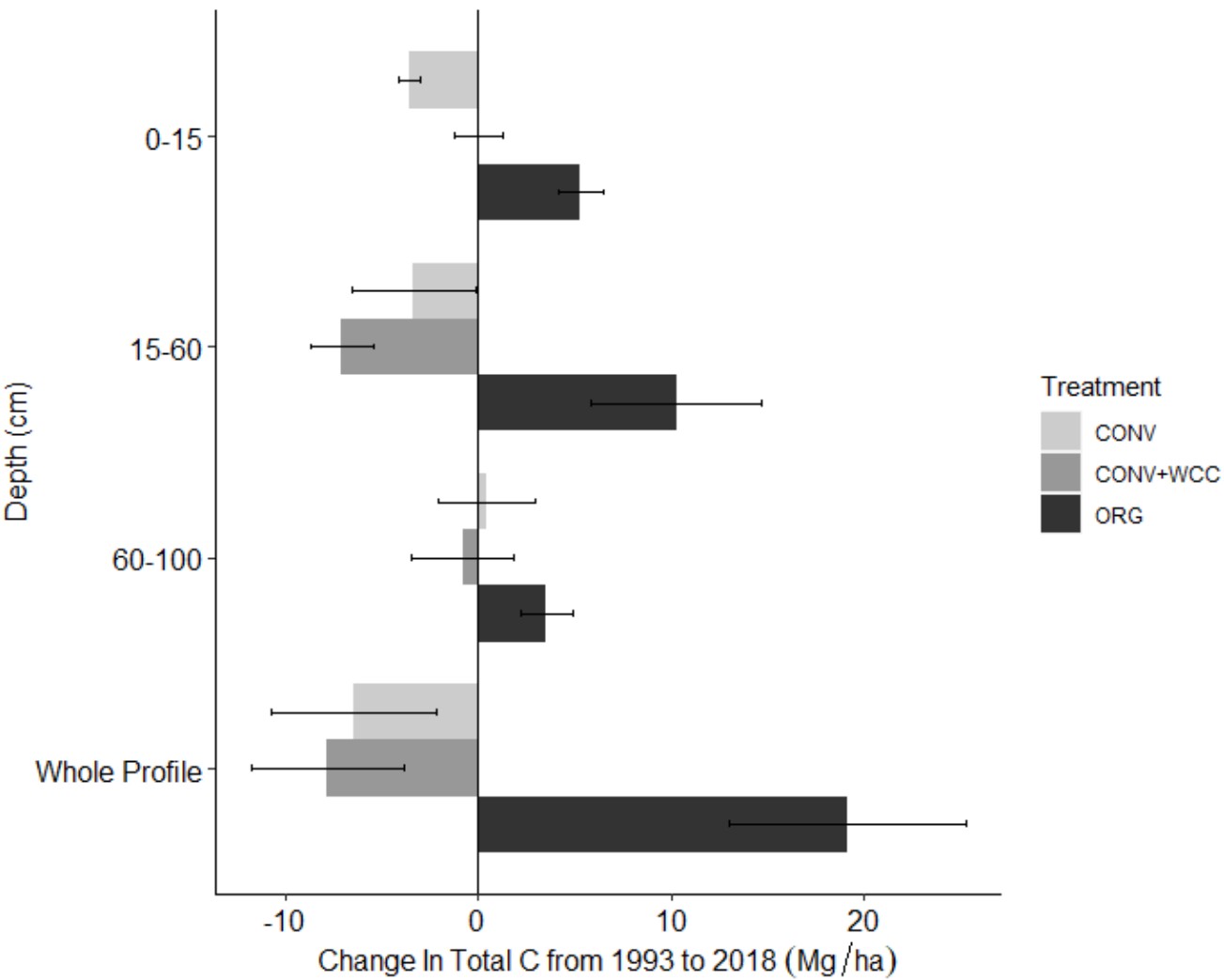

Figure 3. Change in C stocks of ORG, CONV+WCC and CONV systems from 1993-2018 by depth. Values were obtained by subtracting C stocks in 1993 from 2018 stocks for individual systems, and then averaging by management system. Error bars denote standard error. (* = significantly different from 0, p-value <0.05, + = significantly different from 0, p-value 0.05<x<0.1)

### 3.3 Moisture Content, Hydraulic Conductivity, and Aggregation

Cover cropped systems (ORG and CONV+WCC) stored approximately 10% more water than non cover cropped systems (CONV) in the upper 1m of the soil profile during the 2019 winter (Fig 4). There was no difference in moisture content between ORG and CONV+WCC systems. Averaged hydraulic conductivity measurements showed differences among all three systems, but treatments with cover crops (ORG and CONV+WCC) had values that spanned 3 orders of magnitude compared to treatments without cover crops (CONV) (Fig. 5). There was no significant difference in MWD of aggregates between all three systems at any depth (Supplementary Figure A2).

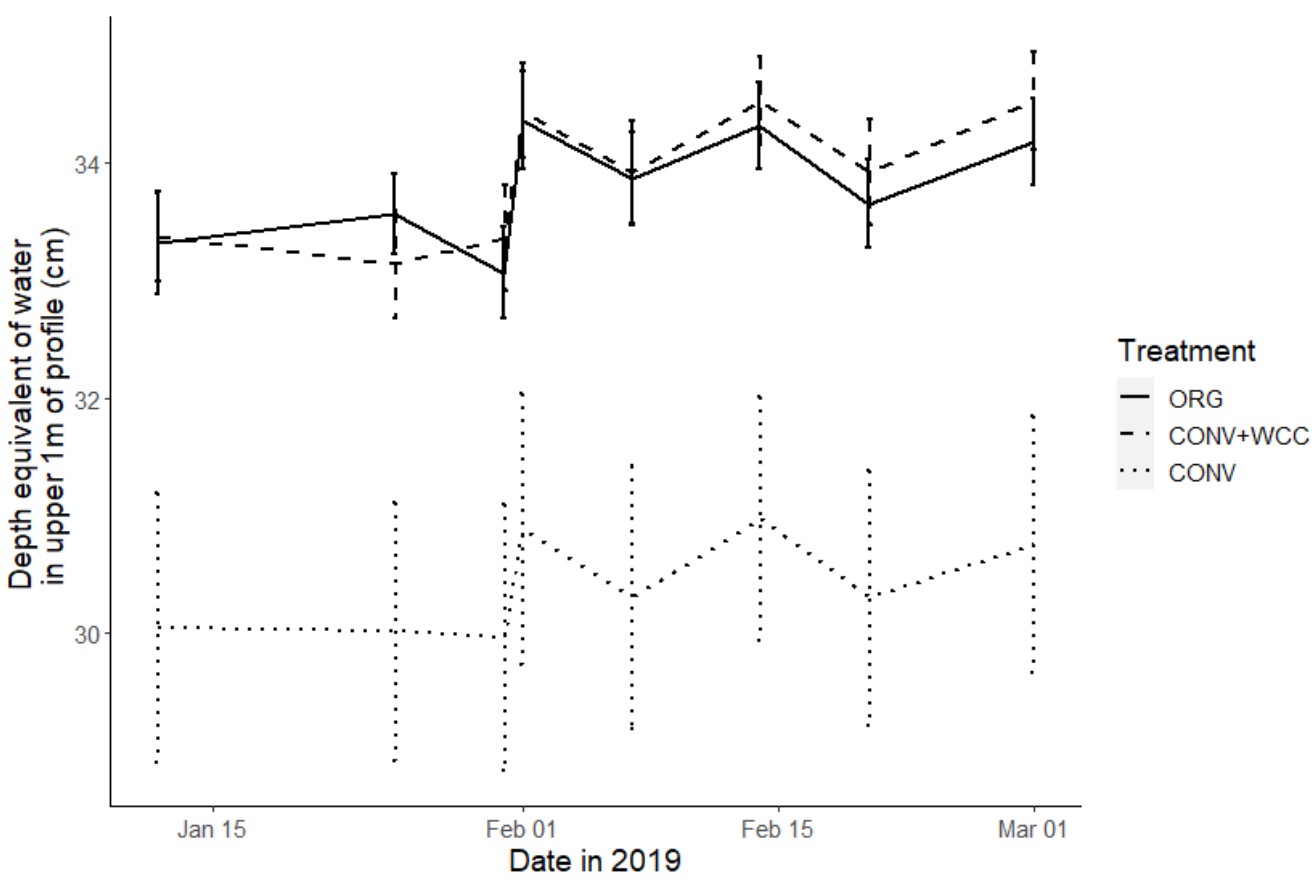

**Figure 4. Depth equivalent of water (in cm) in the upper 1m of ORG, CONV+WCC and CONV profiles during the Jan -Mar 2019 winter season. Error bars represent standard error.**


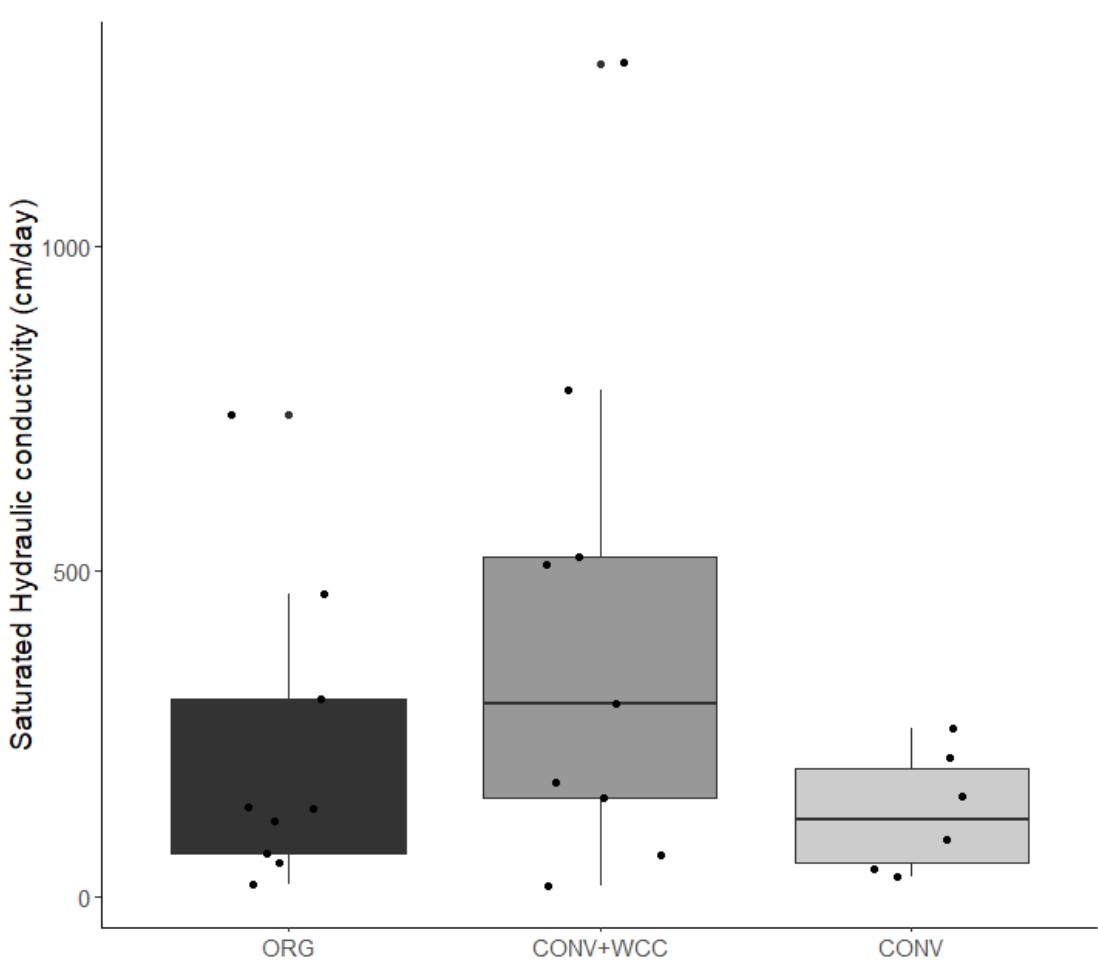

**Figure 5. Saturated hydraulic conductivity (cm/day) in ORG, CONV+WCC and CONV systems taken in August 2018.**

### 3.4 Soil Nutrient Content: Extractable Organic Carbon, Mineral Nitrogen, Phosphorus, Sulfur


Composted systems (ORG) had higher amounts of extractable organic carbon (EOC) (p<0.01), plant available phosphorus (p<0.01) and sulfur (p<0.01) in the 1m profile than non-composted systems (CONV+WCC and CONV) averaged across all dates the 2018-2019 year (Fig. 6). These differences were most pronounced in the upper 15 cm, where ORG systems had approximately 2x more EOC (p<0.01), 3x more phosphorus (p<0.01) and 1.75x more sulfur (p<0.01) than CONV+WCC or
CONV systems (Supplementary Figure 1).

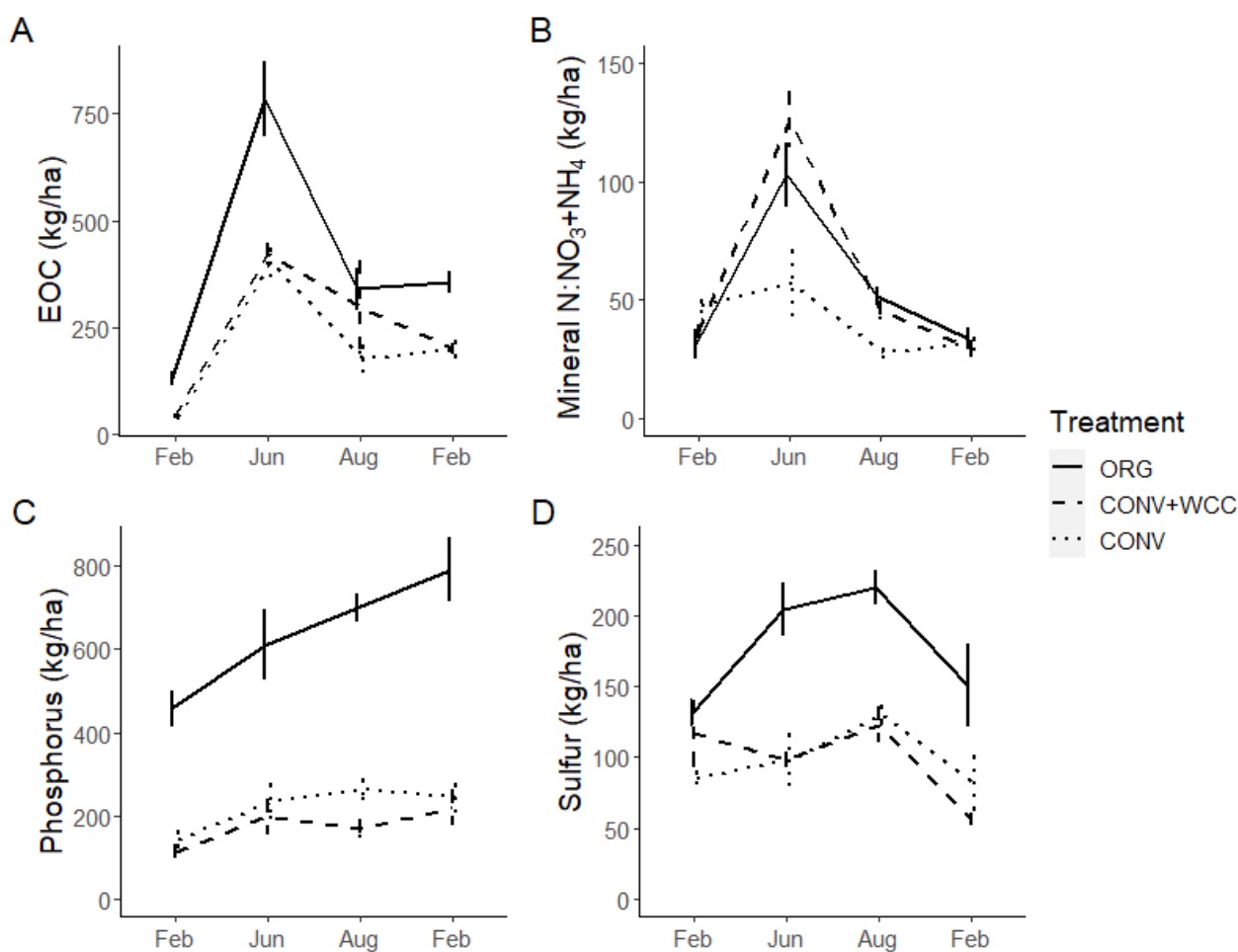

**Figure 6a-d. Extractable organic C, mineral N (NO₃⁻+ NH₄⁺), phosphorus and sulfur in 0-100cm profiles of ORG, CONV+WCC and CONV systems over the Feb 2018- Feb 2019 season. All values are given on a mass basis (kilograms/hectare). Error bars represent standard error.**


CONV+WCC systems had more mineral N (NO3+NH4) than CONV systems during the June and August timepoints (p=0.04), with up to 3.5x more mineral N than CONV systems mid-season, and 1.6x more mineral N at harvest. ORG systems trended towards higher mineral N during the April - September growing season but the magnitude of this difference was small (p=0.17).


Nutrient values showed large seasonal variation, with the highest levels of C and N observed during the June timepoint and highest sulfur levels at the August timepoint. EOC, mineral N, and sulfur values were lowest during the winter (Nov - Feb), which coincided with the period of highest rainfall. Phosphorus levels increased slightly throughout the 2018-2019 year.

Differences among systems and seasonal variation were also noted at a depth of 60 cm. ORG systems had more EOC (p<0.01), phosphorus (p<0.001), and sulfur at 60-100 cm than CONV+WCC or CONV systems. Mineral N values did not show significant differences between any of the three systems at 60-100 cm, though ORG and CONV+WCC systems trended higher during the growing season (Figure 7).


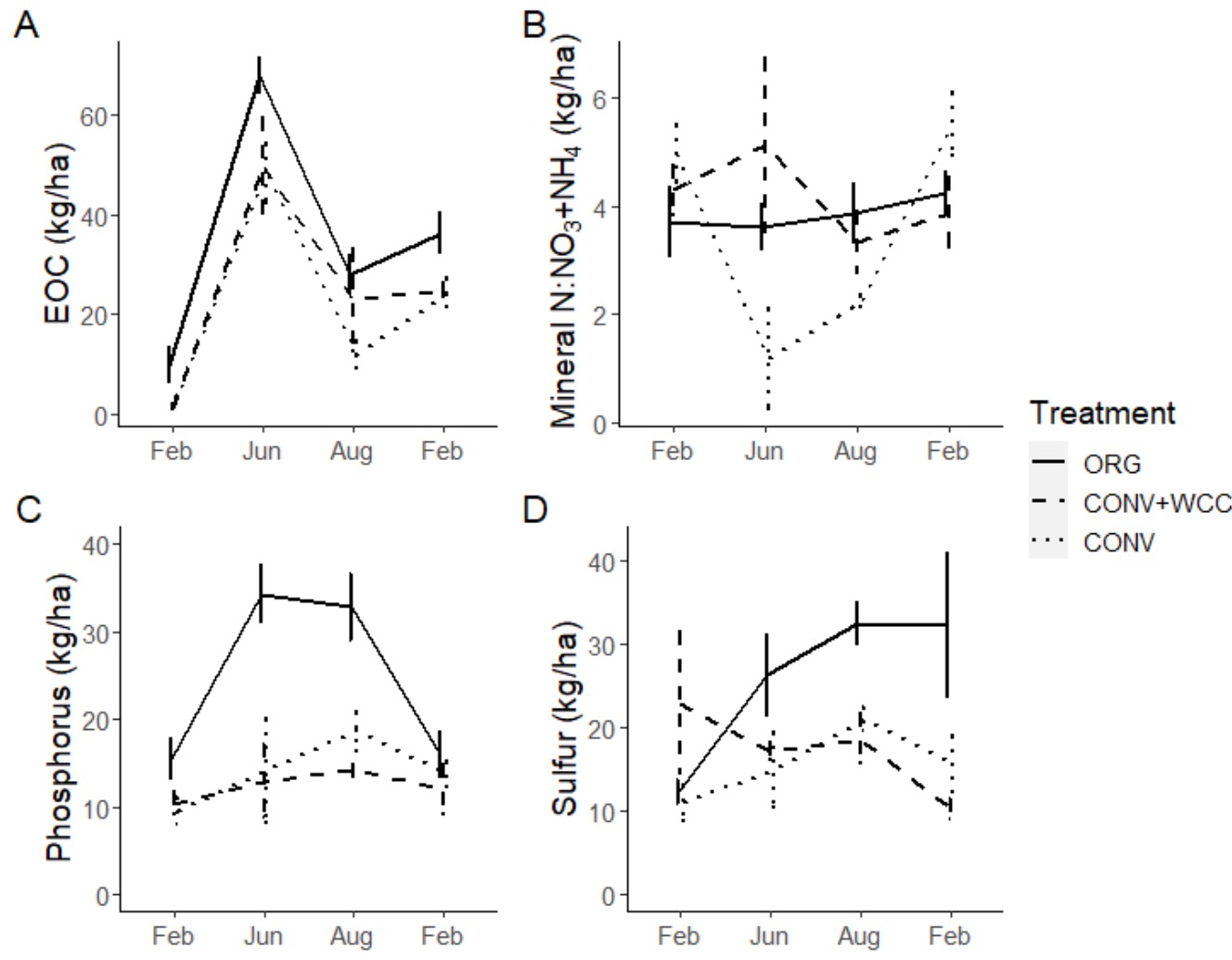


**Figure 7a-d. Extractable organic C, mineral N, phosphorus, and sulfur stocks at 60-100 cm in ORG, CONV+WCC and CONV systems over the Feb 2018- Feb 2019 season. All values are given in kg/ha. Error bars represent standard error.**

### 3.5 SOM Composition via FTIR

Spectral subtractions of 1993 from 2018 FTIR spectra revealed positive peaks (increased absorbance) from 1900 to 1200 cm$^{-1}$

$^{1}$ in all systems, indicating an increase in C functional groups within this region (e.g., aromatic, carboxyl) (Fig. 8A). FTIR band assignments are presented in Supplementary Table A2. All treatments showed positive peaks indicating an increase in carboxylate functional groups between 1993 and 2018, as denoted by bands at 1625 cm$^{-1}$ and 1400 cm$^{-1}$ (Fig 8a). However, ORG and CONV+WCC showed these distinct peaks at 15-60 and 60-100 cm depths, while CONV systems showed distinct peaks only at the 0-15 and 15-60 cm depths. CONV systems also showed a lower aromatic:carboxylate peak intensity ratio at

all depths than ORG and CONV+WCC systems from 1993-2018 (Table 1).

ORG and CONV+WCC systems showed distinct positive peaks associated with carboxylate functional groups at the 60-100 cm depths in 2018 when compared with CONV systems (bands at 1631 cm$^{-1}$), and slightly higher peaks associated with aromatic functional groups from 0-15 cm for CONV+WCC and 15-60 cm for ORG (bands at 1662 cm$^{-1}$) (Figure 8b). ORG also showed positive peaks associated with aromatic functional groups from 0-15 and 15-60 cm when compared to CONV+WCC (bands at 1662 and 1602 cm$^{-1}$) in 2018. Aromatic:carboxylate ratios provide an indication of the intensity of carboxyl peaks relative to aromatic peaks, which can be related back to concentrations of these functional groups in the sample. A lower aromatic:carboxyl ratio can indicate either more carboxyl or less aromatic functional groups, while a higher ratio can mean increased aromatic or decreased carboxyl groups. Aromatic:carboxylate peak intensity ratios decreased with depth for ORG and CONV systems when looking at changes from 1993-2018, but CONV+WCC ratios increased with depth (Table 1).

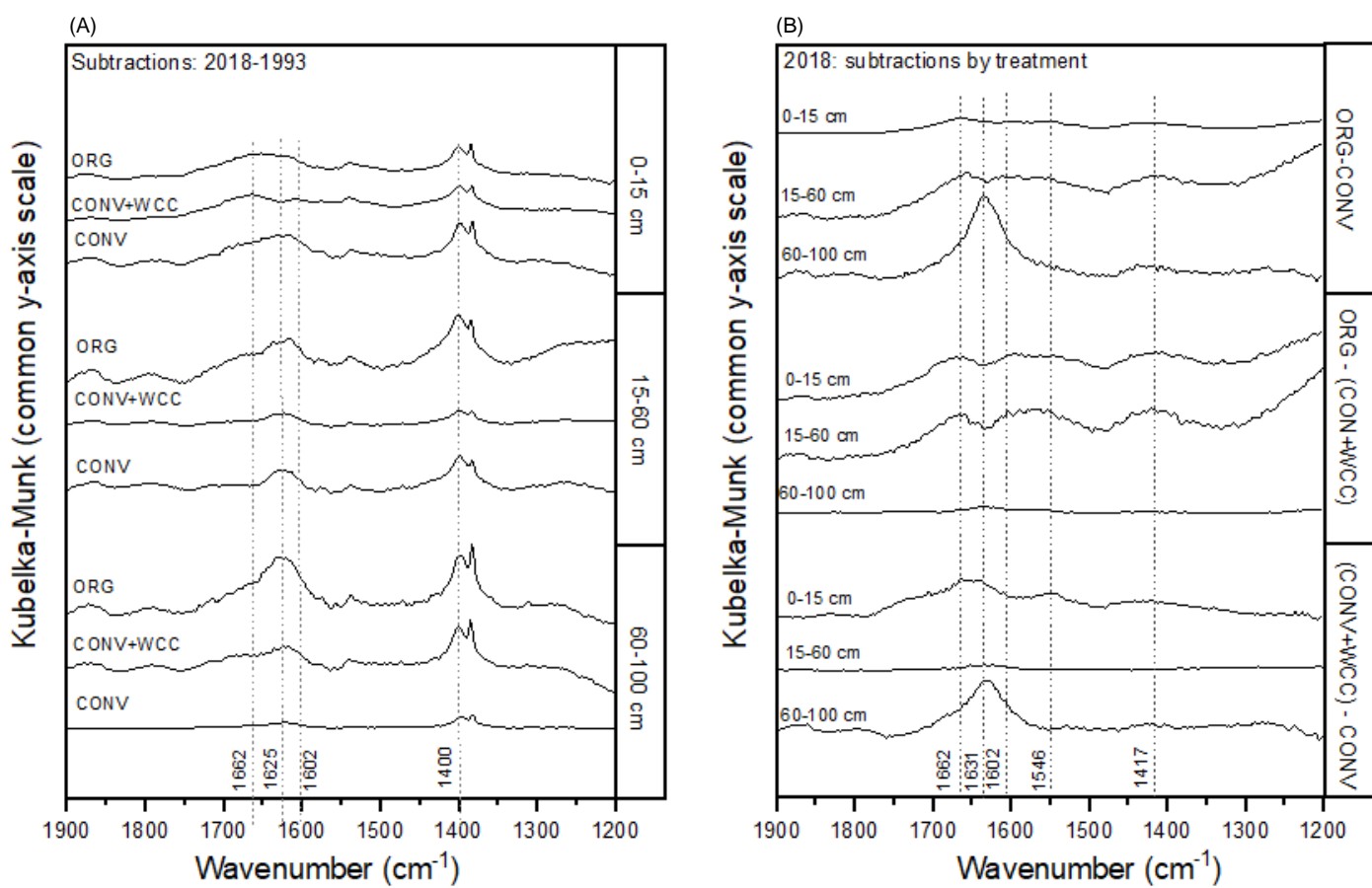

Figure 8a,b. DRIFT spectral subtractions for the 1900-1200 cm$^{-1}$ range comparing (A) 2018-1993 spectra for ORG, CONV+WCC, and CONV, and (B) ORG, CONV+WCC, and CONV spectra in 2018. Spectra are plotted with Kubelka-Munk units on a common y-axis scale, and are offset from one another for ease of comparison.

| | Depth | Peak Intensity Ratio ($1662\ cm^{-1}$ : $1631\ cm^{-1}$) | | |
|---|---|---|---|---|
| | | ORG | CONV+WCC | CONV |
| Subtraction: 2018-1993 | 0-15 cm | 1.38 | 1.22 | 0.45 |
| | 15-60 cm | 1.21 | 1.40 | 0.77 |
| | 60-100 cm | 1.17 | 2.50 | 0.014 |
| | | | | |
| | | ORG-CONV | ORG-(CONV+WCC) | (CONV+WCC)-CONV |
| Subtraction by treatment: 2018 | 0-15 cm | 1.39 | 1.18 | 0.46 |
| | 15-60 cm | 1.20 | 1.42 | 0.47 |
| | 60-100 cm | 0.38 | 0.64 | 0.45 |

Table 1. Peak Intensity Ratios for aromatic ($1662\ cm^{-1}$) to asymmetric carboxyl ($1631\ cm^{-1}$) groups in spectral subtractions.

**3.~~7~~6 Microbial Biomass and Stress Indicators - July 2018**

Microbial biomass decreased with depth in all systems. ORG and CONV+WCC systems had more microbial biomass at 0-15cm than CONV systems ($p=0.04$ & $p=0.04$ respectively), while ORG systems had more microbial biomass at the 15-60 cm depth than CONV+WCC or CONV systems ($p=0.03$ & $p=0.06$ respectively). Saturated: Unsaturated fatty acid ratio and Cyclopropyl 19: precursor ratio increased with depth, with CONV systems showing a weaker trend of higher Cy19: pre ($p=0.12$) and saturated: unsaturated fatty acid ratios ($p=0.07$) than ORG at 60-100 cm. Gram+: Gram- ratio also increased with depth, with CONV systems having a higher ratio than ORG or CONV+WCC systems at 60-100 cm ($p=0.01$ and $p=0.01$ respectively).

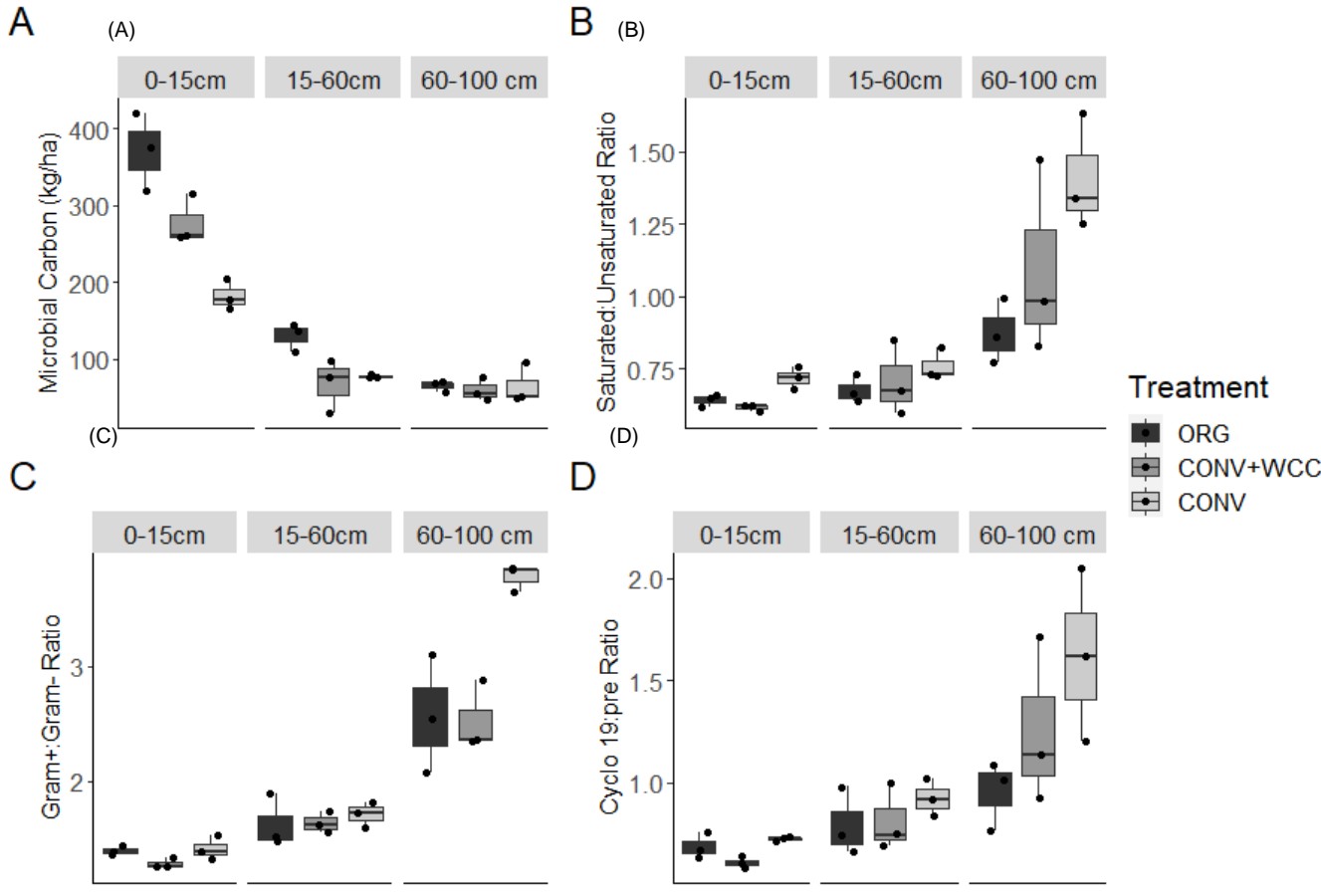

**Figure 9a-d. Microbial biomass and PLFA stress indicators measured in ORG, CONV+WCC and CONV systems during mid-season (July 2018). Ratios are unitless, while microbial biomass is given in kg/ha.**

## 4 Discussion

The ~19 Mg/ha increase in SOC over the 1m ORG profile after 25 years was attributed to a synergistic effect between cover crops and compost, which resulted in the movement of mobile C and nutrients deeper into the soil profile. We believe that high concentrations of mobile C and essential nutrients for microbial activity provided by the compost, combined with the easier movement of water downward associated with a history of cover-cropping, helped transport the material needed to build C in the subsoil.

### 4.1 Cover crop roots increase water storage and movement into subsoils

The higher moisture contents noted in CONV+WCC and ORG than CONV systems during the winter growing season are likely due to the presence of more water-filled spaces from cover crop roots (Figure 4). Cover crops increase both soil macroporosity and pore connectivity in fine textured soils, leading to an increase in both infiltration and hydraulic conductivity \over longer time scales (Scott et al., 1994, Haruna et al., 2018; Çerçioğlu et al., 2019; Gulick et al., 1994). A similar impact of cover crops have been noted in previous work done in Russell Ranch soils: a cover crop mix of purple vetch (*Vicia benghalensis L.*), common vetch (*Vicia sativa L.*) and oats (*Avena sativa L.*) increased soil moisture-holding capacity during

saturated conditions (Joyce et al. 2002); a cover crop of common vetch produced no changes in bulk density after 10 years (Colla et al. 2000) and a wheat (*Triticum aestivum L.*) cover crop increased infiltration by 43% and decreased DOC export by 54% in a furrow irrigated system, causing the soil profile to become a DOC sink (Mailapalli et al., 2012). Proposed mechanisms are that cover crops increase infiltration and hydraulic conductivity by increasing  soil structure through aggregate formation, reduced soil crusting, and reduced  soil compaction due to increased organic matter content and formation of root channels (Chen and Weil, 2010; Franzluebbers, 2002). Other potential, albeit less likely, explanations for increased moisture content could include lateral subsurface flow (unlikely due to the <1% slope of this field) and differences in runoff and runon (also unlikely due to low slope). Given that our cover crop mix is known to have extensive root networks that extend deeper than the 30cm plow layer (Fan et al., 2016), the observed differences in moisture content were attributed to increased biopores created by roots (Hangen et al., 2002).

Despite the presence of cover crops, we found no significant difference in aggregate MWD among systems (Supplementary Figure A2). This may be because root-induced soil alterations, such as aggregation, are highly localized and dependent on the root architecture of the cover crops. Specifically, cover crops with prominent tap roots, such as fava bean, are effective at creating continuous bio-pores, while fibrous roots such as in oats and hairy vetch, are particularly effective at promoting soil aggregate formation (Ogilvie et al., 2021). Therefore, the mixture of cover crops planted at the site likely resulted in a large amount of variation in  aggregation and pore connectivity and may have resulted in the non-significant aggregate MWD values.

While mean hydraulic conductivity values were also not significantly different between treatment systems, hydraulic conductivities were more variable in the two systems with than without cover crops (Figure 5).  Roots may increase macroporosity by opening up channels as they decay (Ghestem et al., 2011), and increased water movement through these macropores can result in hydraulic conductivity values that can range over three orders of magnitude (Øygarden et al., 1997) similar to what we observed. In addition, the sample size used for $K_{sat}$ measurements (cross-sectional area of 250 cm$^2$) may be too small to capture the effects of cover crop roots, whose impacts are likely to be detected at a larger scale (Ozelim and Cavalcante, 2017). It is well recognized that hydraulic conductivity measurements can vary widely across fields and landscapes (Rahmati et al. 2018) and often do not reflect the presence of macropores (Brooks et al. 2004).  Though our measurements do not reveal statistically significant differences between treatments, the scattered high-permeability zones in the cover-crop treatments likely play a role in rapid moisture redistribution and may explain the elevated deep moisture contents in ORG and CONV+WCC plots compared to CONV plots. We therefore attributed the more variable hydraulic conductivity and increased moisture content in ORG and CONV+WCC than CONV systems to the deeper, more abundant root-derived macropores from cover crops.

**4.2 Compost + Cover Crops increased the amount of EOC and carboxylate functional groups in subsoils**

Increased EOC levels (Figure 6a) in ORG plots and relatively more oxidized carboxylate C in the bottom 60-100 cm of ORG and CONV+WCC plots relative to CONV plots (Figure 8b) point to an accelerated cascade process (Kaiser and Kalbitz 2012) in these systems relative to CONV systems. The presence of elevated EOC levels in ORG plots are due to the higher amounts of soluble C found in compost (Wright et al., 2008; Zmora-Nahum et al., 2005), the increase in dissolved and water extractable organic C expected from cover crop residue (Singh et. al 2021 - cereal rye and hairy vetch), and the larger amounts of C added to ORG plots (Figure 1a). Compost also contains a large proportion of aromatic functional groups derived from lignin and other biomolecules (Leifeld et al., 2002) which tend to be rapidly removed from the soil solution at the surface (Leinemann et al., 2018). This preferential removal at the soil surface may be a function of the relatively low solubilities of non-polar aromatic functional groups (Maxin and Kogel-Knabner, 1995), as well as their tendency to partition into other non-polar, insoluble organic matter (Pignatello, 1999). When these aromatic functional groups are eventually oxidized by microbes, the higher solubility of carboxylate and other O-rich functional groups may allow for greater C transport. Carboxylate functional groups' ability  to form mineral-associated organic matter through association with charged surfaces or cation bridging (Aquino et al., 2011) would also promote MAOM formation and increase C storage times (Cotrufo et al., 2013; Leinemann et al., 2018) though our results do not provide sufficient support to determine the stability of the increased C stocks at 60-100 cm. As the C cascade is triggered by fresh C inputs which are preferentially sorbed within the top 30 cm (Liebmann et. al 2020), the

regular application of soluble C-rich compost and WCC residue, combined with increased hydraulic conductivity due to WCC roots can accelerate the process leading to greater subsoil C transport.

Cover crops are associated with elevated EOC levels and more aromatic functional groups in topsoil SOC (Ding et. al 2005, Zhou et. al 2012). Application of cover crops in CONV+WCC did not increase EOC content deeper than 15 cm compared to CONV (Figure 7a) but did have an impact on carboxylate functional group presence at the 60-cm depth (Figure 8b), possibly
indicating that cover crop residues are associated with smaller inputs of soluble C that are not as easily detected as the larger soluble C inputs coming from the compost. Additionally, the trend of increasing aromatic:carboxylate ratio with depth in CONV+WCC systems (Table 1) indicates that CONV+WCC systems may be accumulating more aromatic C relative to carboxylate C in deeper soil layers. This accumulation of aromatic C in CONV+WCC subsoils may be due to cover crop root residue introducing lignin and cellulose directly into the subsoil, while the carboxylate-C obtained from the decomposition of
surface residues can be potentially mineralized before being transported deeper as DOC (Chantigny, 2003; White et al., 2020a).

### 4.3 Compost + Cover crops increased nutrient availability and decreased microbial stress in subsoils

The higher P and S values noted in ORG subsoils (Figure 7c,d) can be attributed both to the higher organic P and S inputs associated with compost (Preusch et al., 2002), as well as the increased mobility of these inputs. Differences in crop uptake also play a potential role, but the lack of significant treatment effects at RR on crop nitrogen use efficiency (Kong et. al 2009),
or P cycling (Maltais-Landry et. al 2014) in these plots make crop uptake less likely to be a significant factor in nutrient availability. Organic phosphorus is more mobile than mineral P (Laos et al., 2000; Sharpley and Moyer, 2000), and mineralization of organic S into more soluble sulfate could also facilitate its movement (Edwards, 1998). Though our results were not able to detect significant differences in mineral N in the measured subsoils, the higher amount of organic N added in compost was also likely mineralized into more soluble nitrate (Vinten et al., 1994) (Figure 7b). Although soil microbial
communities are primarily water and C limited (Soong et. al 2020), the addition of N, P and S in ratios similar to that found in soil organic matter may increase transformation of C inputs into SOM by up to 52% by promoting microbial anabolism (Coonan et. al 2020).

Greater C and nutrient inputs were associated with the lower Gram +: Gram − ratios observed in subsoil ORG soils (Fig 6b, d). Higher values for these ratios, such as those
observed in CONV plots, have been associated with nutrient and energy limitation (Bossio and Scow, 1998: Petersen and Klug, 1994). Increases in these ratios represent an overall shift away from the thinner, more permeable cell membranes associated with Gram − bacteria and monounsaturated fatty acids: towards more tightly packed, less permeable cell membranes associated with Gram+ bacteria and saturated fatty acids (Silhavy et. al
2010). An increase in the Gram+:Gram− ratio has been associated with a decrease in easily available water and C (Fanin et. al 2019, Fierer et. al 2003b, Bossio et. al 1998), while an increase in the saturated: unsaturated ratio and cy17:pre ratios has been associated with low water potentials (−1.3 to −0.9 MPa) and potential dehydration (Moore−Kucera et. al 2007). The stress indicator trends in our data support our observations of increased
soluble C and water content in ORG systems.

Adding compost and cover crop residue increases microbial biomass at the 0-15 and 15-60 cm but not 60-100 cm depth relative to CONV systems. This greater biomass increase in ORG than CONV plots was attributed to compost providing a favorable nutrient stoichiometry for biomass formation (Kirkby et al., 2011; Richardson et al., 2014). Increased microbial biomass in surface layers of soil is an important potential source of C and other nutrients to subsoil layers through cell lysis from predation
and wet-dry cycles (Bonkowski, 2004; Xiang et al., 2008). It is associated with increased C storage through microbial necromass formation (Buchmann and Schaumann, 2018; Jilling et al., 2020), and we hypothesize that transport of microbial

products downwards through the profile could have contributed to the SOC increase in the subsoil observed in ORG systems (Figure 2b).

## 4.4 Compost + Cover Crops increased profile C stocks after 25 years, but Cover Crops alone did not

We found evidence that the SOC increases under the ORG system after 25 years (Figure 2) were due to the increased mobility of compost-added C and nutrients combined with increased infiltration due to cover crop roots, as well as the larger amounts of C added to ORG plots. Larger SOC increases under yard waste compost + cover crops relative to cover crops alone have also been noted in other California long term experiments on a loamy sand soil (White et. al 2020a - rye, fava bean, pea, common vetch, purple vetch), indicating that C input from cover crops alone may not play a large role in increasing subsoil C.

These experiments also note the importance of belowground carbon inputs on SOC stocks, a factor that was not included in this analysis. While cover crop biomass does represent significant C and N input to surface soils, the channels their roots create for mobile nutrients (either organic or mineral) to move downwards may be as important as their C and N inputs to subsoil SOC dynamics.

We noted significant seasonal variation in EOC, mineral N, P and S levels throughout the 2018-2019 growing season, though
ORG plots consistently had higher EOC and P than CONV+WCC or CONV plots at all timepoints (Figures 6a,c) and higher S during the growing season. These soluble C and nutrient inputs peaked during the growing season likely due to the influence of compost application, root exudates and fertigation. Since the months of April-September are the driest months of the year at the study site, the large C and nutrient inputs during the growing season may have depended on irrigation water to be transported into subsoil layers, highlighting the importance of irrigation amounts and types (drip, furrow) to understanding
changes in subsoil C stocks. The shifts in soluble C:N:P:S ratios during the course of the year may also indicate that C:nutrient stoichiometry is more suitable for microbial biomass growth in these row cropped plots during the growing season than it is during the winter rainy season.

In contrast to the ORG system, SOC stocks did not significantly increase in the CONV+WCC plots after 25 years. While our FTIR results suggest that cover crop residues have an impact on subsoil C by increasing the proportion of carboxylate-C
relative to the CONV system (Figure 8b), they do not suggest a clear reason behind the lack of an increase in SOC stocks. A possible hypothesis is that small inputs of C and N over time from cover crop roots primed decomposition of native SOC, potentially by stimulating phosphatases and accelerating MAOM breakdown (Cui et al., 2020; Mise et al., 2020). Additionally, common root exudates such as oxalic acid may have dispersed organomineral complexes (Keiluweit et al., 2015), making that C more accessible for decomposition. While any priming of SOC due to cover crop root exudates would also be occurring in
the ORG systems, we believe this was counteracted by the higher EOC inputs and more favorable nutrient stoichiometry for microbial biomass provided by the compost.

We also observed a continual decline in SOC in subsoil in the conventional with cover crops treatment (CONV + WCC) as observed in Tautges and Chiartas et. al (2019); however, the rate of decline was lower over the last 7 years than in the first 19 years of study. This slower decline in subsoil C stocks from 2012-2019 may be due to the switch from furrow to drip irrigation
in 2014. Lower water inputs with drip reduces microbial activity and C and N cycling enzyme activities (e.g. beta-glucosidase and N-acetyl-glucosaminidase) in a large part of the bed in the surface of these same plots (Schmidt et al., 2018). The shift in irrigation and reduced water inputs potentially increased the prevalence of complex SOM by reducing microbial mineralization, and may have facilitated greater DOC transport during the winter rainy season.

## 5 Conclusion

The combination of growing cover crops and compost amendment created a unique set of conditions conducive to C transport and accumulation in the subsoils of a tilled row crop rotation. This was, in part, likely due to increased hydraulic conductivity facilitated by cover crop roots leading to higher rates of transport of soluble C and nutrients from the surface to subsoil. In turn, higher transport led to increased C stocks, reduced levels of microbial stress and higher available C, P and S values

throughout the year in ORG systems. The accumulation of oxygen-rich carboxylate C in subsoil horizons under all treatments,
attributed to an increase in microbially-processed C, provides support for the "cascade theory" of C transport. These results demonstrate the potential for subsoil C storage in tilled agricultural systems, and highlight a potential pathway for increasing C transport, storage, and sequestration in subsoil layers.

# 6 Appendices

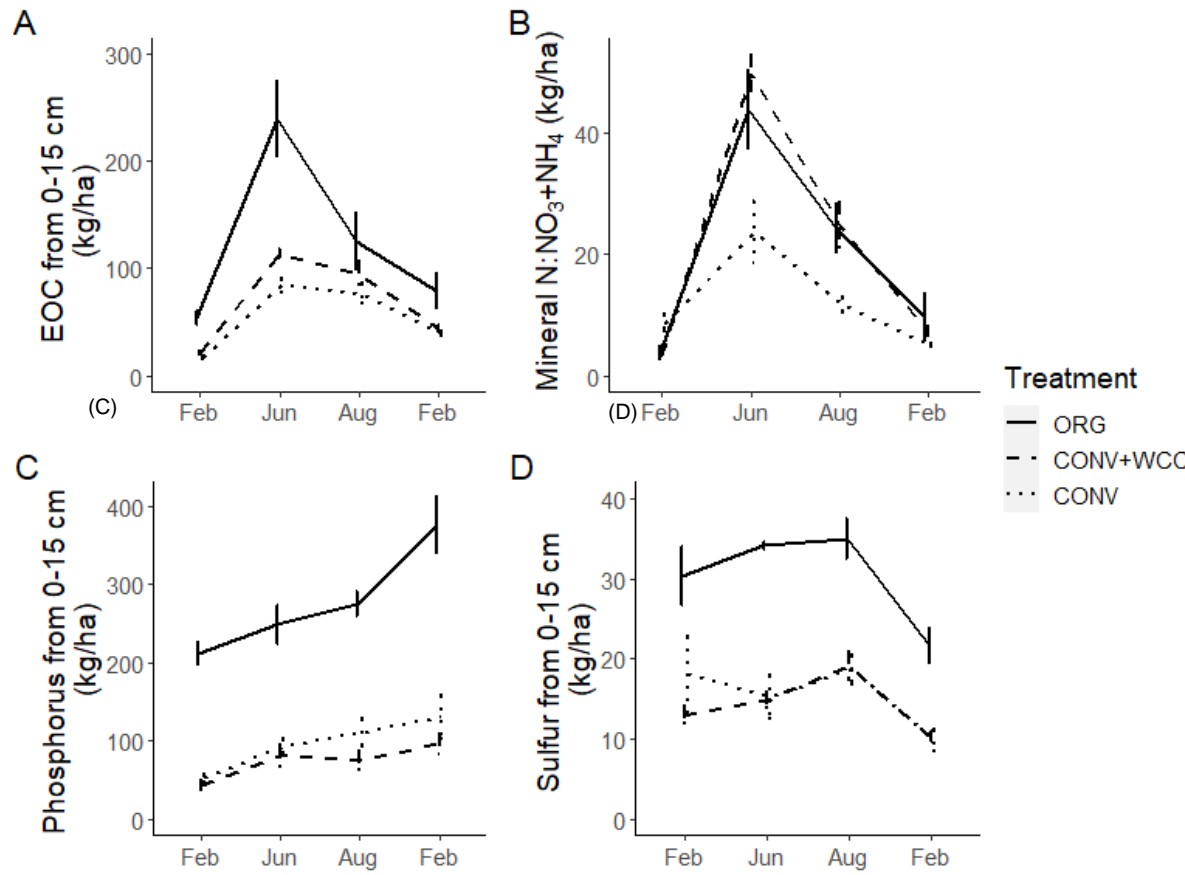

**Figure A1. Extractable organic C, mineral N, phosphorus, and sulfur stocks at 0-15 cm in ORG, CONV+WCC and CONV systems over the Feb 2018- Feb 2019 season. All values are given in kg/ha. Error bars represent standard error.**


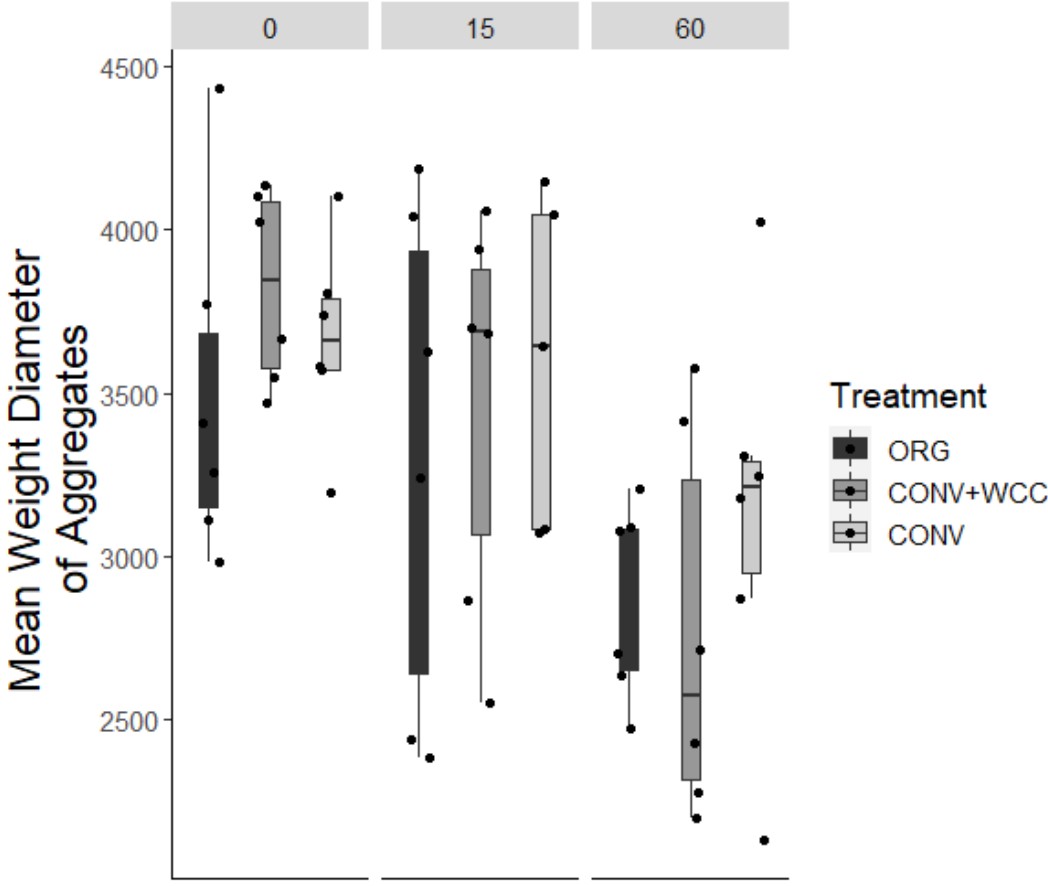

**Figure A2. Mean weight diameter of aggregates obtained by wet sieving for 0-15, 15-60 and 60-100 cm depth intervals in ORG, CONV+WCC and CONV systems.**


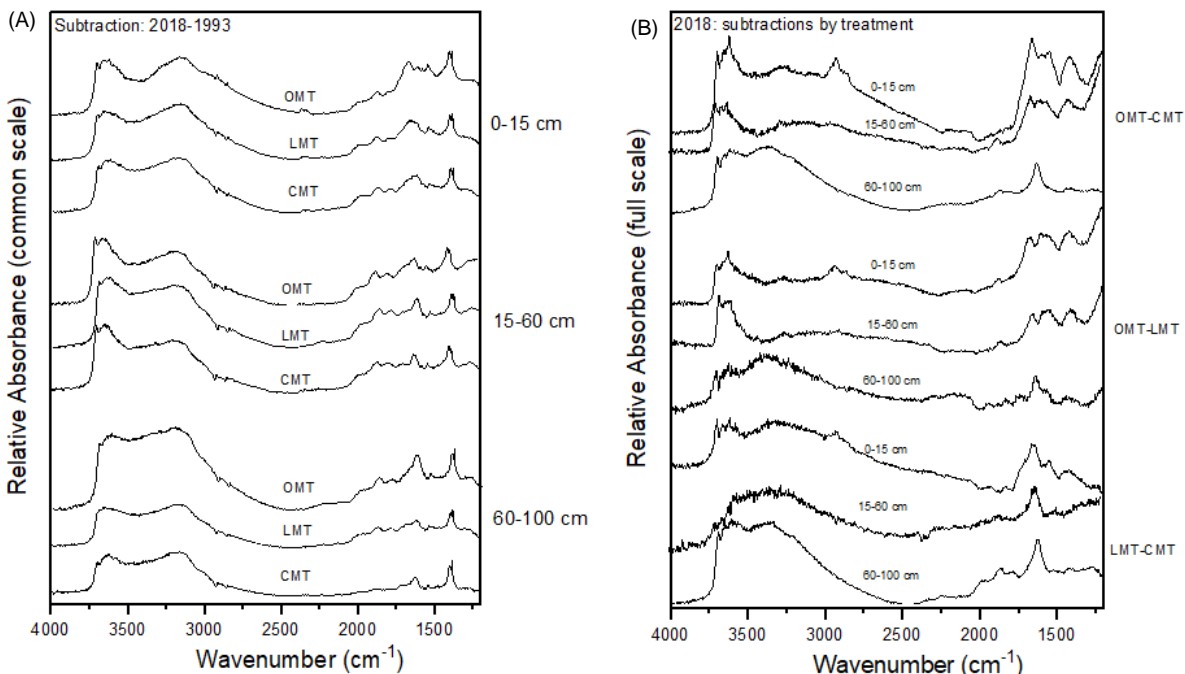

**Figure A3a,b. FTIR spectral subtractions for the 4000-1200 cm$^{-1}$ range comparing (A) 2018 -1993 spectra for ORG, CONV+WCC and CONV, and (B) ORG, CONV+WCC and CONV spectra in 2018.**


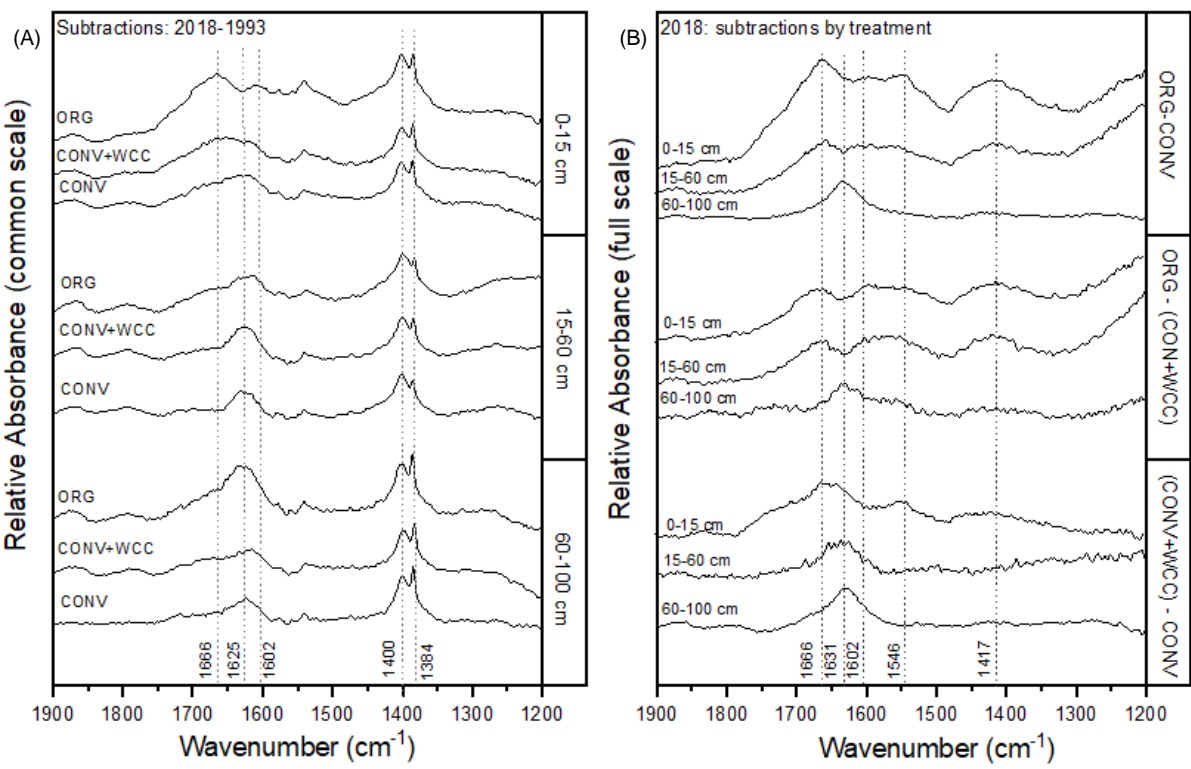

**Figure A4a,b. FTIR spectral subtractions for the 1900-1200 cm$^{-1}$ range comparing (A) 2018 -1993 spectra for ORG, CONV+WCC and CONV, and (B) ORG, CONV+WCC and CONV spectra in 2018.**

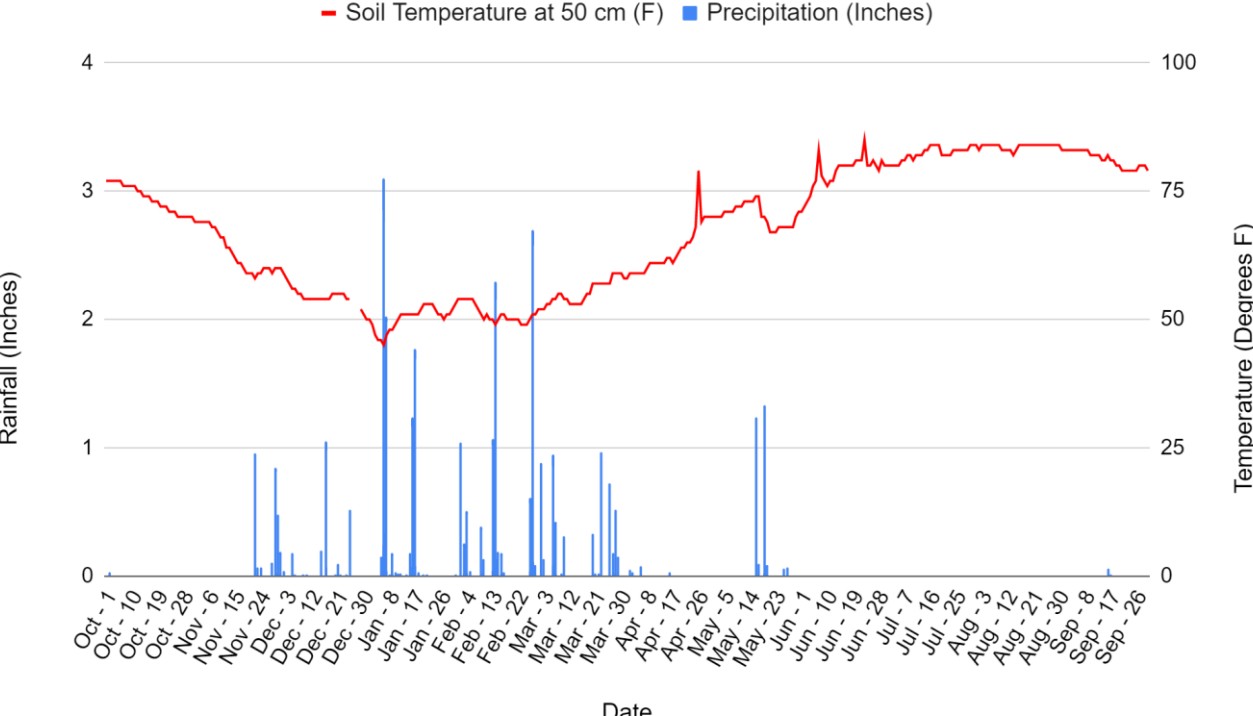

Figure A5. Rainfall and soil temperature at 50 cm at Russell Ranch from Oct 2018 to Oct 2019. Data taken from http://atm.ucdavis.edu/weather/uc-davis-weather-climate-station


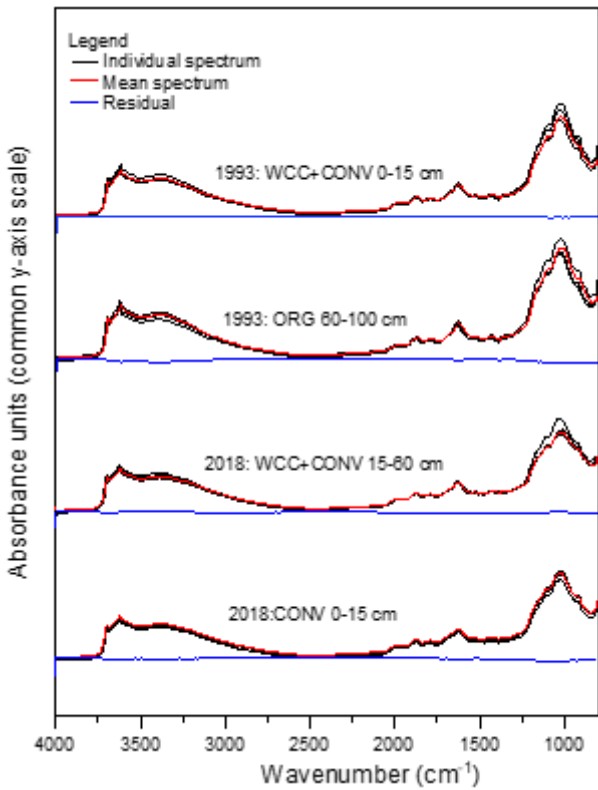

Figure A6. Example DRIFT spectra showing three replicate spectra, the mean spectra and the residual spectra for 1993 and 2008 samples.


| Gram Positive | 15:1 iso w6c; 15:0 iso; 15:0 anteiso; 16:0 iso, 17:1 iso w10c, 17:1 iso w9c; 17:1 anteiso w9c; 17:0 iso, 17:0 anteiso; 18:0 iso |
|---|---|
| Gram Negative | 16:1 w9c; 16:1 w7c; 17:1 w8c; 17:0 cyclo w7c; 18:1 w7c; 19:0 cyclo w7c; 20:1 w9c; 21:1 w3c |
| Saturated | 12:0; 14:0; 15:0; 16:0; 17:0; 20:0 |
| Monounsaturated | 16:1 w5c; 16:1 w7c; 18:1 w9c; 18:1 w7c |
| Cyclopropyl Indicator | 19:0 cyclo w7c / 18:1w7c |

**Table A1 – PLFA (Phospholipid Fatty Acid) Assignments taken from Bossio and Scow (1998).**

| Wavenumber (cm$^{-1}$) | IR Assignment |
|---|---|
| 2800-3100 | aliphatic $\nu_s(CH_2)$, $\nu_{as}(CH_2)$, $\nu_s(CH_3)$, $\nu_{as}(CH_2)$ |
| 1700-1765 | $\nu(C=O)$ |
| 1666 | aromatic $\nu(C=C)$ |
| 1620-1631 | $\nu_{as}(COO)$ |
| 1602 | skeletal $\nu(C=C)$ |
| 1546 | aromatic $\nu(C=C)$ |
| 1417 | $\delta(C-H)$ |
| 1400 | $\nu_s(COO)$ |
| 1384 | $\nu(C-O)$ vibration aromatic and $\delta(C-H)$ vibrations in $CH_3$ and $CH_2$ |

**Table A2 - FTIR Peak Assignments\* used for analysis of spectra**
\*Assignments taken from
Baes, A.U., Bloom, P.R., 1989. Diffuse reflectance Fourier transform infrared (DRIFT) of humic and fulvic acids. Soil Sci. Soc. Am. J. 53, 695–700. doi:10.2136/sssaj1989.03615995005300030008x**;**
Hesse, M., Meier, H., & Zeeh, B. (2005). Spektroskopische Methoden in der Organischen Chemie. (In German.) Georg Thieme Verlag,
Stuttgart. doi:10.1002/pauz.19960250417
Parikh, S.J., A.J. Margenot, F.N.D. Mukome, F. Calderon, and K.W. Goyne. 2014. Soil Chemical Insights Provided through Vibrational Spectroscopy. Adv. Agron. 126:1-148
Orlov, D.S., 1986. Humus acids of soil. Rotterdam: Balkema. doi:10.1002/jpln.19871500116


| Soil Series | Rincon | Yolo |
|---|---|---|

| Soil Taxonomic Class | Fine, smectitic, thermic Mollic Haploxeralfs | Fine-silty, mixed, superactive, nonacid, thermic Mollic Xerofluvents |
|---|---|---|
| | | |
| Horizon Designation | AP | AP |
| Depth (cm) | 0-10 | 0-5 |
| Texture | SiCL | SiL |
| pH | 6.5 | 6.7 |
| Organic Matter Content (%) | 2.4 | 2.4 |
| Clay Content (%) | 31 | 30 |
| | | |
| Horizon Designation | A12 | AP2 |
| Depth (cm) | 10-41 | 5-20 |
| Texture | SiCL | SiL |
| pH | 6.5 | 7.1 |
| Organic Matter Content (%) | 2.4 | 2.4 |
| Clay Content (%) | 31 | 30 |
| | | |
| Horizon Designation | B21t | A1 |
| Depth (cm) | 41-64 | 20-48 |
| Texture | SC | SiL |
| pH | 7 | 7.2 |
| Organic Matter Content (%) | 0.75 | 1.8 |
| Clay Content (%) | 40 | 30 |
| | | |
| Horizon Designation | B22t | A2 |
| Depth (cm) | 64-79 | 48-66 |

| Texture | SC | SiL |
|---|---|---|
| pH | 7.9 | 7.3 |
| Organic Matter Content (%) | 0.75 | 1.3 |
| Clay Content (%) | 40 | 30 |
|  |  |  |
| Horizon Designation | B3tca | C1 |
| Depth (cm) | 79-102 | 66-84 |
| Texture | SCL | SiL |
| pH | 8 | 7.4 |
| Organic Matter Content (%) | 0.75 | 1 |
| Clay Content (%) | 40 | 28 |
|  |  |  |
| Horizon Designation |  | C2 |
| Depth (cm) |  | 84-104 |
| Texture |  | SiL |
| pH |  | 7.4 |
| Organic Matter Content (%) |  | 0.8 |
| Clay Content (%) |  | 25 |

Table A3. Texture, pH, OM, and clay content for the Rincon and Yolo soil series found at Russell Ranch.

| Input (Mg/Ha/yr) | CONV | CONV+WCC | ORG |
|---|---|---|---|
| Tomato Residue | 1.73 | 1.57 | 1.54 |
| Corn Residue | 2.43 | 1.96 | 2.25 |
| Compost | 0 | 0 | 1.9 |
| WCC Residue | 0 | 1.086 | 1.48 |
| WCC Nitrogen | 0 | 0.09 | 0.11 |
| Compost Nitrogen | 0 | 0 | 0.19 |

| | | | |
|---|---|---|---|
| Mineral N | 0.18 | 0.13 | 0 |
| Mineral Phosphorus | 0.04 | 0.04 | 0 |
| Organic Phosphorus | 0 | 0 | 0.13 |
| Mineral Sulfur | 0.08 | 0.09 | 0 |
| Organic Sulfur | 0 | 0 | 0.05 |

**Table A4. Average annual inputs of Carbon, Nitrogen, Phosphorus and Sulfur to ORG, CONV+WCC and CONV plots between 1993-2018. Please note that Sudangrass and Wheat C inputs were excluded from this table, as they were only grown for a limited amount of time.**


| Practices - 2018-2019 | CONV | CONV+WCC | ORG |
|---|---|---|---|
| Compost Application | NA | NA | Apr 2018 and Oct 2019 |
| Corn Harvest | Sept 10th | Sept 10th | Sept 10th |
| Tomato Harvest | Aug 2nd | Aug 2nd | Aug 2nd |
| Total Amount of Irrigation (mm/hectare) | 245.13 | 239.43 | 538.27 |
| Number of tractor passes/yr (7.5cm deep) | 4 | 13 | 16 |
| Number of tractor passes/yr (20.5cm deep) | 4 | 4 | 4 |

**Table A5. Management summary for ORG, CONV+WCC and CONV plots for the 2018-2019 year.**

## 7 Code availability

The code for the graphs and analyses in this manuscript is available at https://github.com/danrath/2018_RRCARBON_DEPTH (DOI: https://zenodo.org/badge/latestdoi/181972884)

## 8 Data availability

The data included in this manuscript is part of the Russell Ranch long term dataset and is available at https://github.com/danrath/2018_RRCARBON_DEPTH. (DOI: https://zenodo.org/badge/latestdoi/181972884)

## 9 Author contribution

NB, AAB, SY, KS, SP and DR acquired funding for this project. DR, NT, NB, KS and DW designed the experiment and DR, NB, DW, LD, SP, NT collected the data. DR, NB, LD, and SP performed data analysis and interpretation with assistance from DW, NT, KS, SY and AAB. DR prepared the manuscript with contributions from all co-authors.

## 10 Competing interests

The authors declare that they have no conflict of interest.

## 11 Acknowledgements

Funding for this research was made possible by the University of California Office of the President, and the USDA National Institute of Food and Agriculture (NIFA) through the UCOP Catalyst Grant, the USDA-NIFA AFRI ELI Predoctoral Fellowship, Hatch Formula Funding CA 2122-H and multistate regional project W-2082. All opinions, findings, conclusions, and recommendations expressed in this publication are those of the authors and do not necessarily reflect the view of the National Institute of Food and Agriculture (NIFA), nor the United States Department of Agriculture (USDA). Authors would

also like to express gratitude to Israel Herrera and the Russell Ranch staff for field assistance during the project, and Danielle Gelardi and Oren Hoffman for offering comments on the draft version of the manuscript.

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
