# Peer review of "Synergy between compost and cover crops in a Mediterranean row crop system leads to increased subsoil carbon storage"

_SOIL, 2021_

## Author Comment (AC2)

**Reviewer 1 Comments**

**In this manuscript, the authors present a compelling hypothesis of how compost in combination with winter cover crops can lead to accumulation of aromatic-rich subsurface soil carbon. The hypothesis is complex but plausible whereby cover crop roots improve soil structure/porosity facilitating greater transport of soluble C and nutrients derived from the compost directly to the subsurface where this C can be stabilized. While the hypothesis is compelling, unfortunately, I do not think the authors have collected the right data to test this hypothesis.**

Response: We would like to thank the reviewer for their well-thought out, fair, and constructive feedback. We will address their concerns by making clearer what evidence we have for the different components of the hypothesis, better clarifying where there are limitations of the data, and drawing more on existing literature to support our synthesis as described below.

**Utilizing a long-term field trial should be a great way of trying to address this hypothesis. However, a major limitation of the study is that there is no compost-only treatment, so there is no way to separate the effect of compost alone from the interactive effect of compost and cover crops together. There is nothing the authors can do about this except recognize this as a limitation of the study design.**

Response: We will make this limitation of the experimental design clearer, stating that there is no compost-only treatment in the methods, and restructuring the discussion section to make clear that we cannot differentiate between the effects of compost and winter cover crops (WCC) individually.

**A major feature of the author's hypothesis is that cover crop roots have created greater porosity that facilitates greater water flow down the soil profile. The data simply do not support this notion. The authors find no difference in saturated hydrologic conductivity at 35 cm (although there was a trend for much greater variability in the compost + cover crop treatment) and no difference in soil aggregates across treatments.**

Response: We agree that the discussion needs to be clearer and restructured.  We will modify the discussion as follows:
1. We will clarify the non-significant treatment means of the hydraulic conductivity data.
2. We will incorporate data from previous studies at Russell Ranch to support our hypothesis of increased water storage and movement in cover-crop plots.
3. We will clarify that our conclusion of greater porosity due to WCC roots is based on previous observations and not a parameter that we directly measured in this study.

**The only significant difference was greater water content in the two treatments with cover crops but the authors did not measure bulk density in the 2018**

**samples and they did not measure porosity so it is difficult to come up with an explanation for this observation.**

Response: The idea that cover crops increase infiltration and hydraulic conductivity in fine-textured soils is not new (see Haruna et al., 2018; Çerçioğlu et al., 2019; Gulick et al., 1994). Previous work carried out at Russell Ranch over the last 30+ years has shown that the cover crop mix of hairy vetch, faba beans and oats used in this study increased infiltration and DOC input into the soil profile (Mailapalli et al., 2012). It also increased soil moisture-holding capacity during saturated conditions (Joyce et al. 2002) with no significant differences in bulk density (Colla et al. 2000) and reduced soil surface strength (Folorunso et al. 1992).

Proposed mechanisms behind the increased infiltration and moisture storage include increased aggregation, reduced crusting, and increased macroporosity. It is important to note that these root-induced soil alterations are highly localized and dependent on the root architecture of the cover crops. Specifically, cover crops with prominent tap roots (faba bean) are effective at creating continuous bio-pores, while fibrous roots (oat and hairy vetch) are effective at forming aggregates (Oglive et al., 2021). Therefore, the mixture of cover crops planted at the site likely resulted in widely variable aggregation and pore connectivity effects.

In the context of these previous observations, we attribute the increased variability of our hydraulic conductivity measurements and the increased moisture storage observed in ORG and CONV+WCC plots to the presence of WCCs.

In addition, the sample size used for Ksat measurements (cross-sectional area of 250 cm2) may be too small to capture the effects of cover crop roots, whose impacts are likely to be detected at a larger scale (Ozelim and Cavalcante, 2017). It is well recognized that hydraulic conductivity measurements can vary widely across fields and landscapes (Rahmati et al. 2018) and often do not account for the presence of macropores (Brooks et al. 2004). Ksat values can span 3 orders of magnitude within a short distance (Øygarden et al. 1997). Thus, although our measurements do not reveal statistically significant differences between the treatments, the scattered high-permeability zones in the cover-crop treatments are likely to contribute to rapid moisture redistribution and may explain the elevated deep moisture.

Other potential, albeit less likely, explanations for increased moisture content include lateral subsurface flow (unlikely due to low slope of field) and differences in runoff and runon (also unlikely due to low slope).

[revised manuscript text omitted]

**The next major component of the hypothesis is that compost leads to greater soluble C and N. The authors use salt-extractions of soil samples at four time points during the 2018 season to generate supporting data. Salt-extractable C is an interesting carbon pool (a potentially soluble pool of C) but there is ample evidence that this lab-extracted pool has little relationship to DOC when collected in lysimeters in the field.**

Response: We recognize that the distinction between dissolved organic carbon (DOC - total organic carbon dissolved in the liquid phase of a bulk soil sample) and extractable organic carbon (EOC - total dissolvable organic carbon in the liquid and solid phases of a bulk soil sample) is important. We will make this differentiation clear by replacing all mentions of "DOC"  with "EOC", and also reflect the limitations of our extraction approach in our discussion. We recognize that lysimeters are invaluable for making realistic measurements of DOC in soils, but due to moisture limitations at our study site it was not feasible to use them. Measurements of EOC are commonly used to estimate soluble carbon (Slessarev et. al 2020, Matlou et. al  2007) and we have followed recommended best practices for extraction (Li et. al 2018) using field moist samples with $K_2SO_4$ extractant at a 1:5 ratio.

1.  Li, S., Zheng, X., Liu, C., Yao, Z., Zhang, W. and Han, S.: Influences of observation method, season, soil depth, land use and management practice on soil dissolvable organic carbon concentrations: A meta-analysis, Sci. Total Environ., 631–632, 105–114, https://doi.org/10.1016/j.scitotenv.2018.02.238, 2018.

2. Matlou, M. C. and Haynes, R. J.: Soluble organic matter and microbial biomass C and N in soils under pasture and arable management and the leaching of organic C, N and nitrate in a lysimeter study, Appl. Soil Ecol., 34(2–3), 160–167, https://doi.org/10.1016/j.apsoil.2006.02.005, 2006.
3. Slessarev, E. W., Lin, Y., Jiménez, B. Y., Homyak, P. M., Chadwick, O. A., D'Antonio, C. M. and Schimel, J. P.: Cellular and extracellular C contributions to respiration after wetting dry soil, Biogeochemistry, 147(3), 307–324, https://doi.org/10.1007/s10533-020-00645-y, 2020.

**Without direct collection of DOC diffusing and advecting down the soil profile it is difficult to say whether the differences in the extractable pools are actually leading to more DOC flux to the subsoil under compost addition.**

Response: We agree that it would have been desirable to directly measure DOC diffusing and advecting down in the soil profile; however it was not feasible to use lysimeters in our study due to the extremely dry conditions at Russell Ranch during the summer months.

Tension lysimeters primarily obtain soil solution from macropores (Chantigny et. al 2003) that may be empty during the summer at our experimental location, while EOC extractions can account for soluble carbon in the solid phase (bound to mineral surfaces) as well as in smaller micropores. Additionally, EOC may be more sensitive to recent carbon and litter inputs, while DOC is more indicative of older, extant soil carbon (Froberg et. al 2007). While DOC has been used in studies of carbon transport in both grassland and forest ecosystems (Sanderman and Amundson 2008), EOC may be more suitable for answering questions on the impacts of carbon input, nitrogen amendment and tillage (Li et. al 2018), such as in our manuscript.

While EOC measurements vary seasonally less than DOC measurements, there is still a significant amount of variation that may occur during a growing season (Li et. al 2018). Our sampling regime at multiple timepoints was meant to account for some of that variation in both winter and summer months, and the EOC extraction method was chosen as the best way to compare soluble C measurements at different timepoints and soil water contents.

Finally, our inference of greater soluble carbon under compost application is supported by the multiple lines of evidence that we present in the manuscript:

1) More soluble C in ORG subsoils,
2) Observations of higher water infiltration and storage under cover crops
3) Greater amounts of soluble organic carbon in compost, and
4) Reduced subsoil microbial stress indicators under ORG systems (attributed to more C availability)

1. Chantigny, M. H.: Dissolved and water-extractable organic matter in soils: A review on the influence of land use and management practices, in Geoderma, vol. 113, pp. 357–380, Elsevier, https://doi.org/10.1016/S0016-7061(02)00370-1, , 2003.
2. Fröberg, M., Berggren Kleja, D. and Hagedorn, F.: The contribution of fresh litter to dissolved organic carbon leached from a coniferous forest floor, Eur. J. Soil Sci., 58(1), 108–114, https://doi.org/10.1111/j.1365-2389.2006.00812.x, 2007.

3. Sanderman, J. and Amundson, R.: A comparative study of dissolved organic carbon transport and stabilization in California forest and grassland soils, Biogeochemistry, 89(3), 309–327, https://doi.org/10.1007/s10533-008-9221-8, 2008.
4. Li, S., Zheng, X., Liu, C., Yao, Z., Zhang, W. and Han, S.: Influences of observation method, season, soil depth, land use and management practice on soil dissolvable organic carbon concentrations: A meta-analysis, Sci. Total Environ., 631–632, 105–114, https://doi.org/10.1016/j.scitotenv.2018.02.238, 2018.

**The third component of the hypothesis relates to the preferential partitioning of DOC chemistry down the soil profile. The evidence here is particularly weak. Mid infrared FTIR spectroscopy is not a quantitative analytical tool for determining abundance of specific compounds. If it were, labs wouldn't spend millions of dollars on more precise equipment. FTIR spectroscopy is good for identifying compounds in simple mixtures but not for quantifying their abundance in simple or complex mixtures (and soil is one of the most complex there is).**

Response: First, we want to make clear that we are not quantifying the amount of any specific compound in our samples. In this manuscript we use IR peak locations to demonstrate the presence (or absence) of particular chemical moieties, and the peak height as an indicator of abundance. We fully agree that quantification of specific chemical functional groups in soil organic matter via FTIR is challenging, and that completely accurate methods do not exist. There are some advances being made in this area (e.g., work by Francisco Calderon), but it remains pseudo-quantitative at best (more on this below).

FTIR is limited by the wavelength of IR radiation and thus development of more precise instrumentation is challenging. Since the development of the Michaelson interferometer in 1880 the IR instrument has remained relatively unchanged, with recent advances coming in data processing (such as the Cooley-Tukey algorithm) and sample collection. The ability of computers to conduct a Fourier transform of data, using the Cooley-Tukey algorithm in 1965 was a huge advance, but for data processing. Since that time, there have been many more advances on the software side and for sample collection (e.g., ATR). Some of the more cutting-edge advances are being made using synchrotron sources, which have greater intrinsic brilliances, combined with other techniques such as atomic force microscopy (AFM). However, these advances are most beneficial for reducing the "spot size" or "number of pixels" being probed; going from micron to nanometer scales. When it comes to analysis of complex mixtures, like soil, improvements in quantification will likely arise from more accurate peak fitting/deconvolution, or perhaps the use of internal standards (e.g., Calderon et al. 2013). Unfortunately, due to funding availability in different disciplines, advances in most analytical techniques are developed for other types of samples, such as polymers for industrial purposes, and trickle down for soils.

**Peak features depending on if they are due to vibrations, wiggles, combinations or overtones all have different relationships between abundance of the specific bonding environment and absorption – basically, you would have to prove that there is a linear relationship between "aromatics" and those two peak features in order to do a spectral subtraction and have any confidence that the difference spectrum represents real differences in chemistry.**

Response: We agree that the analysis of heterogeneous samples by mid-FTIR pose many challenges such as the overlapping of peaks and spectral artifacts. However, there is a deep history of the use of FTIR to study the chemical composition of soil organic matter. The reviewer is correct that FTIR cannot be used as a strictly quantitative tool for determining specific compounds in mixed samples, although the basic theory of FTIR does allow for this. As the reviewer mentions, IR absorption occurs when molecular bonds exhibit a dipole moment, and the resulting bands can result from bond stretching, wagging, rocking, etc. However, this absorption that can be quantified as the absorption of IR light by a specific molecular bond at a specific electromagnetic frequency (wavenumber) follows the Beer-Lambert Law (Beer's Law) (e.g., Margenot et al 2016, Smith 2001). Thus, the height and area are proportional to the concentrations of molecules in a sample (linear relationship). While this holds true for some FTIR sampling methods, such as transmission and attenuated total reflectance (e.g., Smith 2011, Parikh et al. 2008, Ge et al. 2014), there is deviation from Beer's Law when using reflectance techniques such as diffuse reflectance Fourier transform (DRIFT) spectroscopy (e.g., Baes and Bloom 1992; Niemeyer et al 1992), as the reviewer suggests. However, it has been shown that > 90% of the photons have shallow penetration and travel short pathlengths when diffusely reflected (Brauns 2014) and thus deviation from Beer's Law is expected to be somewhat minor.

Even taking in consideration those potential limitations, previous studies with DRIFTS in both the near-infrared (Dalal and Henry, 1986) and mid-infrared regions (Demyan et al., 2012; Margenot et al., 2015; West et al., 2020; Deiss et al., 2021) have shown direct associations between soil organic carbon concentration and absorbance at specific frequencies (depicted as peak height or area of single peaks or peak ratios). The non-linearity of concentration and absorbance arising from DRIFT can be partially corrected using the Kubelka-Munk (KM) function. However, since a KM correction does not fully correct spectra (e.g. Clark and Roush, 1984) we have still chosen to include the spectra uncorrected, but those are now in the appendix on a non-common scale with the spectral amplification factor noted for each spectrum. The Kubelka-Munk corrected spectra, on a common y-axis scare, will be placed in the main document.

[Figure]

Figure A3. DRIFT spectral subtractions for the 1900-1200 cm$^{-1}$ range comparing (A) 2018-1993 spectra for ORG, CONV+WCC, and CONV, and (B) ORG, CONV+WCC, and CONV spectra in 2018. Spectra are plotted with absorbance units on a non-common y-axis scale. The spectral amplification factor is noted on the right-hand side of each spectrum.

[Figure]

Figure 8. DRIFT spectral subtractions for the 1900-1200 cm$^{-1}$ range comparing (A) 2018-1993 spectra for ORG, CONV+WCC, and CONV, and (B) ORG, CONV+WCC, and CONV spectra in 2018. Spectra are plotted with Kubelka-Munk units on a common y-axis scale.

Of course, quantification is incredibly difficult in complex mixtures, and thus the concept of pseudo-quantitation is used by comparing spectral features within, and between, samples. However, in this case we expect a minor deviation in the linear relationship between aromatic C=C and $COO^-$ with a change in concentration due to data collection via DRIFTS as some unknowns remain regarding path length and scattering effects. Most important, for our study, is the presence and absence of peaks and the relative differences in spectral contributions from each peak. Assuming that band assignments are correct (they are consistent with literature), the presence of a peak at 1666 $cm^{-1}$ suggests greater abundance of aromatics in the surface horizons, especially for the ORG treatment. We will also update the manuscript to provide peak intensity ratios of aromatic to carboxyl moieties [$v(C=C):vas(COO^-)$ (1662 $cm^{-1}$:1631$cm^{-1}$)] to show the relative difference within each spectrum. Thus, we believe that the FTIR data collection and subtractions are appropriate to show semi-quantitative results of changes in soil chemical functional groups as affected by land use and soil depth.

Table 1. Peak Intensity Ratios for aromatic (1662 $cm^{-1}$) to asymmetric carboxyl ( 1631 $cm^{-1}$) groups in spectral subtractions.

| | Depth | Peak Intensity Ratio (1662 $cm^{-1}$ : 1631 $cm^{-1}$) | | |
|---|---|---|---|---|
| | | ORG | CONV+WCC | CONV |
| Subtraction: 2018-1993 | 0-15 cm | 1.38 | 1.22 | 0.45 |
| | 15-60 cm | 1.21 | 1.40 | 0.77 |
| | 60-100 cm | 1.17 | 2.50 | 0.014 |
| | | | | |
| | | ORG-CONV | ORG-(CONV+WCC) | (CONV+WCC) -CONV |
| Subtraction by treatment: 2018 | 0-15 cm | 1.39 | 1.18 | 0.46 |
| | 15-60 cm | 1.20 | 1.42 | 0.47 |
| | 60-100 cm | 0.38 | 0.64 | 0.45 |

**I also find it problematic that all treatments have showed the same increase in carboxylate functional groups over 25 years – wouldn't we expect the conventional treatment to be more or less at steady state, so we shouldn't see the same changes as seen in the cover crop and compost + cover crop treatments?**

Response: The subtraction spectra do not represent the total amount of COO etc, they represent the change in that functional group. But the reviewer is correct that they appear very similar when plotted using absorbance units, and we thank them for pointing this out. However, when the spectra are shown with the KM correction the spectral contributions for the change in carboxyl groups are no longer similar. We believe that these spectra along with the peak intensity ratio table will help make this clear. Additionally, the band at 1384 cm$^{-1}$ represents both aromatic and aliphatic carbon; however, we have determined that the intensity of that peak in the 2018-1993 spectra is impacted by an artifact of spectral reflection in the original spectra that is amplified during subtraction. It is possible that the artifact is a Restsrahlen band (e.g., carbonate), but it is more likely that trace amounts of $CH_2$ and $CH_2$ on the KBr beamsplitter (which are always present) resulted in a spectral reflectance artifact at 1384 cm$^{-1}$ when the soil samples diluted with KBr were analyzed and amplified via spectral subtractions. Although the intensity of the 1400 cm$^{-1}$ band is still meaningful, we focus on the asymmetric COO band at 1633 cm$^{-1}$ for pseudo quantification.

**Lastly, what is the actual magnitude of the "increase" in aromatic features in the compost treatment over the conventional treatment? There are no units on the y-axis. The authors have replicates so they could run statistics to see if this increase was significant.**

Response: In our original Figure 8a, FTIR spectra are shown on a common scale (displaying 0.07 absorbance units for each spectrum). The y-axis does not have units as the spectra are stacked. Overlaying the spectra would make observing differences very difficult. In the original Figure 8b, FTIR spectra are shown on a full scale (amplified) to maximize the observable differences between spectra. We felt this necessary as the differences between treatments in 2018 are much less than the differences between treatments from 1993 to 2018. However, in response to the reviewers concerns we will now use a) the spectra with KM units on a common y-axis scale and b) the spectra with absorbance units with a non-common y-axis scale labeled with their spectral amplification factor. Again, we believe the peak intensity ratios help to elucidate the relative changes in aromatic carbon between treatments and years.

We did not run statistics because of the very low variability between the spectral replicates of the same experimental treatments. We provide below an example of spectral replicates and residual spectra for some selected samples to demonstrate the low variability between spectra of the same treatment and depth (on a common y-axis scale).

[Figure]

Figure AX. Example DRIFT spectra showing three replicate spectra, the mean spectra and the residual spectra for 1993 and 2008 samples.

Response: We acknowledge that our manuscript did not describe the importance of these microbial stress indicators in detail, and will modify our discussion and introduction accordingly. We have also noted that our measurement of Sat:Unsat ratios is not significant with a p of 0.07, and will change our wording accordingly. We believe that the trends displayed in the sat: unsat ratio and cy17:pre ratios are indicative of a greater trend in microbial stress levels with depth despite their non-significance, and so have chosen to continue including them in the results section.

Phospholipid Fatty Acid analysis (PLFA) is a useful tool for looking at living cells in the soil environment (Zhang et. al 2019). Measurements of microbial stress responses via PLFA agree with those obtained via the more recent 16s rRNA metabarcoding (Orwin et. al 2018). PLFA stress ratios are useful as an indicator of limited carbon and water availability, as they represent an overall shift away from the thinner, more permeable cell membranes associated with Gram - bacteria and monounsaturated fatty acids; towards more tightly packed, less permeable cell membranes associated with Gram+ bacteria and saturated fatty acids (Silhavy et. al 2010).  An increase in the Gram+:Gram- ratio has been associated with a decrease in easily available water and carbon (Fanin et. al 2019, Fierer et. al 2003, Bossio et. al 1998), while an increase in the saturated: unsaturated ratio and cy17:pre ratios are associated with dehydrated conditions (Moore-Kucera et. al 2007). The stress indicator trends in our data agree with and support our observations of increased soluble carbon and water content in ORG systems.

Using PLFA to make predictions of functional or structural change in microbial communities is difficult, as many PLFAs are not specific to distinct species and instead distributed across taxa (Ruess et. al 2010). Rather than providing detailed information on taxonomy, PLFA permits evaluation of rapid changes in the cell walls and membranes of metabolically active soil microbes (Frostegård et. al 2010). As such, we have used the PLFA data to support the inferences drawn from our nutrient and moisture measurements. These results do indicate that a more detailed investigation into the carbon use efficiency and functional potential of these subsoil microbial communities may yield interesting results, and is the focus of another paper currently being drafted by the first author.

1. Bossio, D. A. and Scow, K. M.: Impacts of carbon and flooding on soil microbial communities: Phospholipid fatty acid profiles and substrate utilization patterns, Microb. Ecol., 35(3), 265–278, https://doi.org/10.1007/s002489900082, 1998.
2. Fanin, N., Kardol, P., Farrell, M., Nilsson, M. C., Gundale, M. J. and Wardle, D. A.: The ratio of Gram-positive to Gram-negative bacterial PLFA markers as an indicator of carbon availability in organic soils, Soil Biol. Biochem., 128, 111–114, https://doi.org/10.1016/j.soilbio.2018.10.010, 2019.
3. Frostegård, Å., Tunlid, A. and Bååth, E.: Use and misuse of PLFA measurements in soils, Soil Biol. Biochem., 43(8), 1621–1625, https://doi.org/10.1016/j.soilbio.2010.11.021, 2011.
4. Fierer, N., Schimel, J. P. and Holden, P. A.: Variations in microbial community composition through two soil depth profiles, Soil Biol. Biochem., 35(1), 167–176, https://doi.org/10.1016/S0038-0717(02)00251-1, 2003.
5. Moore-Kucera, J. and Dick, R. P.: PLFA profiling of microbial community structure and seasonal shifts in soils of a Douglas-fir chronosequence, Microb. Ecol., 55(3), 500–511, https://doi.org/10.1007/s00248-007-9295-1, 2008.
6. Orwin, K. H., Dickie, I. A., Holdaway, R. and Wood, J. R.: A comparison of the ability of PLFA and 16S rRNA gene metabarcoding to resolve soil community change and predict ecosystem functions, Soil Biol. Biochem., 117, 27–35, https://doi.org/10.1016/j.soilbio.2017.10.036, 2018.
7. Ruess, L. and Chamberlain, P. M.: The fat that matters: Soil food web analysis using fatty acids and their carbon stable isotope signature, Soil Biol. Biochem., 42(11), 1898–1910, https://doi.org/10.1016/j.soilbio.2010.07.020, 2010.
8. Silhavy, T. J., Kahne, D. and Walker, S.: The bacterial cell envelope., Cold Spring Harb. Perspect. Biol., 2(5), https://doi.org/10.1101/cshperspect.a000414, 2010.
9. Zhang, Y., Zheng, N., Wang, J., Yao, H., Qiu, Q. and Chapman, S. J.: High turnover rate of free phospholipids in soil confirms the classic hypothesis of PLFA methodology, Soil Biol. Biochem., 135, 323–330, https://doi.org/10.1016/j.soilbio.2019.05.023, 2019.

**Just to reiterate, I think the hypothesis laid out here for subsoil C accumulation under compost and cover crops is entirely plausible but the evidence in this study to support the hypothesis is not particularly strong.**

Response: We would like to again thank the reviewer for the time and energy they put into this review. These comments are extremely helpful. We hope that we have provided a more convincing argument with our responses to both reviewers' comments.

---

## Author Comment (AC3)

**Reviewer 2**

**Summary:**

**This manuscript leverages data from a long-term agricultural experiment at the Russell Ranch in California and a year of more detailed measurements to explore interacting cover crop and compost effects on subsurface soil carbon dynamics. Authors blend historical measurement of carbon stocks with present day analyses of carbon (bulk C, FTIR), nutrients (Mehlich-III), soil physical properties (aggregation, moisture content, and hydraulic conductivity), and microbial biomarkers (PLFA) at four sampling dates. An ANOVA was used to assess the effect of time, depth, and management, with subsequent separate analysis of differences between management treatments at each of three depths.**

**Although the experimental design and methods are sound, there is a disconnect between the objective to assess interaction of cover crops and compost and the data analysis. The discussion ties in interesting concepts such as the 'cascade theory' and microbial stress indicators that must be brought up further into the introduction to create a threat throughout the paper.**

RESPONSE: We greatly appreciate the amount of time and effort put into this review as evidenced by the extremely constructive comments provided! We will address the reviewer's concerns by reorganizing the manuscript, highlighting the limitations of our data, and drawing on previous research at Russell Ranch as described below..

**Below please find my recommendations to reframe the paper and utilize historic data, specific questions and a few line edits for authors. Most edits occur in the first half of the paper, which may help to connect the methods and results into the compelling discussion.**

**Title: I recommend making this more specific. The final sentence of the discussion states that "care should be taken when applying these results to different soil types and climates"; therefore, adding the soil type or climate (or both) into the title seems prudent.**

RESPONSE: Thanks for this helpful comment. We agree and will change the title to "Synergy between compost and cover crops in a Mediterannean row crop system leads to increased subsoil carbon storage"

**Abstract:**

**Throughout the paper, can authors use the treatment names as in the original experimental dataset (Wolf, 2018 page 6): CONV = CMT conventional maize-tomato, ORG=OMT organic maize-tomato, and (page 5) WCC – winter cover crop? I understand that Tautges and Chiartas 2019 used the CONV, ORG notation, but a brief explanation would be helpful.**

RESPONSE: As the 2018 Wolf paper was primarily a data paper, and the 2019 Tautges and Chiartas paper is more directly comparable to this manuscript, we chose to use the 2019 abbreviations. We will include a short explanation in the methods to make the difference between abbreviations used in the 2019 Tautges and Chiartas paper and 2018 Wolf paper clearer.

**The theory of cover crops providing a macropore system for transport of DOC is interesting, but the data do not support this theory (no measurement of porosity, change in bulk density, or changes in soil hydraulic properties). It is appropriate for a discussion, but I might exclude this as a main finding from the abstract.**

RESPONSE: We will adjust the abstract to remove the last two sentences, and replace them with a less explicit statement:  "Our results show the potential for increased subsoil carbon storage under compost + cover crops in tilled agricultural systems, and identify potential pathways for increasing carbon transport and storage in subsoil layers."

Introduction:

**I appreciate that the abstract and introduction mention soil health, but there is no clear definition or explanation of its importance to the paper. Either simply remove this term and focus solely on soil carbon and microbial processes, or please directly connect soil health and often associated shallow sampling regimes to this "outsized perceived role in ecosystem services".**

Response: We agree that inclusion of reference to soil health is not consistent with the topic of this manuscript. We will remove the mention of soil health from both the abstract and introduction. We will also clarify the sentence "outsized perceived role in ecosystem services" to indicate that increased sampling of the top 15 cm of soil marks a growing reliance on the soil surface to answer questions about processes in the entire soil profile, and runs the risk of subsoils being treated merely as "more dilute topsoils" (Salomé et. al 2010).

1. Salomé, C., Nunan, N., Pouteau, V., Lerch, T. Z. and Chenu, C.: Carbon dynamics in topsoil and in subsoil may be controlled by different regulatory mechanisms, Glob. Chang. Biol., 16(1), 416–426, https://doi.org/10.1111/j.1365-2486.2009.01884.x, 2010.

**This is a good argument and dataset to support deeper sampling. Authors may also include references summarized by Mobley et al 2015 in their article "Surficial gains and subsoil losses of soil carbon and nitrogen during secondary forest development": Post & Kwon, 2000; West & Post, 2002  review 360 articles on land use change, with only 10% sampling below 30cm.**

Response: We will include the suggested references in the introduction.

**In this paragraph, please clarify, at what depth are the authors designating topsoil v subsoil for this study?**

Response: We will insert a statement that we are designating 0-15cm as topsoil and 15-100 cm as subsoil.

**This first paragraph of the introduction discusses "longer C residence times" of deep soil C, which requires further explanation.**

Response: We will insert a sentence into the "C chemistry" section in the reorganized introduction explaining the trend of greater $C_{14}$ ages in subsoil layers. The age of carbon below 30 cm generally increases with depth, with estimates of carbon as old as $10^3$-$10^4$ years in deeper regions of the subsoil (Rumpel et. al 2012). This is in contrast to the younger $C_{14}$ ages of $10^2$-$10^3$ years in the top 30 cm (Paul 2001).

1. Rumpel, C., Chabbi, A. and Marschner, B.: Carbon storage and sequestration in subsoil horizons: Knowledge, gaps and potentials, in Recarbonization of the Biosphere: Ecosystems and the Global Carbon Cycle, pp. 445–464, Springer Netherlands, https://doi.org/10.1007/978-94-007-4159-1_20, , 2012.
2. Paul, E. A., Collins, H. P. and Leavitt, S. W.: Dynamics of resistant soil carbon of midwestern agricultural soils measured by naturally occurring 14C abundance, Geoderma, 104(3–4), 239–256, https://doi.org/10.1016/S0016-7061(01)00083-0, 2001.

**Overall, the introduction structure can be strengthened by clarifying topic sentences (e.g., specify cover crops L51) and adding updated references. Can you support the Jenny citation with more modern references, even Brady and Weill Nature and Properties of Soils, or USDA technical information "Designations for Horizons and Layers" in Soil Survey Manual – Ch 3 (https://www.nrcs.usda.gov/wps/portal/nrcs/detail/soils/ref/?cid=nrcs142p2_054253#designations).**

Response: We will update the references to include more recent citations related to soil formation.

**The introduction structure may flow better using paragraphs separated into chemical, physical and biological controls or layered as (1) depth; (2) chemistry of C inputs and stabilization at depth; (3) management impacts at depth – specifically cover crops; and (4) management interaction with other factors (microbial).**

Response: We agree and thank you for this suggestion. We will restructure the introduction to provide separate paragraphs for the impact of depth, C chemistry (introducing FTIR and the cascade theory), management impacts (tillage, cover crops, compost application) and microbes (PLFA stress biomarkers, nutrient stoichiometry) on C storage.

**The introduction touches upon stoichiometry, a critical highly manipulated factor in managed conventional systems that effects soil C storage. To go further in depth on soil chemistry (e.g., at L40), authors can address changes over time in stoichiometric constraints on decomposition (e.g., see Soong et al 2019 "Microbial carbon limitation: The need for integrating microorganisms into our understanding of ecosystem carbon cycling").**

Response: We will include the suggested reference, and insert a short discussion of the importance of stoichiometry in the microbes and C storage introduction section.

**Also, authors can mention higher physical disturbances in surface soils (L55), and the types of management associated with cover crops, such as crimping/rolling.**

Response: We will include a short discussion of tillage effects in the "management" section of the introduction.

**Please also include specific soil type, climate and cropping system when comparing to other studies, otherwise direct comparisons are not particularly informative.**

Response: We will add detail about soil type, climate and cropping system whenever we make comparisons to other studies.

**Can also cite McClelland et al 2020 "Management of cover crops in temperate climates influences soil organic carbon stocks: a metaanalysis" that analyzed soils only down to 30cm.**

Response: We will include this citation in the manuscript.

**As for the sampling strategy by depth, can the authors please describe why they separated out into these depths 0-15, "intervening", and the subsurface as 60-100cm? How do these depths compare to the horizons in these two soils? (Looking up the series descriptions Yolo has A horizons down to 66cm and then C horizons, and Rincon has A down to 20, B 20-100cm. Should the analysis be completed on A and B horizons rather than depth profiles?) How do these depths relate to roots of corn (100cm+), tomato (60cm+) and cover crops (variable)?**

**Can authors please justify why 15-60cm is combined into a single sample in 2018, when historical data had an additional delineation? (Is it simply limited time/costs or another reason?)**

Response: The depths used in this experiment were selected for the ability to compare directly to the results of the Tautges and Chiartas (2019) paper. Combining the 15-30 and 30-60 cm depth intervals that were used in Tautges & Chiartas et. al 2019 was originally done due to limitations of time and resources in processing samples. Authors endeavoured to have all time sensitive analyses done within 1 week of sampling, and were forced to limit the total number of samples to make that happen.

The decision to keep 0-15 cm separate instead of combining the 0-15 and 15-30 cm depths was based on preliminary data showing significant differences among treatments in genes coding for carbon degrading enzymes at 0-15 cm. These differences were not present at deeper depths (15-30, 30-60 or 60-100cm ) (D. Rath, unpublished data).

The Yolo soil series is a young soil (Entisol) without strong horizonation or changes in texture with depth, which we used to justify using depth-based increments instead of horizon-based increments. We did not detect a significant effect of block (correlated with soil type) on any of the variables measured in the experiment, and so applied the same

depth increments to the plots in the Rincon soil series, despite the fact that it has slightly more evidence of clay accumulation (Alfisol).

**Please stay consistent with the terms "subsoil" versus "subsurface soil", as depth is a major component of this study.**

Response: We originally used "subsurface soil" to refer to soils between 15 and 100 cm and "subsoil" to refer to soils deeper than 100 cm based on Rumpel et. al (2012). In order to reduce confusion we will use the term "subsoil" to refer to soils at all depths below 15 cm.

**The overarching question and hypothesis require further editing to clearly lead into the results and discussion. There seems to be a disconnect between the main question and the methods of this paper. The main question includes "carbon formation" (does that mean microbially processed C? or stabilizedC formation?) and "storage processes" (that obviously includes aggregation, but the carbon content of these size classes was not measured). Also, what is meant by the term "SOC-related indicators", does that mean SOC stability or reactivity-related indicators? As written the hypotheses are just predictions, there is no description as to the mechanisms behind the described expected results.**

Response: We agree that we need to strengthen our question and hypotheses and thank you for your suggestions. Accordingly , we will edit our manuscript to clarify the overarching question and hypotheses and clarify "carbon formation" to mean stabilized carbon formation.

We will also remove the term "storage processes" from the overarching question - we did not do any measurements of actual processes, but our results taken together suggested a mechanism by which carbon stocks were increased in ORG systems. The term SOC-related indicators refers to the connections between C and nutrient input, microbial biomass and SOC formation; we will remove this term and replace it with a more specific statement.

We will modify the hypothesis section to highlight that the main hypothesis of the paper is that "*high concentrations of mobile C and essential nutrients for microbial activity provided by the compost, combined with the easier movement of water downward associated with a history of cover-cropping, helped transport the material needed to build C in the subsoil*" as suggested in a later comment.

**An interesting hypothesis arises in the discussion around cascade theory, can authors pull that into the introduction? This can provide a way to integrate the study of carbon chemistry (FTIR) and microbial biomarkers that otherwise are not included in the hypotheses.**

Response: We will include a short discussion of the cascade theory in the introduction, under the reorganized "C chemistry" section.

**Finally, I agree with the previous reviewer comment, that the treatments CONV (fertilizer), CONV+WCC (fertilizer + cover crops), and ORG (compost + cover crops) do not disentangle the effect of compost. I don't think there is there a treatment in the Century Experiment that was maize-tomato plus compost only or fertilizer + compost, but this should be mentioned as a limitation in the study, particularly in the subtraction of FTIR spectra.**

Response: We will include a clear statement in the methods and introduction that there is no compost-only treatment, and make it clear that we cannot compare the effects of compost alone to the effects of cover crops alone due to the experimental design. We will also include this in our discussion of the limitations of the data.

**This manuscript covers many aspects of deep soil C and management, no need to emphasize the complicated factors of global change (L87) at the end of the introduction, unless those are also analyzed over time.**

Response: We agree and will remove this final sentence from the introduction.

**Materials and Methods:**

**Thank you for a concise description of the site and experiment. I recommend authors also add basic climate data such as climate type, mean annual mix and max temperature, mean annual precipitation, and also specific 2018-19 climate data for comparison.**

**Authors write that the 'horizon information' is available from Wolf et al, but I only can find soil chemistry by depth, not the soil description in that dataset (horizon delineations are online). Can authors add in the horizon depth into the methods for both the Yolo and Rincon soils, and key chemistry such as pH and texture? A table in the materials and methods section could organize all of this soil and climate information for quick reference.**

**This could also include the other key management notes that will impact DOC transport, such as the conversion from furrow to drip in 2014, as well as information from the 2018-2019 season such as crop planting/harvest dates, total irrigation amount, and the anomalous compost application in September 2019. These details can then be incorporated smoothly into the discussion.**

Response: Thank you for the suggestion to include additional data on climate, horizon and management. We agree that the Wolf et al. manuscript does not include soil description (horizons); this was an oversight on our part. We will include tables in the Methods/Appendix section describing:

1) Climate variables (average, 1993, 2018) including MAT, MAP, Min Temp, Max Temp
2) Soil horizon variables including texture, layer depths, pH, and %C estimates from SSURGO data

3) Management practices and dates

**The differentiation between the sampling and analysis of the older data and 2018-2019 methods is now clearer. Thank you for the new methods section. However, without hypotheses asking seasonal questions over time – why sample at four time points in a single year? Particularly as the authors state that a single year of data is not sufficient to look at differences at depth (L81-82) to justify use of the historical data. Perhaps authors can create one or two hypotheses for the 2018-19 season, and other for the long-term effects and historical data.**

Response: We sampled at 4 time points throughout the year based on the hypothesis that EOC and available nutrients in ORG systems would be higher only at 1 or 2 time points (June and Aug), and to get an idea of how much variation there was during the year. We did not include this discussion of seasonal variation during the year for brevity's sake, choosing instead to focus on what the overall trends indicated about C and nutrient transport. We will modify the manuscript to include a short discussion of seasonal variation in our measurements, and the potential implications for stoichiometry , microbial biomass and C formation.

**The use of PLFA and FTIR is not justified from the hypotheses or introduction. The use of these techniques, particularly stress ratios for PLFA needs to be explained within a wider context in the introduction.**

Response: We will include short descriptions and justifications for use of these methods in the introduction.

**2.7 Please clarify the statement that 9 out of 18 plots were sampled for hydraulic conductivity (those under tomato). Were half of the plots under corn and the other under tomato during this 2018 sampling? That needs to be included in the methods section. Or are you referencing the full 18 plots of all the Century experimental treatments? Finally, why are 8 dates included for soil moisture content, when soils are sampled only 4 times?**

Response: To clarify, there were 18 plots total included in our 2018 sampling - 6 ORG, 6 CONV+WCC, and 6 CONV. For each of these treatments, 3 plots were planted with tomato, and 3 plots were planted with corn. This gives a total of 9 plots under corn, and 9 plots under tomato.

For the hydraulic conductivity sampling in August, we only sampled the 9 plots under tomato as we were not able to access the corn plots due to harvesting activities. We will modify the methods section to make this clearer by including the number of replicates in a table.

In regards to the dates for soil moisture content, this was an error on our part: moisture content was measured 8 times, not 4 times. We will correct our methods section accordingly.

**2.8 I have some concern over the use of averaging and subtraction of the spectra. What was the variance between the historic soils of 15-30 and 30-60 cm?**

Response: The variation between 15-30 and 30-60 cm soils in 1993 was negligible for all three systems. We have included a sample graph of the residuals in 1993 ORG soils to illustrate this point.

[Figure]

Figure A5. Averaged spectra and residuals for 15-30 and 30-60 cm, and weighted average and weighted mean of the residual for 15-60 cm in 1993 ORG samples.

**What information is provided via subtraction of the conventional plus cover crop from the organic spectra?**

Response: This subtraction allows us to compare how the C signature in soils under cover crops with mineral fertilizer compare to cover crops with compost. As noted in the right-hand figure below, there is not much difference between the ORG and CONV+WCC spectra at 60-100 cm. This suggests that the difference in the aromatic or

carboxylate C content of these two treatments at 60-100 cm is tiny in relation to the ORG-CONV or (CONV+WCC)-CONV comparison.

[Figure]

**I am unfamiliar with this subtraction analysis, so I am curious, what information is revealed from subtraction as the reflectance intensity does not represent quantity, but rather soil chemical signature?**

Response: While FTIR is not a strictly quantitative tool for identifying specific compounds in mixed samples, it can be used pseudo-quantitatively due to the fact that the absorption of IR light by a specific molecular bond at a specific electromagnetic frequency follows the Beer-Lambert Law (Beer's Law) (e.g., Margenot et al 2016, Smith 2001). Therefore, the height and area of a spectral peak are proportional to the abundance of molecules in a sample (linear relationship), and comparing the presence and absence of peaks and the relative differences in spectral contributions from each peak in a subtraction can suggest differences in C chemistry.

Previous studies with DRIFTS in both the near-infrared (Dalal and Henry, 1986) and mid-infrared regions (Demyan et al., 2012; Margenot et al., 2015; West et al., 2020; Deiss et al., 2021) have shown direct associations between soil organic carbon concentration and absorbance at specific frequencies (depicted as peak height or area of single peaks or peak ratios). The non-linearity of concentration and absorbance arising from DRIFT can be partially corrected using the Kubelka-Munk (KM) function, which we will include alongside the uncorrected spectra in the manuscript, since a KM correction does not fully correct spectra (e.g. Clark and Roush, 1984) .

Assuming that band assignments are correct (they are consistent with literature), the presence of a peak in the subtraction at 1666 cm-1 suggests a difference in aromatic C in the surface horizons, especially for the ORG treatment. We will also update the

manuscript to provide peak intensity ratios of aromatic to carboxyl moieties [v(C=C):vas(COO-) (1662 cm$^{-1}$:1631cm$^{-1}$)] as an additional way of showing the relative difference within each subtraction.

1. Clark, R.N., and Roush, T.L., 1984. Reflectance spectroscopy: Quantitative analysis techniques for remote sensing applications, *J. Geophys. Res.*, 89( B7), 6329– 6340, doi:10.1029/JB089iB07p06329.
2. Dalal, R.C. and Henry, R.J., 1986. Simultaneous Determination of Moisture, Organic Carbon, and Total Nitrogen by Near Infrared Reflectance Spectrophotometry. Soil Science Society of America Journal, 50: 120-123. https://doi.org/10.2136/sssaj1986.03615995005000010023x
3. Deiss, L., Sall, A., Demyan, M.S., Culman, S.W., 2021. Does crop rotation affect soil organic matter stratification in tillage systems. Soil & Tillage Research 209, 104932.
4. Demyan, M.S., Rasche, F., Schulz, E., Breulmann, M., Müller, T., Cadisch, G., 2012. Use of specific peaks obtained by diffuse reflectance Fourier transform mid-infrared spectroscopy to study the composition of organic matter in a Haplic Chernozem. Eur. J. Soil Sci. 63, 189–199. doi:10.1111/j.1365-2389.2011.01420.x
5. Margenot, A.J., Calderón, F.J., Bowles, T.M., Parikh, S.J., Jackson, L.E.. 2015. Soil organic matter functional group composition in relation to organic carbon, nitrogen, and phosphorus fractions in organically managed tomato fields *Soil Science Society of America Journal* 79:772-782.
6. Margenot, A.J., F.J. Calderon, and S.J. Parikh. 2016. Limitations and potential of spectral subtractions in Fourier-transform infrared spectroscopy of soil samples. Soil Sci. Soc. Am. J. 80:10-26.
7. Smith, B.C., 2011. Fundamentals of Fourier Transform Infrared Spectroscopy. Second Edition. CRC Press, Taylor and Francis Group.
8. West, J.R., Cates, A.M., Ruark, M.D., Deiss, L., Whitman, T., Rui, Y., 2020. Winter rye does not increase microbial necromass contributions to soil organic carbon in continuous corn silage in North Central US. Soil Biol. Biochem. 107899. doi:10.1016/j.soilbio.2020.107899

**2.9 Can the authors please describe the details of the ANOVA. Was this a mixed effect model accounting for the block design? Was there an effect of block? (That difference would be interesting to see due to the two soil types). It would be helpful if the authors state that they checked normality of the data prior to ANOVA.**

Response: We will include this missing detail in the methods section. We first used a mixed effect model with block as a random effect. Since the block was not significant for any of the variables measured, we removed it from the model. We also checked normality and assumptions of the linear model prior to ANOVA. Due to the limited number of samples we included all measurements in our statistical tests, including potential outliers.

**If variability was high for certain metrics (hydraulic conductivity), it seems there may be some outliers, how were those assessed?**

In the case of the hydraulic conductivity measurements, these outliers potentially represent the contributions of macropores. It is well recognized that hydraulic conductivity measurements can vary widely across fields and landscapes (Rahmati et al. 2018) and often do not account for the presence of macropores (Brooks et al. 2004). Ksat values can span 3 orders of magnitude within a short distance (Øygarden et al. 1997). Thus, although our measurements do not reveal statistically significant differences between the treatments, the scattered high-permeability zones in the cover-crop treatments represented by the outliers are likely to contribute to rapid moisture redistribution and may play an important role in explaining the elevated deep moisture.

1. Brooks, E. S., Boll, J. and McDaniel, P. A.: A hillslope-scale experiment to measure lateral saturated hydraulic conductivity, Water Resour. Res., 40(4), 4208, https://doi.org/10.1029/2003WR002858, 2004.
2. Øygarden, L., Kværner, J. and Jenssen, P. D.: Soil erosion via preferential flow to drainage systems in clay soils, Geoderma, 76(1–2), 65–86, https://doi.org/10.1016/S0016-7061(96)00099-7, 1997.

3.  Rahmati, M., Weihermüller, L., Vanderborght, J., Pachepsky, Y. A., Mao, L., Sadeghi, S. H., Moosavi, N., Kheirfam, H., Montzka, C., Van Looy, K., Toth, B., Hazbavi, Z., Al Yamani, W., Albalasmeh, A. A., Alghzawi, M. Z., Angulo-Jaramillo, R., Antonino, A. C. D., Arampatzis, G., Armindo, R. A., Asadi, H., Bamutaze, Y., Batlle-Aguilar, J., Béchet, B., Becker, F., Blöschl, G., Bohne, K., Braud, I., Castellano, C., Cerdà, A., Chalhoub, M., Cichota, R., Císlerová, M., Clothier, B., Coquet, Y., Cornelis, W., Corradini, C., Coutinho, A. P., De Oliveira, M. B., De Macedo, J. R., Durães, M. F., Emami, H., Eskandari, I., Farajnia, A., Flammini, A., Fodor, N., Gharaibeh, M., Ghavimipanah, M. H., Ghezzehei, T. A., Giertz, S., Hatzigiannakis, E. G., Horn, R., Jiménez, J. J., Jacques, D., Keesstra, S. D., Kelishadi, H., Kiani-Harchegani, M., Kouselou, M., Jha, M. K., Lassabatere, L., Li, X., Liebig, M. A., Lichner, L., López, M. V., Machiwal, D., Mallants, D., Mallmann, M. S., De Oliveira Marques, J. D., Marshall, M. R., Mertens, J., Meunier, F., Mohammadi, M. H., Mohanty, B. P., Pulido-Moncada, M., Montenegro, S., Morbidelli, R., Moret-Fernández, D., Moosavi, A. A., Mosaddeghi, M. R., Mousavi, S. B., Mozaffari, H., Nabiollahi, K., Neyshabouri, M. R., Ottoni, M. V., Ottoni Filho, T. B., Pahlavan-Rad, M. R., Panagopoulos, A., Peth, S., Peyneau, P. E., Picciafuoco, T., Poesen, J., Pulido, M., Reinert, D. J., Reinsch, S., Rezaei, M., Roberts, F. P., Robinson, D., Rodrigo-Comino, J., Rotunno Filho, O. C., Saito, T., et al.: Development and analysis of the Soil Water Infiltration Global database, Earth Syst. Sci. Data, 10(3), 1237–1263, https://doi.org/10.5194/essd-10-1237-2018, 2018.

**The lack of differences in the field may simply be due to low power with only three field replicates. Rather than splitting the data by depth to do comparisons between treatments, can the authors run an analysis that accounts for autocorrelation over depth? On that same note, do authors need to account for repeated measures across sampling dates in 2018-2019 and within the historical data?**

Response: We will attempt to do the suggested autocorrelation analyses. One note is that for comparisons of the historical data, we only used the harvest time point in 2018 (TP3) and the harvest time point in 1993 to avoid potential issues with seasonal variation and repeated measures.

**I appreciate access to the data and code used for this analysis. Thank you for supporting transparency in data analysis.**

Response: Thank you!

**3.1 The cumulative inputs over 25 years are useful, but would be more comparable to other studies if averaged per year. This data also may be well suited for a table including all C inputs and nutrient inputs over the 25 year period (transform Fig 1 to Table 1 using Mg/ha/yr). Perhaps with the level of detail from the Century Experiment on all organic inputs, the statistical analysis could incorporate the treatments as continuous variables (amount of mineral/organic N input) rather than categorical variables?**

Response: Thanks for this idea. We will convert this graph to a table, and re-run the analyses using the amount of C and N added as a continuous variable.

**L227 If a result is non-significant, than I would remove any interpretation of 'increase'.**

Response: We will remove any interpretation of "increase", but will still include the graphs in our results and discussion sections, as we believe they are indicative of an overall trend in the data.

**Fig 2. Extremely clear pattern here. Can the significant differences be noted in some way on the figure? I would remove the lines between the points, as there are no actual measurements there, and the trends are obvious.**

Response:We appreciate this comment but believe that having lines linking data points helps visualize the trends and makes interpreting the graph more intuitive than just having the points. We have included sample plots for nutrient stocks at 60-100 cm to illustrate this point: We will add indications of significance at p<0.05 to the figure in the updated manuscript.

[Figure]

Sample Plot 1. Nutrient stocks from 60-100 cm displayed using only means and standard error bars

[Figure]

Sample Plot 2. Nutrient stocks from 60-100 cm displayed using means and error bars, with lines connecting points.

**Fig. 4 I would change the layout of this figure. You can zoom in on the y-axis and add precipitation and irrigation events. Otherwise, a simple average across the time and bar graph or box plot would tell the story more clearly, since the statistical analysis was not over time.**

Response: We will make the suggested changes to the y-axis.

**L270 Why do authors state  "largest seasonal variation" in nutrient data was in June, when only mineral N and DOC were highest in June? S and P were higher in August.**

Response: The statement referenced above is

"Nutrient values showed large seasonal variation, with the highest levels of carbon, nitrogen and sulfur observed during the June timepoint. DOC, mineral nitrogen, and sulfur values were lowest during the winter (Nov - Feb), which coincided with the period of highest rainfall. Phosphorus levels increased slightly throughout the 2018-2019 year."

We will correct  this statement to state that DOC and  mineral N were highest in June, whereas S was slightly higher in August. P was highest in Feb 2019.

**3.7 Authors must introduce microbial stress indicators earlier in the introduction and hypothesis. How does this relate to stoichiometry and soil C stability?**

Response: We will include a description/importance of stress indicators in the introduction and hypotheses. PLFA stress ratios are indicators of limited carbon and water availability to microbes. They represent an overall shift away from the thinner, more permeable cell membranes associated with Gram - bacteria and monounsaturated fatty acids towards the more tightly packed, less permeable cell membranes associated with Gram+ bacteria and saturated fatty acids (Silhavy et. al 2010).

These stress ratios are also connected to the availability of soil C. An increase in the Gram+:Gram- ratio has been associated with a decrease in easily available water and carbon (Fanin et. al 2019, Fierer et. al 2003, Bossio et. al 1998), while an increase in the saturated: unsaturated ratio and cy17:pre ratios are associated with dehydrated conditions (Moore-Kucera et. al 2007).

We have cited these stress indicators as supporting the idea of increased soluble carbon and water content in ORG systems, but have avoided making predictions about what they mean for carbon stability or how they respond to changes in nutrient stoichiometry due to inconsistencies in the literature. Making functional predictions about microbial carbon usage using PLFA are difficult, as many PLFAs are not specific to distinct species and instead distributed across taxa (Ruess et. al 2010). Additionally, the relationship between nutrient stoichiometry and PLFA biomass and ratios is inconsistent, and changes over time (Huajun et. al 2016, Ng et. al 2014).

1. Bossio, D. A. and Scow, K. M.: Impacts of carbon and flooding on soil microbial communities: Phospholipid fatty acid profiles and substrate utilization patterns, Microb. Ecol., 35(3), 265–278, https://doi.org/10.1007/s002489900082, 1998.
2. Fanin, N., Kardol, P., Farrell, M., Nilsson, M. C., Gundale, M. J. and Wardle, D. A.: The ratio of Gram-positive to Gram-negative bacterial PLFA markers as an indicator of carbon availability in organic soils, Soil Biol. Biochem., 128, 111–114, https://doi.org/10.1016/j.soilbio.2018.10.010, 2019.
3. Fierer, N., Schimel, J. P. and Holden, P. A.: Variations in microbial community composition through two soil depth profiles, Soil Biol. Biochem., 35(1), 167–176, https://doi.org/10.1016/S0038-0717(02)00251-1, 2003.
4. Moore-Kucera, J. and Dick, R. P.: PLFA profiling of microbial community structure and seasonal shifts in soils of a Douglas-fir chronosequence, Microb. Ecol., 55(3), 500–511, https://doi.org/10.1007/s00248-007-9295-1, 2008.
5. Ng, E. L., Patti, A. F., Rose, M. T., Schefe, C. R., Wilkinson, K. and Cavagnaro, T. R.: Functional stoichiometry of soil microbial communities after amendment with stabilised organic matter, Soil Biol. Biochem., 76, 170–178, https://doi.org/10.1016/j.soilbio.2014.05.016, 2014.
6. Ruess, L. and Chamberlain, P. M.: The fat that matters: Soil food web analysis using fatty acids and their carbon stable isotope signature, Soil Biol. Biochem., 42(11), 1898–1910, https://doi.org/10.1016/j.soilbio.2010.07.020, 2010.
7. Silhavy, T. J., Kahne, D. and Walker, S.: The bacterial cell envelope., Cold Spring Harb. Perspect. Biol., 2(5), https://doi.org/10.1101/cshperspect.a000414, 2010.
8. Huajun, Y., Phillips, R. P., Liang, R., Xu, Z. and Liu, Q.: Resource stoichiometry mediates soil C loss and nutrient transformations in forest soils, Appl. Soil Ecol., 108, 248–257, https://doi.org/10.1016/j.apsoil.2016.09.001, 2016.

**Discussion:**

**Authors list the key finding of increased SOC and then write what I perceive as the hypothesis of the paper: "that high concentrations of mobile C and essential nutrients for microbial activity provided by the compost, combined with the easier movement of water downward associated with a history of cover-cropping, helped transport the material needed to build C in the subsurface." Having this in the introduction will help to set up the statistical analysis, results, and discussion. However, this hypothesis was not supported by the aggregation data or the hydraulic conductivity data.**

Response: We agree with the reviewer that including this hypothesis in the introduction will better set up the rest of the paper, and will do so in the revised manuscript.

We will adjust our discussion to reference previous data from Russell Ranch that supports the idea of increased infiltration under cover crops. Work carried out at Russell Ranch over the last 30+ years has shown that the cover crop mix of hairy vetch, faba beans and oats used in this study increased infiltration and DOC input into the soil profile (Mailapalli et al., 2012). It also increased soil moisture-holding capacity during saturated conditions (Joyce et al. 2002) with no significant differences in bulk density (Colla et al. 2000) and reduced soil surface strength (Folorunso et al. 1992).

Proposed mechanisms behind the increased infiltration and moisture storage include increased aggregation, reduced crusting, and increased macroporosity. It is important to note that these root-induced soil alterations are highly localized and dependent on the root architecture of the cover crops. Specifically, cover crops with prominent tap roots (faba bean) are effective at creating continuous bio-pores, while fibrous roots (oat and hairy vetch) are effective at forming aggregates (Oglive et al., 2021). Therefore, the mixture of cover crops planted at the site likely resulted in widely variable aggregation and pore connectivity effects.

In the context of these previous observations, we attribute the increased variability of our hydraulic conductivity measurements and the increased moisture storage observed in ORG and CONV+WCC plots to the presence of WCCs.  Our inference of greater soluble carbon

input to subsoil layers under compost application is also supported by the multiple lines of evidence that we present in the manuscript:

1) More soluble C in ORG subsoils
2) Observations of higher water infiltration and storage under cover crops
3) Greater amounts of soluble organic carbon in compost, and
4) Reduced subsoil microbial stress indicators under ORG systems (attributed to more C availability)

1) Colla, G., Mitchell, J. P., Joyce, B. A., Huyck, L. M., Wallender, W. W., Temple, S. R., Hsiao, T. C. and Poudel, D. D.: Soil physical properties and tomato yield and quality in alternative cropping systems, in Agronomy Journal, vol. 92, pp. 924–932, American Society of Agronomy, https://doi.org/10.2134/agronj2000.925924x, , 2000.
2) Folorunso, O., Rolston, D., Prichard, P. and Louie, D.: Cover crops lower soil surface strength, may improve soil permeability, Calif. Agric., 46(6), 26–27 //calag.ucanr.edu/archive/?article=ca.v046n06p26, last access: 6 May 2021, 1992.
3) Joyce, B. A., Wallender, W. W., Mitchell, J. P., Huyck, L. M., Temple, S. R., Brostrom, P. N. and Hsiao, T. C.: INFILTRATION AND SOIL WATER STORAGE UNDER WINTER COVER CROPPING IN CALIFORNIA'S SACRAMENTO VALLEY, Trans. ASAE, 45(2), 315–326, 2002.
4) Mailapalli, D. R., Horwath, W. R., Wallender, W. W. and Burger, M.: Infiltration, Runoff, and Export of Dissolved Organic Carbon from Furrow-Irrigated Forage Fields under Cover Crop and No-Till Management in the Arid Climate of California, J. Irrig. Drain. Eng., 138(1), 35–42, https://doi.org/10.1061/(asce)ir.1943-4774.0000385, 2012.
5) Ogilvie,C.M.;Ashiq,W.; Vasava, H.B.; Biswas, A. Quantifying Root-Soil Interactions in Cover Crop Systems: A Review. Agriculture 2021, 11, 218. https://doi.org/10.3390/ agriculture11030218

**Please go into more detail on how no differences in aggregation "rule out" increased pore space as the increase in water content. What is the alternative explanation? Is this just an issue with statistical power?**

Response: We agree that "rule out" was not the best choice and will reword that sentence - We believe that the increase in moisture content is not due to an increase in pore space from increased aggregation. Instead, we make the argument that the increase in moisture content was due to an increase in root-related macropores.

**L335 This also seems like a great candidate sentence for another hypothesis: "Due to the fact that tillage in all systems would likely eliminate differences among them in the top 30 cm, we would expect any differences in macroporosity and infiltration among treatments to be most affected by those roots that extend below the 30 cm plow layer". This is the first mention of tillage depth. Please specify the depth of disking in the methods, and if this was applied to the conventional fields as well.**

Response: Tillage is part of the management of all three farming systems sampled in the manuscript. We have avoided hypotheses related to tillage in the manuscript as the difference in the amount of tillage between our experimental systems is not very large (an additional 1-2 passes/year in CONV+WCC and ORG plots to incorporate cover crops). Additionally, the historical data does not include the amount of tillage per plot. The quoted sentence was intended to highlight that all of the WCC mix used in the RR plots have roots that extend deeper than 30 cm.

We now include disking depth in the tables added to the Methods section.

**L340-345 This paragraph on cascade theory describes why FTIR analysis was necessary. This also should be included, or at least alluded to, in the introduction. This is a really interesting discussion (L350-355), and could also be a good place to bring up the variability in the conductivity data.**

Response: We will include a short introduction to the cascade theory in the introduction.

**L371 Figure referenced should be Fig 9.**

Response: We will correct this figure reference.

**L375 Support with values from the results. The nutrient values may all be better represented by tables, although the graphs show dynamics across the season, I would argue that depth, not season, is the key factor in this analysis.**

Response: We will include rough CNPS ratios in our discussion to support this point. We did not include actual nutrient ratios in our discussion as the reference nutrient stoichiometry values reported in Richardson et.al 2014 were calculated using total nutrient values (digestion) as opposed to the available nutrient values used in this paper. We will include this caveat in our discussion.

We agree that depth is a more important factor than seasonal variation in this manuscript. However, EOC in Mediterranean systems has been shown to vary greatly across the course of a single year (Steenwerth et. al 2008), and showing data across multiple timepoints demonstrates that our observation of increased EOC in ORG systems is not constrained to a single sampling date or depth. This is not true for mineral N and S however, which indicates that future projects involving available N or S in this system should carefully select timepoints for measurement if they mean to compare results to the literature.

We will include a discussion of the importance of seasonal variation in our dataset, and highlight our reasoning behind sampling multiple times throughout the year in the manuscript revision.

1. Richardson, A. E., Kirkby, C. A., Banerjee, S. and Kirkegaard, J. A.: The inorganic nutrient cost of building soil carbon, Carbon Manag., 5(3), 265–268, https://doi.org/10.1080/17583004.2014.923226, 2014.
2. Steenwerth, K. and Belina, K. M.: Cover crops enhance soil organic matter, carbon dynamics and microbiological function in a vineyard agroecosystem, Appl. Soil Ecol., 40(2), 359–369, https://doi.org/10.1016/j.apsoil.2008.06.006, 2008.

**L380 Consider rewriting this section title, as there was no direct comparison to a compost treatment alone.**

Response: We agree and will reword this section title, and restructure the manuscript by including a clear statement in the methods and introduction that there is no compost-only treatment, and make it clear that we cannot compare the effects of compost alone to the effects of cover crops alone due to the experimental design. We will also include this in our discussion of the limitations of the data.

**L382 Is the microbial processing near the surface based on the FTIR data? Please reference.**

Response: The conclusion of increased microbial processing near the surface in ORG systems was based on our observations of increased microbial biomass in 2018 as well as increased carboxylate carbon in ORG systems at 0-15 cm from 1993-2018. This increase in carboxylate carbon from 1993-2018 was attributed to microbial oxidation of carbon inputs. We will reference both our biomass and FTIR data in this section (Figure 8a, 9a).

**L388-L390 This paragraph seems speculative. Please input FTIR data that supports these ideas (C chemistry from this dataset).**

Response: This paragraph is meant to present hypotheses and will be reworded as such. This hypothesis was based on observations of similar EOC levels in CONV vs CONV+WCC, even though the CONV+WCC Gram+:Gram- ratio was lower than in CONV. Cover crops have been shown to increase DOC inputs in the literature (Steenwerth et. al 2008), and so a lower stress ratio in CONV+WCC could suggest a small trickle of DOC, enough to promote priming, but not enough to increase stocks as they may have done in the ORG system. We will include citations of our results and further literature to provide more evidence for our hypothesis of why there was no increase in C in the CONV+WCC subsoils.

1.  Steenwerth, K. and Belina, K. M.: Cover crops enhance soil organic matter, carbon dynamics and microbiological function in a vineyard agroecosystem, Appl. Soil Ecol., 40(2), 359–369, https://doi.org/10.1016/j.apsoil.2008.06.006, 2008.

**L388 What does "high variability of soil C measurements" refer to? Dry combustion measurements of total C are very consistent.**

Response: We were referring to the variability in C measurements across a field, and during a growing season. Total C measurements in upper soil layers can vary by as much as 8% as part of normal seasonal variation (Wuest 2014), and detecting even large changes in subsoil SOC stocks can require a prohibitive number of samples (Kravchenko and Robertson 2011) due to the inherent spatial variability of subsoil SOC.

1.  Kravchenko, A. N. and Robertson, G. P.: Whole-Profile Soil Carbon Stocks: The Danger of Assuming Too Much from Analyses of Too Little, Soil Sci. Soc. Am. J., 75(1), 235–240, https://doi.org/10.2136/sssaj2010.0076, 2011.
2.  Wuest, S.: Seasonal Variation in Soil Organic Carbon, Soil Sci. Soc. Am. J., 78(4), 1442–1447, https://doi.org/10.2136/sssaj2013.10.0447, 2014.

**Conclusion:**

**L406-407: "This was facilitated by increased soil macropores created by cover crop roots leading to higher rates of transport of soluble C". Macropores were not analyzed in this study, and no increases were found in hydraulic conductivity or aggregation, please clearly delineate quantified results versus hypotheses in this conclusion.**

Response: We will reword the conclusion to reflect the data that we directly measured and also to better reflect the changes we have made to the paper - this will include acknowledgement of the limitations in our data, and drawing on previous research at Russell Ranch. We will more clearly delineate statements that are quantified results vs. inferences drawn from those results.

We would like to thank the reviewer for their detailed, helpful comments, and hope that we have addressed their concerns in our response.

---

## Author Response (AR1)

**Reviewer 1 Comments**

**In this manuscript, the authors present a compelling hypothesis of how compost in combination with winter cover crops can lead to accumulation of aromatic-rich subsurface soil carbon. The hypothesis is complex but plausible whereby cover crop roots improve soil structure/porosity facilitating greater transport of soluble C and nutrients derived from the compost directly to the subsurface where this C can be stabilized. While the hypothesis is compelling, unfortunately, I do not think the authors have collected the right data to test this hypothesis.**

Response: We would like to again thank the reviewer for their feedback. We have addressed their concerns by making clearer what evidence we have for the different components of the hypothesis, better clarifying where there are  limitations of the data, and drawing more on existing literature to support our synthesis as described below.

**Utilizing a long-term field trial should be a great way of trying to address this hypothesis. However, a major limitation of the study is that there is no compost-only treatment, so there is no way to separate the effect of compost alone from the interactive effect of compost and cover crops together. There is nothing the authors can do about this except recognize this as a limitation of the study design.**

Response: We have added a sentence at the end of section 2.1 clarifying that there is no compost-only treatment, and restructured the titles of the various discussion sections to remove all references to the impact of compost alone.

**A major feature of the author's hypothesis is that cover crop roots have created greater porosity that facilitates greater water flow down the soil profile. The data simply do not support this notion. The authors find no difference in saturated hydrologic conductivity at 35 cm (although there was a trend for much greater variability in the compost + cover crop treatment) and no difference in soil aggregates across treatments.**

Response: We have extensively edited section 4.1 to include data from previous studies at Russell Ranch that support our hypothesis of increased water storage and movement in cover cropped plots. We have ensured that any discussion of increased porosity in the manuscript is clear that this is an inference, and supported this inference with previous literature where appropriate.

**The only significant difference was greater water content in the two treatments with cover crops but the authors did not measure bulk density in the 2018 samples and they did not measure porosity so it is difficult to come up with an explanation for this observation.**

Response: We have included a discussion of several previous studies at Russell Ranch that showed little to no change in bulk density, increased infiltration and increased moisture holding capacity under winter cover crops. We have also included a discussion

of the impact of small Ksat sample size, cover crop root type, and the lack of support for other potential hypotheses for increased moisture content.

**The next major component of the hypothesis is that compost leads to greater soluble C and N. The authors use salt-extractions of soil samples at four time points during the 2018 season to generate supporting data. Salt-extractable C is an interesting carbon pool (a potentially soluble pool of C) but there is ample evidence that this lab-extracted pool has little relationship to DOC when collected in lysimeters in the field. Without direct collection of DOC diffusing and advecting down the soil profile it is difficult to say whether the differences in the extractable pools are actually leading to more DOC flux to the subsoil under compost addition.**

Response: We have changed all references of DOC in the manuscript to EOC, and provided support for our decision to use EOC instead of DOC via lysimeter in section 2.5. We have also noted that measurements of EOC are common in the literature, and though they are not exactly equivalent to DOC, EOC measurements can still be used to draw inferences about the presence and movement of soluble carbon.

We have supported our conclusion of increased soluble C flux to subsoils in the ORG systems by highlighting our multiple lines of evidence:

1) More EOC in ORG subsoils
2) Observations of higher water storage, and potentially increased infiltration under cover crops
3) Greater amounts of soluble organic carbon in compost, and
4) Reduced subsoil microbial stress indicators under ORG systems (attributed to more soluble C and nutrient availability)

**The third component of the hypothesis relates to the preferential partitioning of DOC chemistry down the soil profile. The evidence here is particularly weak. Mid infrared FTIR spectroscopy is not a quantitative analytical tool for determining abundance of specific compounds. If it were, labs wouldn't spend millions of dollars on more precise equipment. FTIR spectroscopy is good for identifying compounds in simple mixtures but not for quantifying their abundance in simple or complex mixtures (and soil is one of the most complex there is). Peak features depending on if they are due to vibrations, wiggles, combinations or overtones all have different relationships between abundance of the specific bonding environment and absorption – basically, you would have to prove that there is a linear relationship between "aromatics" and those two peak features in order to do a spectral subtraction and have any confidence that the difference spectrum represents real differences in chemistry.**

Response: We have modified the introduction, results and methods (section 2.8) sections to support our approach using FTIR spectral subtractions for pseudo quantification of SOM functional groups. We have replaced the previous Figure 8A and

B with a Kubelka-Munk corrected spectra on a common y-axis to correct for non-linearity of concentration and absorbance in our spectra.

We have also included a table of peak intensity ratios of aromatic to carboxyl moieties [$v(C=C):vas(COO^-)$ (1662 cm$^{-1}$:1631cm$^{-1}$)] to support our observations of a change in SOM composition over systems and depth.

**I also find it problematic that all treatments have showed the same increase in carboxylate functional groups over 25 years – wouldn't we expect the conventional treatment to be more or less at steady state, so we shouldn't see the same changes as seen in the cover crop and compost + cover crop treatments?**

Response: When shown with the KM correction, the spectral contributions for the change in carboxyl groups in CONV are no longer similar to those in CONV+WCC and ORG systems.

**Lastly, what is the actual magnitude of the "increase" in aromatic features in the compost treatment over the conventional treatment? There are no units on the y-axis. The authors have replicates so they could run statistics to see if this increase was significant.**

Response: We have included figure A6 to highlight the very small amount of variation between spectra that were averaged for subtraction. We have also included a section for the caption of figure 8A to highlight that the spectra are shown on a common y-axis scale and that they are only offset for ease of comparison.

**Finally, the microbial data is not well integrated into the hypothesis. Would lower microbial stress result in greater carbon stabilization via increased carbon-use efficiency or would it result in greater priming and potential loss of older SOM? Regardless of what microbial stress means for carbon cycling, the data were non-significant across treatments. The only significant difference was in Gram+:Gram- ratio but the ecological significance of this difference was not described.**

Response: We have better integrated both our explanation of the microbial stress indicators and their integration into the manuscript. We highlighted that the microbial stress ratios are primarily meant to support our inference of increased soluble C and nutrient at depth, but have refrained from using them to draw detailed conclusions about the microbial community.

**Just to reiterate, I think the hypothesis laid out here for subsoil C accumulation under compost and cover crops is entirely plausible but the evidence in this study to support the hypothesis is not particularly strong.**

Response: We again thank the reviewer for their constructive comments, and hope that the edits we have made have provided a more convincing argument.

**Reviewer 2**

**Summary:**

**This manuscript leverages data from a long-term agricultural experiment at the Russell Ranch in California and a year of more detailed measurements to explore interacting cover crop and compost effects on subsurface soil carbon dynamics. Authors blend historical measurement of carbon stocks with present day analyses of carbon (bulk C, FTIR), nutrients (Mehlich-III), soil physical properties (aggregation, moisture content, and hydraulic conductivity), and microbial biomarkers (PLFA) at four sampling dates. An ANOVA was used to assess the effect of time, depth, and management, with subsequent separate analysis of differences between management treatments at each of three depths.**

**Although the experimental design and methods are sound, there is a disconnect between the objective to assess interaction of cover crops and compost and the data analysis. The discussion ties in interesting concepts such as the 'cascade theory' and microbial stress indicators that must be brought up further into the introduction to create a threat throughout the paper.**

RESPONSE: We appreciate the reviewer's comments, and have extensively rewritten the introduction, results and discussion to take advantage of their comments.

**Below please find my recommendations to reframe the paper and utilize historic data, specific questions and a few line edits for authors. Most edits occur in the first half of the paper, which may help to connect the methods and results into the compelling discussion.**

**Title: I recommend making this more specific. The final sentence of the discussion states that "care should be taken when applying these results to different soil types and climates"; therefore, adding the soil type or climate (or both) into the title seems prudent.**

RESPONSE: We have changed the title to "Synergy between compost and cover crops in a Mediterranean row crop system leads to increased subsoil carbon storage"

**Abstract:**

**Throughout the paper, can authors use the treatment names as in the original experimental dataset (Wolf, 2018 page 6): CONV = CMT conventional maize-tomato, ORG=OMT organic maize-tomato, and (page 5) WCC – winter cover crop? I understand that Tautges and Chiartas 2019 used the CONV, ORG notation, but a brief explanation would be helpful.**

RESPONSE: We have included a short sentence in section 2.1 to highlight the difference in treatment names.

**The theory of cover crops providing a macropore system for transport of DOC is interesting, but the data do not support this theory (no measurement of porosity, change in bulk density, or changes in soil hydraulic properties). It is appropriate for a discussion, but I might exclude this as a main finding from the abstract.**

RESPONSE: We have removed this inference from the abstract and replaced it with a more general statement.

**Introduction:**

**I appreciate that the abstract and introduction mention soil health, but there is no clear definition or explanation of its importance to the paper. Either simply remove this term and focus solely on soil carbon and microbial processes, or please directly connect soil health and often associated shallow sampling regimes to this "outsized perceived role in ecosystem services".**

Response: We have removed any mention of soil health from the manuscript and clarified that surface soils should not be used to answer questions about the entire soil profile.

**This is a good argument and dataset to support deeper sampling. Authors may also include references summarized by Mobley et al 2015 in their article "Surficial gains and subsoil losses of soil carbon and nitrogen during secondary forest development": Post & Kwon, 2000; West & Post, 2002 review 360 articles on land use change, with only 10% sampling below 30cm.**

Response: We have included the Mobley reference, as well as including several other references on depth of soil sampling.

**In this paragraph, please clarify, at what depth are the authors designating topsoil v subsoil for this study?**

Response: We have included a paragraph discussing our decision to label 0-15 as surface soil, 15-60 as a transition zone, and 60-100 cm as subsoil.

**This first paragraph of the introduction discusses "longer C residence times" of deep soil C, which requires further explanation.**

Response: We have included a sentence in the introduction highlighting that subsoil C can be as old as $10^3$-$10^4$ years, as opposed to younger surface C.

**Overall, the introduction structure can be strengthened by clarifying topic sentences (e.g., specify cover crops L51) and adding updated references. Can you support the Jenny citation with more modern references, even Brady and Weill Nature and Properties of Soils, or USDA technical information "Designations for Horizons and Layers" in Soil Survey Manual – Ch 3 (https://www.nrcs.usda.gov/wps/portal/nrcs/detail/soils/ref/?cid=nrcs142p2_054253#designations).**

Response: We have added the suggested updated references, and clarified topic sentences throughout the introduction and discussion.

**The introduction structure may flow better using paragraphs separated into chemical, physical and biological controls or layered as (1) depth; (2) chemistry of C inputs and stabilization at depth; (3) management impacts at depth – specifically cover crops; and (4) management interaction with other factors (microbial).**

Response: We restructured the introduction into sections highlighting

1) Importance of deeper soils

2) Definition of subsoils

3) Depth and C chemistry

4) Depth and microbial C processing

5) Support for methods used

6) Depth and the impact of agricultural practices

**The introduction touches upon stoichiometry, a critical highly manipulated factor in managed conventional systems that effects soil C storage. To go further in depth on soil chemistry (e.g., at L40), authors can address changes over time in stoichiometric constraints on decomposition (e.g., see Soong et al 2019 "Microbial carbon limitation: The need for integrating microorganisms into our understanding of ecosystem carbon cycling").**

Response: We have included the reference and inserted a short discussion of the importance of C and nutrient stoichiometry in microbial biomass formation into the discussion.

**Also, authors can mention higher physical disturbances in surface soils (L55), and the types of management associated with cover crops, such as crimping/rolling.**

Response: We have included additional disturbance with cover crops as a potential impact in the introduction.

**Please also include specific soil type, climate and cropping system when comparing to other studies, otherwise direct comparisons are not particularly informative.**

Response: We have added data about soil type and cropping system when making direct comparisons (section 4.4), and have clarified the cover crop mix used when comparing results from studies with cover crops.

**Can also cite McClelland et al 2020 "Management of cover crops in temperate climates influences soil organic carbon stocks: a metaanalysis" that analyzed soils only down to 30cm.**

Response: We did not include this citation, but instead added Singh et. al (2021) to better focus on cover crops and lysimeter measurements.

Singh, G., Kaur, G., Williard, K. W. J., & Schoonover, J. E. (2021). Cover crops and tillage effects on carbon–nitrogen pools: A lysimeter study. Vadose Zone Journal, 20(2), e20110. https://doi.org/10.1002/VZJ2.20110

**As for the sampling strategy by depth, can the authors please describe why they separated out into these depths 0-15, "intervening", and the subsurface as 60-100cm? How do these depths compare to the horizons in these two soils? (Looking up the series descriptions Yolo has A horizons down to 66cm and then C horizons, and Rincon has A down to 20, B 20-100cm. Should the analysis be completed on A and B horizons rather than depth profiles?) How do these depths relate to roots of corn (100cm+), tomato (60cm+) and cover crops (variable)?**

**Can authors please justify why 15-60cm is combined into a single sample in 2018, when historical data had an additional delineation? (Is it simply limited time/costs or another reason?)**

Response: We have included a paragraph in the introduction highlighting our reasoning behind using the depth intervals we have chosen (previous RR work, lack of horizonation).

**Please stay consistent with the terms "subsoil" versus "subsurface soil", as depth is a major component of this study.**

Response: We have removed all reference to subsurface soil in the manuscript and replaced them with subsoil, except when referencing subsurface drip and subsurface flow.

**The overarching question and hypothesis require further editing to clearly lead into the results and discussion. There seems to be a disconnect between the main question and the methods of this paper. The main question includes "carbon formation" (does that mean microbially processed C? or stabilizedC formation?) and "storage processes" (that obviously includes aggregation, but the carbon content of these size classes was not measured). Also, what is meant by the term "SOC-related indicators", does that mean SOC stability or reactivity-related indicators? As written the hypotheses are just predictions, there is no description as to the mechanisms behind the described expected results.**

Response: We have reworded the hypothesis and removed references to carbon formation and stabilization processes, as we do not measure any processes in the manuscript.

**An interesting hypothesis arises in the discussion around cascade theory, can authors pull that into the introduction? This can provide a way to integrate the study of carbon chemistry (FTIR) and microbial biomarkers that otherwise are not included in the hypotheses.**

Response: We have introduced the cascade theory in the introduction section.

**Finally, I agree with the previous reviewer comment, that the treatments CONV (fertilizer), CONV+WCC (fertilizer + cover crops), and ORG (compost + cover crops) do not disentangle the effect of compost. I don't think there is there a treatment in the Century Experiment that was maize-tomato plus compost only or fertilizer + compost, but this should be mentioned as a limitation in the study, particularly in the subtraction of FTIR spectra.**

Response: We have added a sentence at the end of section 2.1 clarifying that there is no compost-only treatment, and restructured the titles of the various discussion sections to remove all references to the impact of compost alone.

**This manuscript covers many aspects of deep soil C and management, no need to emphasize the complicated factors of global change (L87) at the end of the introduction, unless those are also analyzed over time.**

Response: We have removed this sentence from the introduction.

**Materials and Methods:**

**Thank you for a concise description of the site and experiment. I recommend authors also add basic climate data such as climate type, mean annual mix and max temperature, mean annual precipitation, and also specific 2018-19 climate data for comparison.**

**Authors write that the 'horizon information' is available from Wolf et al, but I only can find soil chemistry by depth, not the soil description in that dataset (horizon delineations are online). Can authors add in the horizon depth into the methods for both the Yolo and Rincon soils, and key chemistry such as pH and texture? A table in the materials and methods section could organize all of this soil and climate information for quick reference.**

**This could also include the other key management notes that will impact DOC transport, such as the conversion from furrow to drip in 2014, as well as information from the 2018-2019 season such as crop planting/harvest dates, total irrigation amount, and the anomalous compost application in September 2019. These details can then be incorporated smoothly into the discussion.**

Response: We have included a graph of temperature and rainfall (supplementary figure A5) and a table of soil horizon variables (Supplementary Table A4). We have also

included climate and management notes in the Methods section under section 2.1 and 2.3, and in Supplementary Table A5.

**The differentiation between the sampling and analysis of the older data and 2018-2019 methods is now clearer. Thank you for the new methods section. However, without hypotheses asking seasonal questions over time – why sample at four time points in a single year? Particularly as the authors state that a single year of data is not sufficient to look at differences at depth (L81-82) to justify use of the historical data. Perhaps authors can create one or two hypotheses for the 2018-19 season, and other for the long-term effects and historical data.**

Response: We have included a paragraph referencing the seasonal variation in the data in the discussion, as well as support for sampling at multiple time points during the year in the methods section.

**The use of PLFA and FTIR is not justified from the hypotheses or introduction. The use of these techniques, particularly stress ratios for PLFA needs to be explained within a wider context in the introduction.**

Response: We have included a paragraph supporting the use of PLFA and FTIR into the introduction.

**2.7 Please clarify the statement that 9 out of 18 plots were sampled for hydraulic conductivity (those under tomato). Were half of the plots under corn and the other under tomato during this 2018 sampling? That needs to be included in the methods section. Or are you referencing the full 18 plots of all the Century experimental treatments? Finally, why are 8 dates included for soil moisture content, when soils are sampled only 4 times?**

Response: We have clarified that half the plots were under corn, and half were under tomato in the methods section, and highlighted that we only sampled the plots under tomato for hydraulic conductivity sampling. We have also corrected the methods section to state that soil moisture content was sampled 8 times.

**2.8 I have some concern over the use of averaging and subtraction of the spectra. What was the variance between the historic soils of 15-30 and 30-60 cm?**

Response: We have included figure A6 to highlight the very small amount of variation between spectra that were averaged for subtraction.

**What information is provided via subtraction of the conventional plus cover crop from the organic spectra? I am unfamiliar with this subtraction analysis, so I am curious, what information is revealed from subtraction as the reflectance intensity does not represent quantity, but rather soil chemical signature?**

Response: We have modified the introduction, results and methods (section 2.8) sections to support our approach using FTIR spectral subtractions for pseudo quantification of SOM functional groups. We have replaced the previous Figure 8A and

B with a Kubelka-Munk corrected spectra on a common y-axis to correct for non-linearity of concentration and absorbance in our spectra.

We have also included a table of peak intensity ratios of aromatic to carboxyl moieties [$\nu$(C=C):$\nu_{as}$(COO$^-$) (1662 cm$^{-1}$:1631cm$^{-1}$)] to support our observations of a change in SOM composition over systems and depth.

**2.9 Can the authors please describe the details of the ANOVA. Was this a mixed effect model accounting for the block design? Was there an effect of block? (That difference would be interesting to see due to the two soil types). It would be helpful if the authors state that they checked normality of the data prior to ANOVA.**

Response: We have included details about the statistical analysis in section 2.9, including our observation that there was no effect of block, and that we checked for normality.

**If variability was high for certain metrics (hydraulic conductivity), it seems there may be some outliers, how were those assessed?**

Response: We have clarified that these outliers are likely a function of the low number of Ksat cores and the effect of cover crop roots in our systems.

**The lack of differences in the field may simply be due to low power with only three field replicates. Rather than splitting the data by depth to do comparisons between treatments, can the authors run an analysis that accounts for autocorrelation over depth? On that same note, do authors need to account for repeated measures across sampling dates in 2018-2019 and within the historical data?**

Response: We were not able to run the suggested analysis due to time constraints, but believe that the results split by depth provide support for our hypothesis. When comparing the historical data and 2018-2019 samples, we only compared data from the August timepoint to avoid issues with repeated measures.

**I appreciate access to the data and code used for this analysis. Thank you for supporting transparency in data analysis.**

Response: Thank you!

**3.1 The cumulative inputs over 25 years are useful, but would be more comparable to other studies if averaged per year. This data also may be well suited for a table including all C inputs and nutrient inputs over the 25 year period (transform Fig 1 to Table 1 using Mg/ha/yr). Perhaps with the level of detail from the Century Experiment on all organic inputs, the statistical analysis could incorporate the treatments as continuous variables (amount of mineral/organic N input) rather than categorical variables?**

Response: We have kept the data as a figure, as we believe it aids in interpretation, but have included Supplementary Table A6 which lists average C and nutrient inputs per year over 25 years. We were not able to run the suggested analysis due to time constraints.

**L227 If a result is non-significant, than I would remove any interpretation of 'increase'.**

Response: Where appropriate, we have removed the term "increase", but have also highlighted datas that we believe are indicative of larger trends.

**Fig 2. Extremely clear pattern here. Can the significant differences be noted in some way on the figure? I would remove the lines between the points, as there are no actual measurements there, and the trends are obvious.**

Response: We have included significance indicators for C change for Figure 3, and kept the lines between the points for figure 2 as we believe they aid in interpretation, but modified the line widths.

**Fig. 4 I would change the layout of this figure. You can zoom in on the y-axis and add precipitation and irrigation events. Otherwise, a simple average across the time and bar graph or box plot would tell the story more clearly, since the statistical analysis was not over time.**

Response: We have zoomed in on the y-axis, and included precipitation events in figure A5. There were no irrigation events.

**L270 Why do authors state "largest seasonal variation" in nutrient data was in June, when only mineral N and DOC were highest in June? S and P were higher in August.**

Response: We have corrected this statement to state that DOC and mineral N were highest in June, whereas S was slightly higher in August.

**3.7 Authors must introduce microbial stress indicators earlier in the introduction and hypothesis. How does this relate to stoichiometry and soil C stability?**

Response: We have included a discussion of PLFA in the introduction, methods and discussion, but have refrained from relating our observations to stoichiometry and C stability as our measurements do not give clear indications for those trends.

**Discussion:**

**Authors list the key finding of increased SOC and then write what I perceive as the hypothesis of the paper: "that high concentrations of mobile C and essential nutrients for microbial activity provided by the compost, combined with the easier movement of water downward associated with a history of cover-cropping, helped**

**transport the material needed to build C in the subsurface." Having this in the introduction will help to set up the statistical analysis, results, and discussion. However, this hypothesis was not supported by the aggregation data or the hydraulic conductivity data.**

Response: We have included this hypothesis in the introduction.

We have also extensively edited section 4.1 to include data from previous studies at Russell Ranch that support our hypothesis of increased water storage and movement in cover cropped plots. We have included a discussion of several previous studies at Russell Ranch that showed little to no change in bulk density, increased infiltration and increased moisture holding capacity under winter cover crops. We have also included a discussion of the impact of small Ksat sample size, cover crop root type, and the lack of support for other potential hypotheses for increased moisture content.

Finally, we have supported our conclusion of increased soluble C flux to subsoils in the ORG systems by highlighting our multiple lines of evidence:

5) More EOC in ORG subsoils
6) Observations of higher water storage, and potentially increased infiltration under cover crops
7) Greater amounts of soluble organic carbon in compost, and
8) Reduced subsoil microbial stress indicators under ORG systems (attributed to more soluble C and nutrient availability)

**Please go into more detail on how no differences in aggregation "rule out" increased pore space as the increase in water content. What is the alternative explanation? Is this just an issue with statistical power?**

Response: We have removed this sentence and clarified that the increase in moisture content was likely due to an increase in root-related macropores.

**L335 This also seems like a great candidate sentence for another hypothesis: "Due to the fact that tillage in all systems would likely eliminate differences among them in the top 30 cm, we would expect any differences in macroporosity and infiltration among treatments to be most affected by those roots that extend below the 30 cm plow layer". This is the first mention of tillage depth. Please specify the depth of disking in the methods, and if this was applied to the conventional fields as well.**

Response: We have edited this sentence and included a description of tillage depths and number of events in table A5.

**L340-345 This paragraph on cascade theory describes why FTIR analysis was necessary. This also should be included, or at least alluded to, in the introduction. This is a really interesting discussion (L350-355), and could also be a good place to bring up the variability in the conductivity data.**

Response: We have included this paragraph in the introduction.

**L371 Figure referenced should be Fig 9.**

Response: We have corrected this reference.

**L375 Support with values from the results. The nutrient values may all be better represented by tables, although the graphs show dynamics across the season, I would argue that depth, not season, is the key factor in this analysis.**

Response: We have rewritten the referenced section. We have kept graphs showing dynamics across seasons to highlight that the increased EOC and P presence in ORG systems is not limited to a single timepoint, while mineral N and S levels are dependent on timepoint. We have also highlighted the seasonal variation in a paragraph in our discussion.

**L380 Consider rewriting this section title, as there was no direct comparison to a compost treatment alone.**

Response: We have reworded this section title to remove references to compost alone.

**L382 Is the microbial processing near the surface based on the FTIR data? Please reference.**

Response: We have clarified that our inference of increased microbial processing near the surface is based on biomass and the increased presence of more oxidized carboxylate C.

**L388-L390 This paragraph seems speculative. Please input FTIR data that supports these ideas (C chemistry from this dataset).**

Response: We have clarified that this paragraph is our attempt to lay out a hypothesis for the differences in C stocks, and inserted references to the results.

**L388 What does "high variability of soil C measurements" refer to? Dry combustion measurements of total C are very consistent.**

Response: We have removed this reference.

**Conclusion:**

**L406-407: "This was facilitated by increased soil macropores created by cover crop roots leading to higher rates of transport of soluble C". Macropores were not analyzed in this study, and no increases were found in hydraulic conductivity or aggregation, please clearly delineate quantified results versus hypotheses in this conclusion.**

Response: We have reworded the conclusion to better differentiate between our results, hypothesis and inferences.

We would like to thank the reviewer for their detailed, helpful comments, and hope that we have addressed their concerns in our response.

---

## Author Response (AR2)

Reviewer 1

Thank you for your careful revisions of the manuscript. I have two main points, followed by line edits for minor revision.

We appreciate the thoughtful and detailed comments outlined here, and have endeavored to address them appropriately.

The hypothesis does not require the analysis of microbial indicators (PLFA) or C composition data (FTIR): "We hypothesized that 170 significant increases in subsoil carbon stocks were associated with the combination of high concentrations of soluble C and nutrients from compost and increased hydraulic transport associated with cover crops." Please include each of these within the hypotheses, objectives, or research questions, otherwise these methods are not justified to be included in the manuscript. Ideally the hypotheses can link the main metrics (C composition, microbial indicators, hydraulic conductivity, nutrient and C at depth within an interactive mechanism or overarching theory - like cascade theory).

We have modified the hypothesis paragraph to better incorporate all measurements made during the experiment. The modified paragraph is below:

The goal of this study was to explore some of the potential mechanisms behind the observed differences in carbon storage in different RR management systems, and to see how these carbon stores have changed after an additional 7 years. In particular, we focus on the role of cover crops in promoting hydraulic conductivity, and how those hydraulic changes impact water, C chemistry, nutrient distribution, microbial biomass and community composition in the subsoil under the addition of additional C (compost) and N (nitrogen fertilizer). We hypothesized that the combination of cover crops and additional C input would result in large amounts of soluble C and nutrients being transported deeper via hydraulic transport, leading to more microbially processed carbon and increased carbon stocks in the subsoil. We also hypothesized that these differences are not due to seasonal variation, and that increased soluble C and nutrient stocks will be consistent at multiple timepoints throughout the year.

The first half of the discussion needs to integrate paragraphs summarizing the literature (more fit for an introduction) with the paragraphs on the current findings. Specific lines within the discussion are indicated below for these edits.

We have reorganized the beginning of each discussion section to explicitly reference the results.

Line edits:

133 Define SOM - first use in the main text here.

Edited

150 - I don't think you necessarily need the cover crop type listed in the intro, merely when you make direct comparisons within the discussion.

Removed cover crop types from the introduction.

185-193 - Experimental design can be above the hypothesis in the introduction.

Modified experimental design paragraph to better highlight the broader questions of the paper, and moved above the hypothesis section.

190 - You can add climate warming "and subsequent soil C losses" to tie this final sentence into the experiment tightly.

Included

233 - Insert space for "2 m".

Included

246 - Merely a suggestion - this information could go into a table to improve accessibility, but is fine as is.

As we already have 9 figures, we would like to avoid including another table if possible.

273 - Define EOC the first time it is mentioned.

Included EOC definition

305 - Add "For each sample 6g of soil" so that the sentence does not start with a number.

Edited

317 - Again, start with "The" before 2018 carbon and nutrient stocks…

Edited

317 - Previously 'C' and 'N' are used for carbon and nitrogen, stay consistent.

Changed all mentions of "carbon" and "nitrogen" to "C" and "N" where appropriate

325 - Is there a need to account for microaggregate-sized sand? (Simply by rinsing a subset of the microaggregate sample through the 250 sieve completely and weighing before and after dry mass.)

We have particle size analysis data for the soils used in this experiment, and they do not show significantly different sand contents between systems. We would also like to note that Wang et. al (2017) found significant differences in microaggregate size fractions in these plots, without accounting for fine sand. While accounting for microaggregate-sized sand may indeed modify the results, we anticipate that this would be a uniform effect across our samples, and would therefore not change the conclusions of the paper.

328 Delete second "PLFA analysis" in the last part of the sentence

Edited

330 Subscript on N2

Edited

334 Lowercase S on PLFAs

Edited

353 Ensure there is a citation for R here too.

Included citation

373 Stay consistent with "carbon" vs "C" - check throughout

Changed all mentions of "carbon" to "C" in the manuscript where appropriate

390 The first paragraph of the results section is repeated twice.

We are unsure of what precisely this refers to - the first paragraph of the results sections deals with C inputs, while the second deals with N, P and S inputs.

420 These graphs look much better! As a suggestion, generally gray-scale is preferred for accessibility and for readers who may print B+W only. I don't think the treatments (OG, CONV, CONV+WCC require colors).

We have edited all graphs to make them greyscale - thanks for the suggestion!

431 - Typically the alpha level is set prior to the analysis (eg, p = 0.05 or 0.1), this allows for consistent interpretation of the results. Here the ~3Mg ha "increase" is not significant at that level, so it should not be reported as an increase unless the p-value is set to above 0.26. Essentially, there was not high enough power to detect if the observed difference is real.

We have modified the wording to clarify the absence of any significant effect here, and reworded the title of Discussion section 4.4 to read "Compost + Cover Crops increased profile C stocks after 25 years, but Cover Crops alone did not"

455 Recommendation: Flip the y axis for Fig. 4 so the soil 'surface' 0 cm is at the top of the graph to make it more intuitive.

We are unsure of what precisely the reviewer means here: this graph shows "Depth Equivalent of Water", not moisture at different depths in the profile. We have modified the caption and label on the graph to attempt to clarify this.

465 Is that averaged across dates?

Yes, this is averaged across dates. We have clarified this in the wording.

Fig 9 a-d is separated by depth rather than date - Is it possible to present the nutrient data in the same way (Fig 6) - by depth rather than date? It seems the nutrient analysis was not repeated measures, but analyzed as an average over time (L 464). To me, it makes more sense for Fig. 6 to also just be an average over the growing season and presented at different depths. The depth patterns are set up as the most important for hypothesis testing in the introduction, and seasonal dynamics were not as important. Alternatively, you could also add an interesting hypothesis about seasonal dynamics.

We have added a hypothesis about seasonal dynamics in the introduction. Since measurements of soluble nutrients are highly variable during the year, we wanted to be sure that our observations of differences between the plots were not just due to conditions at a single timepoint. We hypothesized that the observed differences would persist throughout the growing season, and have added a sentence in the discussion to address this.

Fig 7 B - This is Mineral N, not total N. Please correct throughout the manuscript.

Corrected all graph labels

573-583 - Delete this paragraph or integrate it within the context of your results, rather than summarizing the literature as if for an introduction.

Changed layout of paragraph to begin with results

583 - Fig. 4 is the depth of moisture, not moisture content. If the volumetric moisture content is the important variable, then present that in the text, and put the water depth into supplemental.

We are unsure of what precisely the reviewer means here: this graph shows "Depth Equivalent of Water" which is a measure of moisture content in the profile similar to cm of rainfall. We have modified the caption and label on the graph to attempt to clarify this.

656 - SOC vs soil organic carbon - stay consistent

All "soil organic carbon" terms in the manuscript have been changed to SOC where appropriate

636-649: Cascade theory was introduced already in the introduction, so you can remove those lines here. Also this whole paragraph does not mention findings of this study. Please start the discussion with the present experimental findings and then bring in supporting information from the literature to help create a seamless discussion.

Changed layout of paragraph to begin with results

703 From Fig. 7B it looks like ORG subsoils did not have significantly different mineral N, so please explain how it was a 'higher value in ORG subsoils'. Also, how are you distinguishing higher movement of nutrients at depth from merely lower crop uptake?

We have modified the paragraph to represent the non-significance in mineral N content, and added a sentence referencing previous research at RR showing that cover crop treatments did not have significant effects on P availability or N use efficiency.

730 - Rather than "we attribute" perhaps "we observed/found evidence that the SOC…"

We have modified the sentence to read "We have found evidence that …"

735 Again, I would change the hypothesis to include something about temporal dynamics, or simply aggregate data across the four dates. Can you explicitly link the seasonal data to the objectives of the experiment?

We have modified the hypotheses to include temporal dynamics, highlighting how variable soluble nutrient measurements are, and the need to measure at multiple timepoints to account for this.

739 - In the response to the review, you stated there were no irrigation events during this study, but here state that nutrient transport depends on irrigation water. Can you please explain the time frame you are referring to here in the discussion?

We are unsure what the reviewer means by no irrigation events during the study - we were unable to find the corresponding comment in our original response. In the previous revision, we included a sentence in the Methods section stating "All plots were irrigated with subsurface drip at the time of sampling, having converted from furrow irrigation to subsurface drip in 2014. " We also included estimates of the amount of irrigation water applied to each plot in Supplementary Table A5.

741 The growing season stoichiometry is more suitable than what? Than off-season stoichiometry? Please reword the sentence.

The sentence has been reworded.

757 I would shift away from emphasizing microbial activity (in the first half of the sentence) to reflect the factor that changed during this study "This shift in irrigation and decrease in water inputs potentially increased..."

Edited

761 This is the first mention of tillage within this experiment. Please list in the methods/experimental design.

We have included a brief description of tillage in the methods, and included Supplementary Table A5 briefly outlining practices on the experimental plots for the 2018-2019 year.

764 "In turn, higher transport led to increased C stocks and reduced levels of microbial stress" This was supported by results, but also needs to be part of the hypothesis and objectives of the study to justify the methods used and inclusion of this as a main concluding point.

We have edited the hypotheses to include C stocks and variation in microbial biomass.

Reviewer 2

This manuscript entitled "Synergy between compost and cover crops in a Mediterranean row crop system leads to increased subsoil carbon storage" is a field-based experiment comparing the long-term (25 years) impact of different agricultural management practices on top- versus sub-soil carbon stocks. In addition to carbon stocks, the authors also assessed a variety of soil biological, chemical, and physical parameters to help explain the environmental processes leading to their observed soil carbon stocks results and thus lend support for testing their hypothesis.

Although this is the first time I have reviewed this manuscript, to my knowledge it has been reviewed at least once before, in which extensive edits by multiple reviewers were called for. When comparing the 'response to reviewers' document with the current version of the manuscript, I am happy to see that the authors have very carefully and thoroughly integrated the reviewers' comments into the manuscript, thus clarifying the majority of questions or inconsistencies from the earlier manuscript version. The resulting manuscript is very well-written, and the introduction, methods, results, and discussion are presented clearly and all well-integrated with each other. Below I give some minor suggestions for further improvement.

Thank you!

General comments:

Was it not possible to estimate root biomass and nutrient content in the crop and cover crop roots? This would of course influence overall C stocks but was not mentioned at all in the text.

We do not have measurements of the root biomass and nutrient content of the crop and cover crop roots, as the only measurements made were aboveground biomass. We have added a sentence in the Methods section to clarify that our figures show only aboveground biomass, as well as a sentence in the Discussion to highlight the potential oversight.

What were the differences in crop production between the different systems? Perhaps this could be mentioned as something to take into consideration (i.e. it is of course ideal to improve soil C, but it is also important from an agronomic perspective to ensure appropriate crop yields, which may not be in alignment with increasing soil C).

We have inserted a sentence highlighting the small difference in tomato yields between our systems in the introduction, and added a citation to support this.

Although you state it in the Results section, I suggest also mentioning in the Discussion section that the ORG system also had significantly more C added to the system, as I imagine this could have had an impact on subsoil C stocks as well as the overall nutrient mobility processes.

We have highlighted the larger amount of C added to ORG plots as a reason for increased EOC and total C stocks in sections 4.2 and 4.4.

Fig 1: the individual panels are not labeled a-d as it describes in the caption. I would also suggest arranging them either 2 X 2, or four side-by-side or stacked on top of each other. The current arrangement is a bit off-putting. Also, was there no phosphorus in the cover crop or crop residue biomass?

We have added the requested panel labels, but would like to keep the graph layout as-is due to the common legends for graphs B,C and D

I would suggest integrating Section 3.5 ("Aggregation") into a different section as there is only one sentence describing the results.

We have integrated the Aggregation section of the results into section 3.4.

Fig. 8: the individual panels are not labeled a-b as in the caption. Also, the figure itself is quite blurry. Is there a way to make it more clear (as in Fig. A3)?

We have added panel labels and changed the resolution of the figure to make it clearer.

Table 1: Is this the information shown in Fig. 8? If so, I would suggest only showing Fig. 8 in the main text, and putting this table in the Supplementary Materials.

The raw information used to make this table was indeed taken from the same measurements used to make Figure 8, but this table was added in response to a previous reviewer's comments highlighting the potential issues with visual estimation of spectral subtractions.

We believe that both figures add unique data and should both be included in the main body of the paper: the spectra provide an overall look at the compositional differences with treatment and time, while the table highlights a specific pair of peaks in the spectra that would be hard to otherwise visualize alone.

Specific comments:

L20: I would replace "the increased abundance. . ." with "an increased abundance. . ."

Edited.

---

## Author Response (AR3)

**Non-public comments to the Author:**

Please double check axis titles where mineral N is concerned. At least one axis says "Mineral N:NO3+NO4" which looks like a ratio. Please change to "Mineral N (NO3+ NH4)" where appropriate.

**We have changed all axes to read "Mineral N (NO3+NH4)"**

Also, please edit the line and error bar pattern in the line graphs. The dotted lines and dotted error bars are very hard to make out. Instead, please use solid lines of different colors or grayscale, or a dotted line with much closer spacing. Currently, wherever the lines overlap or where the error bars are small, it is very difficult to tell where they end.

**We have converted the line graphs to solid greyscale lines, as the dotted lines were not visible when the graph was inserted into the paper**

**Review Validiation Comments**

For the next revision, I kindly ask you to change the coloured symbol on page 17 and the coloured text in the sentence on page 21 of *.pdf manuscript file with black ones.

**We have removed the coloured symbols on page 17 and the colored text on page 21 of the manuscript.**